# The Quotient Bayesian Learning Rule

**Mykola Lukashchuk**[1*]      **Raphaël Trésor**[1]      **Wouter W. L. Nuijten**[1]

**İsmail Şenöz**[2]      **Bert de Vries**[1,3]

[1]Department of Electrical Engineering, Technical University of Eindhoven, the Netherlands
[2]Lazy Dynamics, Utrecht, the Netherlands
[3]GN Hearing, Eindhoven, the Netherlands
{m.lukashchuk,r.v.tresor,w.w.l.nuijten,bert.de.vries}@tue.nl
isenoz@lazydynamics.com

## Abstract

This paper introduces the Quotient Bayesian Learning Rule, an extension of natural-gradient Bayesian updates to probability models that fall outside the exponential family. Building on the observation that many heavy-tailed and otherwise non-exponential distributions arise as marginals of minimal exponential families, we prove that such marginals inherit a unique Fisher–Rao information geometry via the quotient-manifold construction. Exploiting this geometry, we derive the Quotient Natural Gradient algorithm, which takes steepest-descent steps in the well-structured covering space, thereby guaranteeing parameterization-invariant optimization in the target space. Empirical results on the Student-$t$ distribution confirm that our method converges more rapidly and attains higher-quality solutions than previous variants of the Bayesian Learning Rule. These findings position quotient geometry as a unifying tool for efficient and principled inference across a broad class of latent-variable models.

## 1 Introduction

Statistical models with heavy-tailed likelihoods are indispensable when data contain outliers or extreme values that violate Gaussian assumptions. A prime example is the Student-$t$ distribution: its degrees-of-freedom parameter lets the tails stretch or contract, providing the robustness practitioners require.

Fitting such models is considerably harder than specifying them. The latent-scale representation that makes the Student-$t$ analytically convenient also renders Expectation–Maximization painfully slow in high dimensions, while naïve gradient methods stumble on the strong curvature induced by heavy tails. We therefore seek an algorithm that (i) preserves the full tail flexibility of the Student-$t$ *and* (ii) exploits the well-behaved geometry enjoyed by exponential-family (EF) distributions.

The Bayesian Learning Rule (BLR) of Khan and Rue [2023] offers a natural starting point: it frames inference as gradient ascent in distribution space and, when the candidate posterior is an EF member, replaces ill-conditioned Euclidean steps with natural-gradient updates that follow Fisher geodesics. In its manifold formulation [Lin et al., 2020b], the EF's natural parameters form a Riemannian manifold equipped with the Fisher information metric, yielding both elegant theory and fast convergence. Unfortunately, Student-$t$ distributions lie *outside* the exponential family, so standard BLR cannot be applied directly.

The novel extension of the BLR, The "Lie-group BLR" [Kiral et al., 2023] addresses some non-EF cases by using group actions, but the Lie-group BLR framework has yet to be extended to multivariate settings—a significant limitation that our work specifically overcomes while maintaining the desirable information-geometric properties of the original BLR formulation.

39th Conference on Neural Information Processing Systems (NeurIPS 2025).

The central insight motivating our work comes from a fundamental property of the Student-$t$ distribution: it can be represented as the marginal of a Normal-Wishart distribution, the so-called scale-mixture structure, first studied by Andrews and Mallows [1974], where posterior candidates are parametrized through a latent "scale variable" that transforms an arbitrary base distribution. Normal-Wishart is an Exponential Family distribution. Moreover, Normal-Wishart distribution possesses the minimal exponential family parametrization. This representation has been leveraged in various contexts, from mixture modeling [Peel and Mclachlan, 2000] to robust regression [Lange et al., 1989], primarily to facilitate EM-style algorithms through data augmentation.

We take this insight in a new direction by exploring its implications for the geometric structure of the parameter space. Specifically, we observe that this marginalization relationship naturally induces a quotient manifold structure, where the Student-$t$ manifold can be viewed as a quotient of the Normal-Wishart manifold under an equivalence relation defined by identical marginalized distributions.

Our key theoretical contribution lies in showing that the Fisher-Rao metric, which defines a natural Riemannian structure on statistical manifolds, can be extended from the Normal-Wishart manifold to the Student-$t$ manifold through this quotient relationship. Furthermore, by carefully choosing a base measure and a family of scaling distributions in the scale-mixture, a wide range of non-EF models can be captured in this unified framework [Barndorff-Nielsen et al., 1982], enabling robust Bayesian updates generalizing our approach beyond the Student-$t$.

More precisely, we prove that if a distribution is a marginal of a minimal exponential family, then its parameter space inherits a unique Fisher information metric structure as a quotient Riemannian manifold.

Building on this theoretical foundation, we propose an extension of the BLR that leverages the scale mixture representation and the quotient manifold structure. This insight leads us to develop the "Quotient Natural Gradient" algorithm, which efficiently optimizes on the Student-$t$ manifold using horizontal lifts between manifolds. Our approach computes steps in the well-structured Normal-Wishart space and maps them appropriately to the Student-$t$ parameter space through the established quotient relationship. In the remainder of this paper, we formalize these concepts, develop the necessary mathematical framework, and evaluate our approach empirically. We compare the Quotient Natural Gradient against both standard EM and naïve manifold optimization, demonstrating its advantages in terms of convergence speed and solution quality. Our results highlight the practical value of this geometric perspective and suggest broader applications to other statistical models with similar latent variable structures.

## 2 Background and problem setup

### 2.1 Bayesian learning rule

Given a model parameter space $\mathcal{Z}$ and a loss $l(\mathbf{z})$, the *Bayesian Learning Rule* (BLR; Khan and Rue [2023]) optimizes over distributions rather than point estimates

$$q^* = \arg\min_{q \in \mathcal{Q}} \mathbb{E}_q [l] - \tau \mathbb{H}[q] = \arg\min_{q \in \mathcal{Q}} -\mathcal{L}[q], \tag{1}$$

where $\mathcal{Q} = \{q_\xi \mid \xi \in \Xi \subset \mathbb{R}^d\}$ parametrizes candidate posteriors, $\mathbb{H}[q]$ denotes the Shannon entropy, and $\tau > 0$ is a temperature. In other words, BLR minimizes negative ELBO, or maximizes ELBO (we stick to maximization convention); just for the re-use in the future, we will define

$$\mathcal{L}[q] = \tau \mathbb{H}[q] - \mathbb{E}_q [l]. \tag{2}$$

The key component of the BLR is the use of the natural gradient Amari [1998] in place of the naïve Euclidean updates. Euclidean gradients ignore the underlying geometry of the set $\mathcal{Q}$. *Natural-gradient descent* Amari [1998] instead preconditions the gradient by the inverse Fisher information $F^{-1}(\xi)$, yielding steps of constant KL length and trajectories that are invariant to re-parameterization. More formally, the natural gradient update is given by

$$F(\xi) = \mathbb{E}_{q_\xi} \left[ \nabla_\xi \log q_\xi(\mathbf{z}) \nabla_\xi \log q_\xi(\mathbf{z})^\top \right], \tag{3a}$$

$$\widetilde{\nabla}_\xi l := F(\xi)^{-1} \nabla_\xi l(\xi), \tag{3b}$$

$$\xi_{t+1} = \xi_t - \alpha_t \widetilde{\nabla}_\xi l(\xi)\big|_{\xi=\xi_t}. \tag{3c}$$

The obstacle in (3b) is computing and inverting $F(\xi)$, an $\mathcal{O}(n^3)$ operation that quickly becomes prohibitive (for a full $d$-dimensional Gaussian, $n = \mathcal{O}(d^2)$ and the cost is $\mathcal{O}(d^6)$).

For the BLR objective (1), this cost disappears when $q_\lambda$ is a member of the minimal, regular exponential family, which means that $\Lambda$ is an open subset of the Euclidean space and the sufficient statistics maintain independence [Jordan and Sejnowski, 2001, Chapter 3]. The distribution $q_\lambda$ belongs to the exponential family if

$$q_\lambda(\mathbf{z}) = h(\mathbf{z}) \exp\left(\boldsymbol{\lambda}^\top T(\mathbf{z}) - A(\boldsymbol{\lambda})\right) \tag{4}$$

holds, where $h$ is the base measure; $T$ is the sufficient statistic; $\boldsymbol{\lambda}$ are the natural parameters, and $A$ is the log-partition function

$$A(\boldsymbol{\lambda}) = \log \int_{\mathcal{Z}} h(\mathbf{z}) \exp\left(\boldsymbol{\lambda}^\top T(\mathbf{z})\right) \mathrm{d}\mathbf{z}, \tag{5}$$

ensuring that $q_\lambda$ is a probability distribution. The dual (expectation) coordinate [1]

$$\boldsymbol{\theta} = \nabla_\lambda A(\boldsymbol{\lambda}) \tag{6}$$

yields the following gradient identity

$$\widetilde{\nabla}_\lambda \mathcal{L}(\boldsymbol{\lambda}) = \nabla_\theta \mathcal{L}_*(\boldsymbol{\theta}), \tag{7}$$

where $\mathcal{L}_*(\boldsymbol{\theta}) = \mathcal{L}\big(\boldsymbol{\lambda}(\boldsymbol{\theta})\big)$ is the objective expressed in expectation parameters [Khan and Nielsen, 2018, Thm. 1]. No matrix inversion is required: one simply computes an ordinary gradient in $\theta$. Unless explicitly stated otherwise, we assume that all exponential families considered in this paper are minimal and regular.

## 2.2 Reparameterization through marginalization

To make the general concepts of our construction more visible to the reader, we will refer to a running example, the so-called Normal-Gamma distribution, which serves as a univariate preparation for the multivariate Normal-Wishart that is the main example underlying our experiments. Readers seeking an even simpler introduction may first consult Appendix A, where the two-dimensional Negative Binomial example illustrates the quotient geometry with transparent visualizations.

The Normal-Gamma distribution is defined as

$$z \mid \tau \sim \mathcal{N}\big(\mu, \, (\sigma^{-1}\tau)^{-1}\big) \tag{8a}$$
$$\tau \sim \mathcal{G}(\alpha, \beta). \tag{8b}$$

The Normal-Gamma distribution is a four-parameter distribution that defines a joint over two variables $z$ and $\tau$. This four-parameter distribution defining a joint over variables $z$ and $\tau$ exemplifies the broader class of scale-mixture distributions [Andrews and Mallows, 1974] that forms the foundation of our approach. More importantly, in the current context, is that the Normal-Gamma distribution (8) is a minimal exponential family distribution and its marginal over $z$ is a Student-$t$ distribution, which lies outside of the exponential family. The mapping from the standard parametrization to the natural parametrization is given by

$$\boldsymbol{\lambda} = \left(\sigma^{-1}\mu, -\frac{\sigma^{-1}}{2}, \alpha - \frac{1}{2}, -\beta - \frac{\sigma^{-1}\mu^2}{2}\right). \tag{9}$$

The complete exponential family representation of the Normal-Gamma distribution is provided in Appendix B.5, Equation (54).

More generally, consider a joint exponential family density on $\mathbf{z}_{\text{ext}} = (\mathbf{z}, \mathbf{z}_V)$,

$$q_\lambda(\mathbf{z}_{\text{ext}}) = h(\mathbf{z}_{\text{ext}}) \exp\big(\boldsymbol{\lambda}^\top T(\mathbf{z}_{\text{ext}}) - A(\boldsymbol{\lambda})\big), \qquad \boldsymbol{\lambda} \in \Lambda \subset \mathbb{R}^d,$$

where $\mathbf{z} \in \mathbb{R}^d$, $\mathbf{z}_V \in \mathbb{R}^{d_V}$, $d = d + d_V$. Marginalizing over $\mathbf{z}_V$ defines

$$q_\xi(\mathbf{z}) = \int q_\lambda(\mathbf{z}, \mathbf{z}_V) \, \mathrm{d}\mathbf{z}_V, \tag{10}$$

---

[1]See Amari [2016, Chap. 6] for a thorough treatment of the dual affine structure.

and hence a surjection

$$\pi : \Lambda \longrightarrow \Xi, \qquad \boldsymbol{\lambda} \longmapsto \xi(\boldsymbol{\lambda}). \tag{11}$$

For the running Normal–Gamma example (8) we have a correspondence $\mathbf{z} = z$ and $\mathbf{z}_V = \tau$. Writing the natural-parameter vector as $\boldsymbol{\lambda} = (\lambda_1, \lambda_2, \lambda_3, \lambda_4)$ [cf. Eq. (9)], the projection $\pi$ maps the four-dimensional Normal–Gamma space onto the three Student-$t$ parameters $\xi = (\mu, \sigma^2, \nu)$ via

$$\xi = \pi(\boldsymbol{\lambda}) \;=\; \left( \tfrac{\lambda_1}{-2\lambda_2}, \; \tfrac{-\lambda_4 + \lambda_1^2/(4\lambda_2)}{-2\lambda_2\,(\lambda_3 + 1/2)}, \; 2\lambda_3 + 1 \right). \tag{12}$$

Because many distinct $\boldsymbol{\lambda}$'s yield the same triplet $(\mu, \sigma^2, \nu)$, this exemplifies that in general the $\pi$ is not a bijection. But we obtained a minimal exponential family reparameterization of our marginal that lies outside of the exponential family.

## 3  Marginal quotient structure

Natural-gradient steps are cheap in the joint exponential-family space $\Lambda$ but expensive in the marginal coordinates $\Xi$, because the Fisher inverse can be avoided due to relation (7). Our plan is therefore to run the natural gradient scheme completely in $\Lambda$ and afterwards marginalize the result of our procedure $\boldsymbol{\lambda}^*$ by sending it back to $\pi(\boldsymbol{\lambda}) \in \Xi$.

However, our aim is to minimize the BLR objective (1) in the marginal parameter space $\Xi$, rather than in the full natural-parameter space $\Lambda$. This raises two questions:

(i) Is the outcome of the gradient scheme independent of the choice of representative $\boldsymbol{\lambda} \in \pi^{-1}(\xi)$?

(ii) Does running natural–gradient descent in $\Lambda$ and marginalizing each $\boldsymbol{\lambda}_t$ (where $t$ is the interate of the gradient scheme) actually minimize $-\mathcal{L}(\xi)$ in the marginal coordinates $\Xi$?

The resolution hinges on *quotient topology*. Specifically, the marginal parameter space $\Xi$ can be viewed as the quotient set $\Lambda/\sim_\pi$ defined by the following equivalence relation:

$$\boldsymbol{\lambda}_1 \sim_\pi \boldsymbol{\lambda}_2 \iff \pi(\boldsymbol{\lambda}_1) = \pi(\boldsymbol{\lambda}_2), \qquad \boldsymbol{\lambda}_1, \boldsymbol{\lambda}_2 \in \Lambda. \tag{13}$$

The equivalence classes (elements) of $\Lambda/\sim_\pi$ are usually called *fibres*. We will align with this convention.

The quotient manifold theory ensures us that (ii) is resolved if $\Lambda/\sim_\pi$ is a Riemannian quotient manifold and we project the gradient on the horizontal space with respect to $F(\boldsymbol{\lambda})$ [Boumal, 2023][Chap. 9.9 and Def. 9.24]. A small background on quotient manifold theory is provided in Appendix B.

$\Lambda$ is an open subset of a Euclidean space, and, by the moment parametrization assumption on $\Xi$ (see Definition 1), $\Xi$ is an embedded submanifold of a Euclidean space. Under this assumption, $\pi$ is a smooth map between two embedded submanifolds, so the horizontal subspace can be simply expressed as

$$\mathcal{H}_\lambda = \big( \ker D\pi(\boldsymbol{\lambda}) \big)^{\perp_{F_\lambda}},$$

where $D\pi(\boldsymbol{\lambda})$ is the differential of the smooth map between two Euclidean spaces (see Boumal, 2023[Proposition 3.35]), and the orthogonal operator is taken according to the Riemannian metric (Fisher metric) of the manifold $\Lambda$ [Boumal, 2023, Def. 3.10].

**Definition 1** (Moment-parametrized family). *Let $\mathcal{Q} = \{q_\xi : \xi \in \Xi\}$ be a $k$-dimensional family of probability densities on a measurable space $\mathcal{Z}$. We call $\mathcal{Q}$ moment-parametrized if there exist measurable moment functions $m_1, \ldots, m_k : \mathcal{Z} \to \mathbb{R}$ such that*

(i) *$\Xi$ is an embedded $k$-dimensional submanifold of $\mathbb{R}^d$;*

(ii) *For every $\xi \in \Xi$ the expectations $e^i(\xi) = \mathbb{E}_{q_\xi}[m_i(\mathbf{z})]$ exist and are finite;*

(iii) *The mapping $e : \Xi \to \mathbb{R}^k$, $\xi \mapsto (e^1(\xi), \ldots, e^k(\xi))$ is a smooth bijection whose Jacobian has full rank $k$ everywhere on $\Xi$.*

*We refer to $\xi$ (or $m(\xi)$) as the* moment coordinates *of $q_\xi$.*

Definition 1 can be understood as a labeling of each distribution by the values of finitely many expectations (e.g. the mean, the variance, the skewness, . . . ) where those expectations vary smoothly and uniquely according to $\Xi$.

Many common families—including all Student-$t$'s with degrees of freedom $\nu > 1$—fit the pattern of Definition 1, but some heavy-tailed laws such as the Cauchy ($\nu = 1$) do not because their first moments are undefined.

Theorem 1 resolves (i) from our problem statement because the theorem states that $\Xi$ is the quotient manifold of $\Lambda$. A proof of Theorem 1 is given in Subsection B.3 of Appendix B.

**Theorem 1** (Marginalization yields a smooth quotient manifold). *Let $q_\lambda$ be a minimal, regular exponential family with parameter space $\Lambda \subset \mathbb{R}^d$. Suppose a partition $\mathcal{Z}_{ext} = (\mathcal{Z}, \mathcal{Z}_V)$ is chosen so that the marginal family $\{q_\xi\}_{\xi \in \Xi}$ obtained via $\pi : \Lambda \to \Xi$ is moment-parametrized (Definition 1). Then $\Xi$ is the quotient manifold of $\Lambda$ induced by $\pi$.*

Point (ii) is settled by Theorem 2. Theorem 2 shows that $\Xi$ is the Riemannian quotient manifold of $\Lambda$ under the Fisher–Rao metric and that the induced quotient metric coincides with the Fisher information metric of the marginal family itself. Putting the pieces together, Theorem 2 shows that running the natural gradient in $\Lambda$ projected on the horizontal space $\mathcal{H}_\lambda$ is equivalent to running the natural gradient in $\Xi$. The full proof is given in Subsection B.4 of Appendix B.

**Theorem 2** (Induced Fisher–Rao metric). *Assume the setting of Theorem 1 and equip the natural-parameter space $\Lambda$ with its Fisher information metric $F_\lambda$. Then:*

    *(i) The map $\pi$, that project the Riemannian manifold $(\Lambda, F_\lambda)$ on $\Xi$, induces a Riemannian quotient manifold structure on $\Xi$;*

    *(ii) The Riemannian quotient metric on $\Xi$ is then the Fisher metric of $\Xi$.*

Summarizing, our approach replaces a minimization of $\mathcal{L}$ by the natural gradient descent in $\Xi$ by a natural gradient in $\Lambda$ where at each step the gradient vector is projected onto the horizontal space as follows:

$$P_{\left(\ker D\pi(\lambda)\right)^{\perp_{F_\lambda}}} \widetilde{\nabla}_\lambda \mathcal{L}(\boldsymbol{\lambda}), \tag{14}$$

where $P$ is the orthogonal projection for metric $\langle \cdot\,;\,\cdot \rangle_{F_\lambda}$. Theorems 1 and 2 ensure us of a mathematical equivalence between the two approaches.

## 4  The quotient Bayesian learning rule

We now translate the quotient–manifold theory developed in Theorems,1–2 into a concrete optimization procedure for evidence–lower–bound (ELBO) maximization. Throughout this section let $q_\xi(\mathbf{z})$, $\xi \in \Xi$ denote the *marginal* variational family in which we ultimately seek an optimum, and pick $\xi_0 \in \Xi$ such that the prior factor of the model can be written $p(\mathbf{z}) = q_{\xi_0}(\mathbf{z})$.

Assume that the marginal distribution $q_\xi(\mathbf{z})$ arises by marginalizing a minimal, regular exponential family $q_\lambda(\mathbf{z}, \mathbf{z}_V)$, parameterized by natural parameters $\boldsymbol{\lambda} \in \Lambda$, over the extended latent variable $\mathbf{z}_{\text{ext}} = (\mathbf{z}, \mathbf{z}_V)$ (see partition (10)). The map $\pi : \Lambda \to \Xi$, $\boldsymbol{\lambda} \mapsto \xi$, induced by this marginalization is precisely the marginalization map defined earlier.

Choose a representative $\boldsymbol{\lambda}_0 \in \pi^{-1}(\xi_0)$ of the prior and define

$$\mathcal{L}(\boldsymbol{\lambda}) = \mathbb{E}_{q_\lambda}\big[\log q_{\lambda_0}(\mathbf{z}, \mathbf{z}_V) - \log q_\lambda(\mathbf{z}, \mathbf{z}_V)\big] + \sum_{i=1}^{N} \mathbb{E}_{q_\lambda}\big[\log p(x_i \mid \mathbf{z})\big]. \tag{15}$$

Because $\mathcal{L}(\boldsymbol{\lambda})$ is constant along every fibre $\pi^{-1}(\xi)$, moving within a fibre—that is, in a "vertical" direction belonging to $\ker D\pi(\boldsymbol{\lambda})$—changes only the *parameterisation*, not the marginal distribution. By projecting each gradient step onto the Fisher-orthogonal complement $\mathcal{H}_\lambda$ via the operator (16), we ensure that every update alters $\boldsymbol{\lambda}$ solely through its image $\xi = \pi(\boldsymbol{\lambda})$. Hence the optimization trajectory produced in the joint space coincides exactly with the one obtained by running natural-gradient ascent on $\mathcal{L}(\xi)$ in $\Xi$.

Let $\boldsymbol{\theta} = \nabla_\lambda A(\boldsymbol{\lambda})$ denote the expectation parameters of $q_\lambda$. For minimal exponential families the ordinary gradient $\nabla_\theta \mathcal{L}(\boldsymbol{\lambda})$ coincides with the natural gradient in the joint space; see Khan and Nielsen [2018]. We therefore

    (i)  compute $\nabla_\theta \mathcal{L}(\boldsymbol{\lambda})$,

(ii) identify $\widetilde{\nabla}_{\boldsymbol{\lambda}}\mathcal{L} = \nabla_{\theta}\mathcal{L}(\boldsymbol{\lambda})$ via the duality between $\theta$ and $\boldsymbol{\lambda}$,

(iii) project $\widetilde{\nabla}_{\boldsymbol{\lambda}}\mathcal{L}$ onto the horizontal subspace $\mathcal{H}_{\lambda} = (\ker \mathrm{D}\pi(\boldsymbol{\lambda}))^{\perp}$,

(iv) take a step of size $\beta_t$ in that horizontal direction.

Because the horizontal space is orthogonal to the fibers $\pi^{-1}(\xi)$, each update stays within a *single* equivalence class in $\Lambda$, thereby realizing the quotient–natural–gradient flow guaranteed by Theorem 2. The procedure terminates when the horizontal component of the gradient falls below a tolerance $\epsilon$. At convergence, the optimizer $\lambda^*$ is mapped back to the marginal space via $\xi^* = \pi(\boldsymbol{\lambda}^*)$, yielding the desired posterior approximation $q_{\xi^*}(\mathbf{z})$. The complete routine is summarized in Algorithm 1, which we call the quotient Bayesian learning rule (QBLR).

An immediate question is how to make the step (III) in the above scheme efficient. Let $V_{\lambda} := \ker D\pi(\boldsymbol{\lambda})$ be the *vertical* sub-space and the differential of the marginalization map $J_{\pi}(\boldsymbol{\lambda})$, then its right null-space is the vertical subspace of our quotient. Pick some matrix $K(\boldsymbol{\lambda})$ that forms a basis of the $V_{\lambda}$. Then the projection on the horizontal space (in the Fisher-Rao geometry) can be formed by

$$P_{\mathcal{H}}(\boldsymbol{\lambda}) = \mathrm{I} - K(\boldsymbol{\lambda})\big[K(\boldsymbol{\lambda})^{\top} F(\boldsymbol{\lambda}) K(\boldsymbol{\lambda})\big]^{-1} K(\boldsymbol{\lambda})^{\top} F(\boldsymbol{\lambda}). \qquad (16)$$

A short algebraic derivation of the identity (16) is provided in Appendix B, Subsection B.2.

Crucially, the inversion involves only the $\dim V_{\lambda} \times \dim V_{\lambda}$ matrix $K^{\top} F K$; for the Normal–Wishart case $\dim V_{\lambda} = 1$ (Appendix C, Subsection C.2), so (16) collapses to a single scalar divide, and no full Fisher inversion is ever required. The general computational analysis of the expression (16) is given in Appendix E.1.

---

**Algorithm 1** The Quotient Bayesian Learning Rule

---

**Input:** lifted prior parameters $\boldsymbol{\lambda}_0$, canonical projection $\pi : \Lambda \to \Xi$, data set $\mathcal{D} = \{x_i\}_{i=1}^{N}$, ELBO defined in the lifted space $\mathcal{L}(\boldsymbol{\lambda})$ (15), step–size schedule $\{\beta_t\}_{t \geq 0}$, tolerance $\epsilon$

1: $\boldsymbol{\lambda} \leftarrow \boldsymbol{\lambda}_0$         ▷ initialize in the lifted (joint) space
2: **repeat**
3:      $g_{\theta} \leftarrow \nabla_{\theta}\mathcal{L}(\boldsymbol{\lambda})$      ▷ compute natural gradient through the dual coordinates Eq. (6)
4:      $g_{\lambda}^{\perp} \leftarrow \mathrm{Proj}_{\ker \pi^{\perp}}(g_{\theta})$      ▷ project onto the horizontal space, defined in Eq. (16)
5:      $\boldsymbol{\lambda} \leftarrow \boldsymbol{\lambda} + \beta_t\, g_{\lambda}^{\perp}$      ▷ natural-gradient ascent step
6: **until** $\|g_{\lambda}^{\perp}\|_2 < \epsilon$
7: $\xi^* \leftarrow \pi(\boldsymbol{\lambda})$
8: **return** marginal variational posterior $q_{\xi^*}(\cdot)$

---

## 5   Student-$t$ via Normal-Wishart representation

In this section, we present an alternative approach to heavy-tailed posterior approximation using the Normal-Wishart scale mixture representation. While Lin et al. [2020a] developed updates for Student-$t$ distributions through a curved exponential family formulation using the Normal-Inverse Gamma scale mixture, our approach leverages the quotient manifold structure induced by the marginalization map from the Normal-Wishart to the Student-$t$ manifold. We first introduce the Normal-Wishart parameterization and derive the explicit marginalization mapping to the Student-t distribution. Then, we develop natural gradient updates that exploit the geometric structure of this mapping, avoiding the need for reparameterization tricks. We demonstrate how our method retains the computational efficiency of exponential family updates while capturing the heavy-tailed nature of the Student-t distribution, comparing our approach with Lin's Normal-Inverse Gamma formulation both theoretically and empirically.

### 5.1   Comparing parameterization approaches

The fundamental difference between our approach and that of Lin et al. [2020a] lies in how we represent the Student-$t$ distribution. Lin's approach reparameterizes the Student-$t$ as a curved exponential family, ensuring a one-to-one correspondence between the scale mixture parameter space and the distribution space. Their key insight was finding a specific parameterization that maintains this one-to-one correspondence, but at the cost of working with a curved (non-minimal) exponential family.

In contrast, our approach begins with the Normal-Wishart distribution, which is a minimal exponential family distribution (see Appendix C). When marginalized, this yields the multivariate Student-$t$ distribution through a many-to-one mapping, creating a quotient manifold structure. We can work directly in the unconstrained minimal exponential family space, leveraging its well-understood geometric properties. The quotient structure allows us to handle the redundancy in parameterization through the horizontal space projection.

The critical trade-off between these approaches can be summarized as follows:

$$\text{Lin et al. [2020a]}: \quad \text{Curved Exponential Family} \leftrightarrow \text{Student-}t \tag{17a}$$

$$\text{Ours}: \quad \text{Minimal Exponential Family} \xrightarrow{\text{quotient}} \text{Student-}t \tag{17b}$$

Mathematically, these approaches are represented as

$$\text{Lin (NIG):} \quad \begin{cases} p(x|w) \sim \mathcal{N}(\mu, w\Sigma) \\ p(w) \sim \text{InvGamma}(\nu, \nu) \end{cases} \xleftrightarrow{\text{one-to-one}} \quad \mathcal{T}(x|\mu, \Sigma, \nu) \tag{18}$$

$$\text{Ours (NW):} \quad \begin{cases} p(x|S) \sim \mathcal{N}(\mu, (\kappa S)^{-1}) \\ p(S) \sim \text{Wishart}(\nu', \Psi) \end{cases} \xrightarrow{\text{quotient}} \quad \mathcal{T}\left(x\Big|\mu, \frac{\Psi^{-1}}{\kappa(\nu'-d+1)}, \nu'-d+1\right) \tag{19}$$

Instantiating the generic QNG-VI template (Algorithm 1) with the Normal–Wishart lift yields Algorithm 2, the algorithm is provided in Appendix C. Following the scalar-NIG construction of Lin et al. [2020a], we apply the Bonnet- and Price-theorem analogues developed in Appendix C.4 to the parameters $\mu$, $\kappa$, and $\Psi$, obtaining an *unbiased* stochastic natural gradient on the corresponding quotient manifold. For the shape parameter $\nu$, we construct an unbiased gradient estimator with the Implicit Reparameterization Trick of Figurnov et al. [2018].

For each sample $z_n$ we draw an auxiliary scale matrix $\Lambda_n \sim \mathcal{W}_d(\nu, \Psi)$, couple it with the latent vector $z_n$, accumulate the data-fit gradients in the natural parameters $(\lambda_{1:4})$, add the analytic prior terms, and convert the result to expectation-space via the chain-rule identities in Eqs. (70a)–(70d). The stochastic natural gradient is then *projected onto the horizontal subspace* (Alg. 2, Step 3) before a single ascent step in $(\lambda_{1:4})$ is back-transformed to $(\mu, \Psi, \kappa, \nu)$.

Because every intermediate quantity depends on $\Psi$ and $\kappa$ only through the quotient-invariants

$$\widetilde{\mathbf{S}} := \frac{\Psi^{-1}}{\kappa} + \mu\mu^\top, \qquad \gamma := \mu^\top \widetilde{\mathbf{S}}^{-1}\mu,$$

the update is *representation-invariant*: any smooth reparameterization that preserves the marginal Student-$t$—e.g. the joint rescaling $(\Psi, \kappa) \mapsto (\Psi/c, c\kappa)$ with $c > 0$—produces the identical step on the Student-$t$ manifold. The resulting trade-offs vis-à-vis the curved-NIG scheme of Lin et al. [2020a] are summarised in Table 1.

## 6 Experimental validation

The full, version-pinned codebase that recreates every number in Table 2 is archived at `https://anonymous.4open.science/r/MIRWB-C735`. A line-by-line description of the training pipeline, hardware, and hyper-parameters is given in Appendix D; all information needed for exact re-execution therefore lives in one place and does not clutter the main text. All experiments were conducted on a MacBook Pro (2021) equipped with an Apple M1 Pro chip and 32 GB of memory.

We benchmark three variational-inference (VI) optimisers that operate on the *same* Bayesian logistic-regression model:

1. **BBVI-NS** – the score-function-free black-box VI variant of Roeder et al. [2017];
2. **NG-LIN** – the natural-gradient approach of Lin et al. [2020a];
3. **NG-Ours** – the quotient natural-gradient optimizer introduced in this work, using a Normal–Wishart marginal representation.

. We run the methods for four different datasets that are taken from the UCI/OpenML repository:

- **Breast Cancer Wisconsin (Diagnostic)** – 569 samples, 30 features [Wolberg et al., 1993].

| Aspect | Lin et al. (scalar NIG) | Ours (quotient-NG, NW) |
|---|---|---|
| Scale-mixture lift | $\mathcal{N}(z \mid \mu, w\Sigma)\,\mathrm{IG}(w|\nu,\nu)$ | $\mathcal{N}(z \mid \mu, (\kappa S)^{-1})\,\mathcal{W}(\nu, S)$ |
| Minimality of joint EF | curved, rank-3 | **minimal**, rank-4 |
| Parameter-invariance | only to linear re-labelling of same coords | **any** smooth parametrisation (log-scale, NG, etc.) |
| Tail expressiveness | one scalar $w \Rightarrow$ isotropic kurtosis | per-direction (matrix) kurtosis |
| Need explicit $F^{-1}$ | no (mean-grad trick) | **no** (mean-grad trick + 2-scalar projection) |
| Extra work vs. Lin | – | one outer-product ($O(d^2)$) |
| Limit $\nu \to \infty$ | behavior unknown | smoothly becomes Gaussian NG |

**Table 1:** Compact comparison of Lin's Student-$t$ update and our representation-invariant quotient natural-gradient step.

- **Pima Indians Diabetes** – 442 samples, 10 features [Smith et al., 1988].
- **Sonar (Mines vs. Rocks)** – 208 samples, 60 features [Gorman and Sejnowski, 1988].
- **Spambase** – 4 601 samples, 57 features [Hopkins et al., 1999].

Each dataset is split 80:20 (stratified) and feature-standardized using training statistics only.

For every (dataset, method) pair we report test-set accuracy of the *posterior mean* together with the empirical standard deviation estimated from ten posterior samples; see Table 2.

NG-Ours matches or surpasses BBVI-NS on three of the four benchmarks while requiring roughly one-tenth as many optimization iterations. The advantage is most striking on **Sonar**, where the richer Normal–Wishart marginal representation lifts accuracy significantly higher over BBVI-NS and NG-LIN, confirming the benefit of a geometry-aware update coupled with a more expressive variational family. Moreover, BBVI-NS marginals collapsed, so we do not benefit from the Bayesian procedure; we did obtain a collapsed estimate.

| Method | Metric | Breast cancer | Diabetes | Sonar | Spambase |
|---|---|---|---|---|---|
| BBVI-NS | Mean | $0.9314 \pm 0.0210$ | $0.7494 \pm 0.0473$ | $0.7951 \pm 0.1760$ | $0.8894 \pm 0.0078$ |
| | Sample | $0.9314 \pm 0.0210$ | $0.7494 \pm 0.0473$ | $0.7951 \pm 0.1760$ | $0.8894 \pm 0.0078$ |
| | Entropy | $0.0000 \pm 0.0000$ | $0.0000 \pm 0.0000$ | $0.0000 \pm 0.0000$ | $0.0000 \pm 0.0000$ |
| NG-LIN | Mean | $0.8919 \pm 0.0391$ | $0.7022 \pm 0.0356$ | $0.7476 \pm 0.0502$ | $0.8904 \pm 0.0089$ |
| | Sample | $0.9214 \pm 0.0209$ | $0.7526 \pm 0.0260$ | $0.8150 \pm 0.0116$ | $0.8906 \pm 0.0090$ |
| | Entropy | $0.0696 \pm 0.0021$ | $0.0026 \pm 0.0040$ | $0.0905 \pm 0.0010$ | $0.0112 \pm 0.0019$ |
| NG-Ours | Mean | $0.9711 \pm 0.0194$ | $0.7292 \pm 0.0432$ | $0.9095 \pm 0.0417$ | $0.8891 \pm 0.0124$ |
| | Sample | $0.9599 \pm 0.0110$ | $0.7791 \pm 0.0232$ | $0.9142 \pm 0.0178$ | $0.9057 \pm 0.0076$ |
| | Entropy | $0.1751 \pm 0.0153$ | $0.1490 \pm 0.0100$ | $0.1863 \pm 0.0011$ | $0.1046 \pm 0.0060$ |

**Table 2:** Comprehensive evaluation of Bayesian logistic regression performance on four UCI/OpenML datasets. Each entry shows mean $\pm$ standard error across 10 train-test splits with adaptive learning rates. **Mean**: test accuracy using posterior-mean weights (MAP estimation); **Sample**: test accuracy averaged over 100 posterior weight samples (capturing parameter uncertainty); **Entropy**: predictive entropy over test outputs in nats (higher values indicate greater prediction uncertainty). BBVI-NS is the score-function-free black-box VI of Roeder et al. [2017]; NG-LIN is the natural-gradient method of Lin et al. [2020a]; NG-Ours is the quotient natural-gradient optimizer introduced in this work. Note that BBVI-NS collapses to near-point posteriors (entropy $\approx 0$), while NG-Ours maintains the highest uncertainty quantification and achieves superior sample-based accuracy.

## 7 Discussion

**Why horizontal-space projection matters.** Properly removing the vertical component of the stochastic natural gradient stabilizes training: with projection, the ELBO converges to higher ELBO values, whereas without it the optimization drifts and eventually blows up (Fig. 1). This empirical

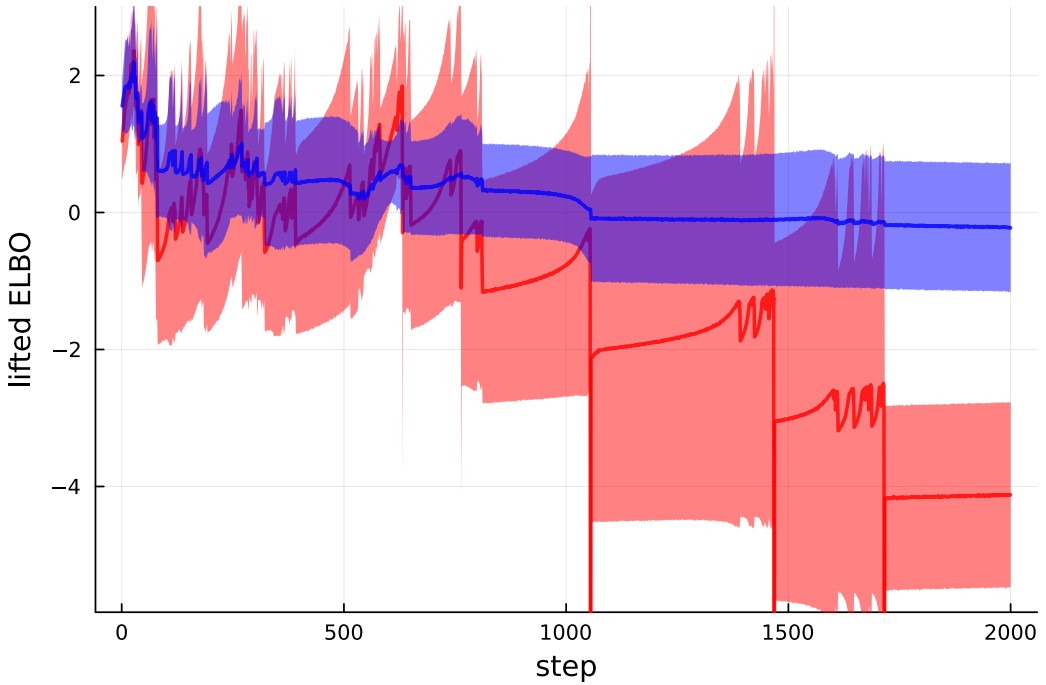

**Figure 1:** Comparison of lifted ELBO convergence with and without Fisher-orthogonal projection in the Poisson–Gamma lift of a Negative–Binomial target (detailed in Appendix A). We initialize five different representatives on the same fiber and optimize for 2000 iterations, estimating the lifted ELBO (15) with 5000 Monte Carlo samples at each step. Curves display the across-representative mean with a $\pm 1$ standard deviation ribbon; the $y$-axis is clipped to the 2–98% quantile range to suppress rare outliers. **With horizontal projection** (blue), optimization remains stable and attains higher ELBO values; **without projection** (red), the flow drifts along the fiber and eventually becomes unstable. The step-size schedule follows the Riemannian distance-over-gradients optimizer Dodd et al. [2024] with initial distance estimate 0.005.

result matches the theoretical analysis of §5: staying in the horizontal subspace keeps every iterate inside a single marginal equivalence class, preventing spurious motion along the gauge orbit.

**Position within the BLR landscape.** Conditional-EF methods of Lin et al. [2020a] rely on non-minimal embeddings and bespoke per-family updates, whereas our *quotient Bayesian learning rule* (QBLR; see §4) uses a minimal embedding and a single closed-form natural gradient for all Normal–Wishart scale mixtures. Lie-group BLR [Kiral et al., 2023] enforces manifold constraints through group actions while keeping the Fisher geometry implicit; the published instantiation handles diagonal covariances, although a full-covariance extension is, in principle, conceivable but has not yet been demonstrated.

**Toward mixture models.** Student-$t$ mixtures (GST-MMs) handle multimodal or heterogeneous data [Meitz et al., 2018, Revillon et al., 2017]. A drop-in combination of our horizontal projection with the variational mixture update of Minh et al. [2025] would yield a fully natural-gradient GST-MM: per-component Normal–Wishart factors follow our update, while the mixing weights use Minh *et al.*'s rule. Derivations and large-scale experiments are deferred to future work.

**Breadth of applicability and open challenge.** Scale mixtures, first systematised by Andrews and Mallows [1974] and greatly expanded by Barndorff-Nielsen et al. [1982], include the Laplace, exponential-power, and many other heavy-tailed families [West, 1987]. Whenever the scale kernel admits a *regular, minimal* exponential-family lift, the quotient structure of Eq. (13) emerges and QBLR applies unchanged. The principal remaining challenge is to construct such lifts for exotic priors—e.g. skewed or asymmetric heavy-tailed laws—so that our template can be used out of the box.

**Concluding remarks.** A single geometric ingredient—the Fisher-orthogonal projection onto the horizontal space—turns natural-gradient BLR into a stable, representation-free optimiser for a broad class of heavy-tailed Bayesian models. Respecting the quotient structure is therefore not a pedantic luxury but a practical necessity for reliable optimisation.

## 8 Conclusions

We introduced the *Quotient Bayesian Learning Rule* (QBLR), which extends natural-gradient variational updates to distributions that fall outside the exponential family yet arise as marginals of *minimal* exponential families. By casting the marginal parameter space as a Riemannian quotient, we showed that it inherits a unique Fisher–Rao metric and derived the associated *quotient natural gradient* (QNG). The algorithm performs steepest descent in the well-conditioned covering space, projects the update horizontally, and thereby preserves parameterization invariance. A closed-form Normal–Gamma/Student-$t$ example makes the construction concrete, and empirical results on Bayesian logistic regression demonstrate faster convergence and superior predictive calibration compared with earlier BLR variants. The same geometric template is readily transferrable to a wide class of scale-mixture priors and their mixture extensions, opening a path toward robust, heavy-tailed Bayesian learning at scale. While our method demonstrates strong geometric properties, its main limitation is computational complexity in high dimensions, which we suggest addressing through structured covariance proposals in Appendix E.2 and see as valuable future work.

### Acknowledgements

We gratefully acknowledge financial support by the Dutch Ministry of Economic Affairs (PPS funding), by the Dutch Research Council (NWO) and by hearing aid manufacturer GN Hearing, under contracts TKI-HTSM/21.0161/2112P09 (project: Auto-AR) and KICH3.LTP.20.006 (Project: RO-BUST).

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

# A   Illustrative Example

We illustrate the QBLR algorithm with a low-dimensional example: *negative binomial* distribution family lifted to a (3)-dimensional scale-mixture distribution family, the so-called *Poisson–Gamma* distribution. The exponential hierarchical lifts for non-exponential family distributions like the Negative Binomial distribution are plentiful in the literature (see Section 7), but they are often *curved* (non-minimal) in their form: the classical Poisson–Gamma parameterization in $(r, p)$ ties together two natural coordinates and thereby lives on a 2D submanifold of a 3D joint. In this section, we present an ad-hoc—yet fully constructive—way to *uncurve* that representation by introducing one free scale, yielding a *minimal* 3-parameter exponential-family lift in natural coordinates. This is exactly the setting required by our quotient-manifold theory.

We first show how to build a minimal marginal exponential representation lift for the Negative Binomial distribution from the Poisson–Gamma distribution in Subsection A.1 and then we show how to use the established lift to instantiate Algorithm 1 for this specific scenario in Subsection A.2. Finally, we discuss the limitations of the same uncurving trick for the Laplace case in Appendix A.3.

We thank the anonymous reviewer (bBHv) who proposed this section.

*Note (erratum).* In our rebuttal we incorrectly stated that the Laplace distribution could also be recovered using this construction technique. However, the resulting marginal family is actually richer than the Laplace family alone. Whether the Laplace distribution can be obtained as a minimal lift through a different non-minimal representation, or requires a fundamentally different construction, remains an open question. We apologize for this oversight; see Appendix A.3 for details.

## A.1   Building the minimal marginal lift

The framework of application of QBLR is quite general as many non-exponential family distributions have some joint exponential family representation. However, our work is limited by an even stronger assumption: the existence of a minimal parameterization of these joint exponential family representations. While we assume that at least some lift to a joint exponential family is given, such representations are often curved (non-minimal) in their standard form. Whether or not a lift can be found is then a crucial question for us.

By investigating this point, this section enables us to understand where and why our QBLR Algorithm should be applied.

The heavy-tailed distributions families that have an exponential family scale mixture representation are well documented in the literature. Regarding the particular case of being a scale mixture of normal distributions, Andrews and Mallows [1974] provides necessary and sufficient conditions in their paper which is applied to Student-$t$, Laplace, and Logistic distributions. More examples can be found in Coelho and Chen [2024].

Many scale-mixture joints found in the literature come in a *curved* form—that is, their sufficient statistics are linearly dependent, so the family is *not* minimal. The textbook parameterisations of both the Normal–Wishart and the Normal–Exponential joints fall into this category. (By contrast, the Normal–Wishart lift we use—see Appendix C—is explicitly minimal; the distinction is made concrete in the Laplace example that follows.) *Discrete* over-dispersed families admit analogous lifts; in particular, the Negative–Binomial (NB) arises as a Poisson–Gamma mixture.

Because a curved exponential family violates the minimal-regular assumption, it cannot serve as a lift for QBLR *unless* one first "uncurves" it by adding extra, independent natural parameters. We now explain why this step is necessary and how those additional degrees of freedom restore minimality.

**Curved vs. minimal lift for the Negative–Binomial distribution.** The textbook Poisson–Gamma mixture

$$\mathrm{NB}(k \mid r, p) \;=\; \int_0^\infty \underbrace{\mathrm{Poisson}\big(k \mid \lambda\big)}_{\text{Poisson}} \underbrace{\mathrm{Gamma}\Big(\lambda \;\Big|\; r,\; \beta = \tfrac{p}{1-p}\Big)}_{\text{Gamma (rate)}} \, \mathrm{d}\lambda \qquad \text{(NB-curved)}$$

is *curved* when viewed as a joint EF in $(k, \lambda)$: the joint density has **three** sufficient statistics, $T_1 = \log \lambda$, $T_2 = \lambda$, $T_3 = k$, but only **two** free parameters $(r, p)$ (equivalently, $T_3$'s natural parameter is fixed at zero). Hence the Jacobian of the sufficient-statistics map has rank 2 and Brown's minimality criterion fails [Brown, 1986, Prop. 1.5].

**Uncurving with one extra degree of freedom.** Introduce an independent positive scale $c > 0$ on the Poisson mean:

$$\mathrm{NB}(k \mid r, p) \;=\; \int_0^\infty \underbrace{\mathrm{Poisson}\big(k \mid c\,\lambda\big)}_{\text{scaled Poisson}} \; \underbrace{\mathrm{Gamma}\big(\lambda \mid r,\ \beta > 0\big)}_{\text{Gamma (rate)}} \; d\lambda. \qquad \text{(NB-minimal)}$$

Now the joint admits a *minimal* EF representation with three independent natural parameters

$$\boldsymbol{\eta} = (\eta_1, \eta_2, \eta_3) = \big(r - 1,\ \ -(\beta + c),\ \ \log c\big),$$

sufficient statistics $T = (\log \lambda,\ \lambda,\ k)$, base measure $h(k, \lambda) = \mathbf{1}_{\{\lambda > 0\}} \lambda^k / k!$, and log-partition

$$A(\boldsymbol{\eta}) = \log \Gamma(\eta_1 + 1) - (\eta_1 + 1) \log\big(-\eta_2 - e^{\eta_3}\big), \quad \mathcal{D} = \{\eta_1 > -1,\ \eta_2 + e^{\eta_3} < 0\}.$$

Crucially, integrating out $\lambda$ gives, for *every* $\eta \in \mathcal{D}$, the Negative–Binomial marginal with parameters

$$r = \eta_1 + 1 \; > \; 0, \qquad p = \frac{e^{\eta_3}}{-\eta_2} \in (0, 1), \qquad q_\eta(k) = \binom{k + r - 1}{k}(1 - p)^r p^k.$$

Equivalently, the marginalisation map written in *natural* coordinates is the smooth surjection

$$\pi : \; \mathcal{D} \to \mathbb{R}_{>0} \times (0, 1), \qquad \pi(\eta_1, \eta_2, \eta_3) = \big(r = \eta_1 + 1,\ p = e^{\eta_3}/(-\eta_2)\big),$$

so $\Xi \cong \mathcal{D}/\sim_\pi$ forms a quotient manifold with a one-dimensional fibre and the rank-one projector used in Algorithm 1 (see Appendix A.2).

**Open question.** We do *not* know whether every curved exponential family can be "uncurved" by judiciously adding degrees of freedom; establishing necessary and sufficient conditions remains, to our knowledge, an open problem in exponential-family theory. In particular, applying the same uncurving strategy to the Laplace family via a Normal–Exponential lift restores minimality in the joint but yields a marginal family that is *strictly richer* than Laplace and therefore does not reproduce Laplace *globally* across the natural domain. We thus present our "add-a-free-hyperparameter" trick as an *empirical recipe*, not a theorem; see Subsection A.3 for details.

### A.2 Instantiation of the QBLR

Let $(k, \lambda) \in \{0, 1, 2, \dots\} \times \mathbb{R}_{>0}$ and define the minimal, regular exponential family

$$q_\eta(k, \lambda) \;=\; h(k, \lambda) \exp\{\eta_1 \log \lambda \;+\; \eta_2 \lambda \;+\; \eta_3 k \;-\; A(\eta)\}, \qquad h(k, \lambda) = \frac{\lambda^k}{k!} \mathbf{1}_{\{\lambda > 0\}}. \quad (20)$$

Minimality is immediate: if $a_1 \log \lambda + a_2 \lambda + a_3 k \equiv \text{const}$ on $\{(k, \lambda)\}$, then varying $k$ forces $a_3 = 0$ and varying $\lambda$ forces $a_1 = a_2 = 0$. Summing over $k$ and integrating in $\lambda$ gives the log-partition

$$Z(\boldsymbol{\eta}) = \int_0^\infty \sum_{k=0}^\infty \frac{\lambda^k}{k!} \exp\{\eta_1 \log \lambda + \eta_2 \lambda + \eta_3 k\} \, d\lambda \;=\; \int_0^\infty \lambda^{\eta_1} \exp\{(\eta_2 + e^{\eta_3})\,\lambda\} \, d\lambda$$

$$= \Gamma(\eta_1 + 1)\,[-(\eta_2 + e^{\eta_3})]^{-(\eta_1 + 1)}, \qquad\qquad (21)$$

which converges on the open domain

$$\widetilde{\Lambda}_\eta \;=\; \big\{\, \eta \in \mathbb{R}^3 : \eta_1 > -1,\ \eta_2 + e^{\eta_3} < 0 \,\big\}.$$

Hence

$$A(\boldsymbol{\eta}) \;=\; \log \Gamma(\eta_1 + 1) \;-\; (\eta_1 + 1) \log\big(-\eta_2 - e^{\eta_3}\big). \qquad\qquad (22)$$

**Marginalization map.** For each fixed $k$, integrate out $\lambda$:

$$q_\eta(k) = \int_0^\infty q_\eta(k, \lambda) \, d\lambda = e^{-A(\eta)} \frac{e^{\eta_3 k}}{k!} \int_0^\infty \lambda^{k + \eta_1} e^{\eta_2 \lambda} \, d\lambda$$

$$= \frac{\Gamma(k + \eta_1 + 1)}{\Gamma(\eta_1 + 1)\,k!} \left(\frac{-\eta_2 - e^{\eta_3}}{-\eta_2}\right)^{\eta_1 + 1} \left(\frac{e^{\eta_3}}{-\eta_2}\right)^k. \qquad\qquad (23)$$

Writing

$$r \;=\; \eta_1 + 1 \; > \; 0, \qquad p \;=\; \frac{e^{\eta_3}}{-\eta_2} \; \in \; (0, 1),$$

(where $p \in (0,1)$ follows from $\eta_2 + e^{\eta_3} < 0$), we obtain

$$q_\eta(k) = \binom{k+r-1}{k} (1-p)^r p^k, \qquad k = 0,1,2,\ldots, \tag{24}$$

i.e. the Negative–Binomial $\mathrm{NB}(r,p)$ for *every* $\eta \in \widetilde{\Lambda}_\eta$. Thus the marginalization map in natural coordinates is the smooth surjection

$$\pi : \ \widetilde{\Lambda}_\eta \longrightarrow \Xi := \mathbb{R}_{>0} \times (0,1), \qquad \pi(\eta_1, \eta_2, \eta_3) = \big(r = \eta_1 + 1, \ p = e^{\eta_3}/(-\eta_2)\big). \tag{25}$$

It is visibly surjective: given any $(r,p) \in \Xi$, take $\eta_1 = r - 1$ and, for arbitrary $c > 0$, set $\eta_2 = -c$ and $\eta_3 = \log(pc)$—then $\eta \in \widetilde{\Lambda}_\eta$ and $\pi(\eta) = (r,p)$.

**Fibres and rank-one projector in natural coordinates.** The Jacobian of (25) is

$$D\pi(\eta) = \begin{pmatrix} 1 & 0 & 0 \\ 0 & \dfrac{e^{\eta_3}}{\eta_2^2} & \dfrac{e^{\eta_3}}{-\eta_2} \end{pmatrix},$$

so $\ker D\pi(\eta) = \mathrm{span}\{\, k_\eta(\eta) \,\}$ with the *vertical* vector

$$k_\eta(\eta) = \big(0,\ \eta_2,\ 1\big)^\top, \qquad D\pi(\eta)\, k_\eta(\eta) = 0. \tag{26}$$

Hence each fibre is the smooth 1D curve

$$\mathcal{F}_\eta = \big\{ (\eta_1, e^t \eta_2, \eta_3 + t) : t \in \mathbb{R} \big\},$$

which leaves $(r,p)$ invariant because $r = \eta_1 + 1$ and $p = e^{\eta_3}/(-\eta_2)$. Equipping the lift with its Fisher metric $F(\eta) = \nabla^2 A(\eta)$, the horizontal projector is rank–one:

$$P_{\mathcal{H}}(\eta)\, g = g - \frac{k_\eta(\eta)^\top F(\eta)\, g}{k_\eta(\eta)^\top F(\eta)\, k_\eta(\eta)}\, k_\eta(\eta), \tag{27}$$

matching the quotient geometry used throughout (Theorems 1 and 2).

Figure 2 illustrates the practical benefit of using the QBLR algorithm. Panel (a) shows the Euclidean gradient field $-\nabla_{(r,p)}\mathrm{KL}(q_{r,p}\|q_{\text{true}})$ computed via finite differences in the marginal coordinates $(r,p)$; these directions ignore the Fisher–Rao geometry and can yield poorly conditioned trajectories. Panel (b) displays the quotient natural gradient: at each $(r,p)$ we lift to an arbitrary representative $\eta \in \pi^{-1}(r,p)$, compute the natural gradient $\nabla_\eta A(\eta)(\eta - \eta_{\text{true}})$ in the 3-parameter Poisson–Gamma space, project it horizontally, and push it forward through $D\pi(\eta)$. The resulting arrows respect the quotient manifold structure and are invariant to the choice of lift within each fibre, thereby guaranteeing parameterization-free optimization on the Negative–Binomial manifold itself.

### A.3 When does the uncurving trick fails?

The Negative–Binomial example showed that introducing a free scale parameter can uncurve a Poisson–Gamma mixture and enable QBLR. Does the same strategy work for the Laplace distribution's Normal–Exponential representation? As we demonstrate below, adding a variance-scaling parameter $\kappa > 0$ does restore minimality, but the resulting marginal family is *strictly richer* than the standard two-parameter Laplace: the uncurved lift introduces degrees of freedom that survive marginalization. This cautionary example illustrates that uncurving is an empirical recipe whose validity must be verified case by case, not a universal construction.

Let[2]

$$\lambda = (\lambda_1, \lambda_2, \lambda_3) \in \Lambda := \big\{ (\lambda_1, \lambda_2, \lambda_3) \in \mathbb{R}^3 : \lambda_3 < 0, \ \lambda_2 + \tfrac{1}{2}\lambda_1^2 < 0 \big\}$$

parameterize the (minimal) Normal–Exponential *variance–mixture* lift with latent variance $\tau \in \mathbb{R}_{>0}$:

$$q_\lambda(z, \tau) = \frac{\exp\left(-\dfrac{z^2}{2\tau}\right)}{\sqrt{2\pi\,\tau}} \exp\Big\{\lambda_1 \frac{z}{\tau} + \lambda_2 \frac{1}{\tau} + \lambda_3 \tau - A(\boldsymbol{\lambda})\Big\}, \qquad \tau > 0. \tag{28}$$

---

[2]*Erratum:* In the rebuttal response we mistakenly wrote the joint without the Gaussian base factor $(2\pi\tau)^{-1/2}\exp\{-z^2/(2\tau)\}$, which makes the $z$-integral non-normalizable. The corrected minimal EF is given by Eqs. (28)–(29).

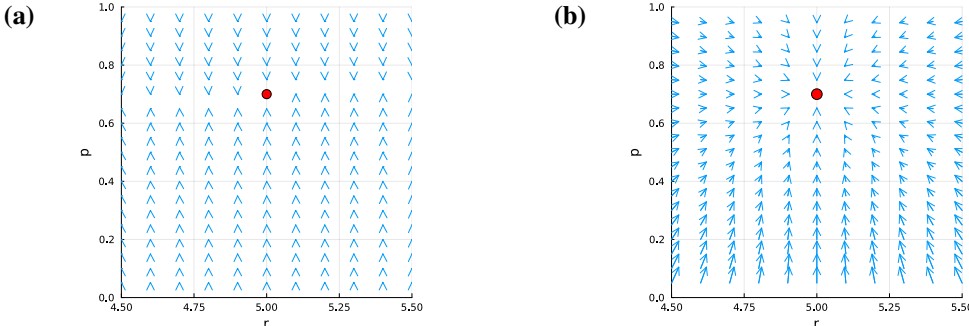

**Figure 2:** Comparison of gradient flows on the Negative Binomial parameter space $(r, p)$ toward the true distribution (red dot at $r = 5.0, p = 0.7$). **(a)** Euclidean gradient field: arrows show the naive gradient $-\nabla_{(r,p)}\text{KL}(q_{r,p}\|q_{\text{true}})$ computed via finite differences. This approach ignores the underlying statistical geometry and can lead to inefficient trajectories. **(b)** Quotient natural gradient field: arrows represent the horizontally-projected natural gradient obtained by lifting to the Poisson–Gamma representation, computing the natural gradient in the 3-parameter exponential family, and projecting onto the horizontal space. The quotient natural gradient respects the Fisher–Rao geometry of the marginal Negative Binomial manifold, yielding parameterization-invariant descent directions that follow geodesics in the information-geometric sense. Both fields converge to the same optimum, but the quotient approach provides more stable and geometry-aware updates.

The log-partition function is

$$A(\boldsymbol{\lambda}) = \log\left(2\sqrt{\tfrac{\gamma}{\beta}}\, K_1\big(2\sqrt{\beta\gamma}\big)\right), \qquad \beta := -\lambda_3 > 0, \;\; \gamma := -\big(\lambda_2 + \tfrac{1}{2}\lambda_1^2\big) > 0, \qquad (29)$$

with the continuous boundary extension $A(\boldsymbol{\lambda}) = -\log\beta$ at $\gamma = 0$. Here $K_1$ denotes the modified Bessel function of the second kind [noa, (10.32.10)] defined by the integral representation

$$K_1(z) = \frac{z}{4}\int_0^\infty \exp\left(-t - \frac{z^2}{4t}\right)\frac{dt}{t^2}, \qquad z > 0. \qquad (30)$$

Note that the form of $A(\boldsymbol{\lambda})$ trivially follows from the integral identity [Moll, 2015, 7.2.12].

Equation (28) is a *minimal, regular exponential family*: the sufficient statistics $\big(z/\tau,\; 1/\tau,\; \tau\big)$ are linearly independent, and $\Lambda$ is an open subset of $\mathbb{R}^3$.

**Why the marginal is not Laplace.** To see why the uncurved lift fails, we compute the $z$-marginal by integrating out $\tau$ from (28). Completing the square in the exponent gives $z^2 - 2\lambda_1 z - 2\lambda_2 = (z - \lambda_1)^2 - 2\gamma$, where $\gamma := -(\lambda_2 + \tfrac{1}{2}\lambda_1^2) > 0$. The Bessel integral then yields

$$q_\lambda(z) \propto \frac{1}{\sqrt{(z - \lambda_1)^2 + 2\gamma}}\,\exp\left\{-\sqrt{-2\lambda_3}\,\sqrt{(z - \lambda_1)^2 + 2\gamma}\right\}. \qquad (31)$$

This is *not* a Laplace distribution. The standard two-parameter Laplace has the form

$$\text{Lap}(z \mid \mu, b) = \frac{1}{2b}\exp\big(-|z - \mu|/b\big) = \frac{1}{2b}\exp\big(-\sqrt{(z - \mu)^2}/b\big),$$

involving only $\sqrt{(z - \mu)^2}$. By contrast, (31) includes an additional constant $2\gamma$ under the square root. This extra degree of freedom survives marginalization, producing a *three-parameter* family strictly richer than Laplace.

The uncurved Normal–Exponential lift does define a valid quotient manifold via Theorems 1–2, but the marginal family overshoots the target: we obtain a generalized symmetric distribution rather than Laplace. The standard Laplace embeds as a constrained two-dimensional submanifold (requiring $\gamma = \text{const}$) within this richer structure. Consequently, while the geometry is correct, the probabilistic target is wrong—illustrating that uncurving is a case-by-case recipe, not a universal construction.

# B  Marginal quotient manifold theory

This appendix gathers the differential-geometric material needed for Sections 2–4 of the main text. We first recall just enough quotient-manifold theory to set notation (Subsection B.1), derive the horizontal projector used in Algorithm 1, and then give the proofs of Theorems 1–2 (Sections B.3–B.4).

**Prerequisite.** The exposition assumes familiarity with embedded submanifolds and the basic vocabulary of Riemannian geometry. Readers new to this topic may find Boumal [2023, Chapter 3] a concise primer before diving in.

## B.1 Quotient manifold theory

A quotient manifold arises when we identify points in a manifold $M$ according to an equivalence relation $\sim$. However, not every equivalence relation on $M$ defines a quotient manifold. The conditions under which an equivalence relation yields a quotient manifold structure have been studied extensively in differential geometry [Absil et al., 2008][Section 3.4 Quotient manifolds].

Formally, the quotient space $M/\sim$ consists of equivalence classes $[x] = \{y \in M : y \sim x\}$. The canonical projection $\pi : M \to M/\sim$ sends each point to its class, $\pi(x) = [x]$. The *fibre*[3] through $x$—the pre-image of that class—is defined as

$$F_x := \pi^{-1}([x]) := \{y \in M : \pi(y) = \pi(x)\} = \{y \in M : y \sim x\}. \tag{32}$$

Throughout we work with an embedded submanifold $M \subset \mathbb{R}^n$ and a projection $\pi : M \to N$ whose image $N \subset \mathbb{R}^m$ is itself embedded. In this context, the general quotient manifold criterion reduces to a simple test:

$$\pi \text{ is smooth} \quad \text{and} \quad \operatorname{rank} D\pi(x) = \dim N \ \forall x \in M. \tag{$*$}$$

If condition $(*)$ holds, then $\pi$ is called a *smooth submersion*; every fibre $\pi^{-1}([x])$ is an embedded submanifold, and the quotient inherits a unique $d$-dimensional smooth structure. Consequently, we can define

$$\dim(M/\sim) := \dim N,$$

secure in the knowledge that this integer is well-defined by the constant–rank condition.

Under condition $(*)$, each fibre $\pi^{-1}([x])$ is an embedded submanifold and there is a *unique* smooth structure on $M/\sim$ that makes $\pi$ a smooth submersion into $M/\sim$ [Absil et al., 2008, Prop. 3.4.2]. Moreover, the quotient $M/\sim$ is automatically Hausdorff and second–countable.

**Vertical space.** The tangent space of $F_x$ is the kernel of the differential

$$T_x F_x = \ker D\pi(x) \subseteq T_x M. \tag{33}$$

We call this subspace the *vertical space* and write $\mathcal{V}_x := \ker D\pi(x)$.

**Horizontal space.** Let $\langle \cdot, \cdot \rangle_x$ be a Riemannian metric on $M$. The orthogonal complement of $\mathcal{V}_x$ is the *horizontal space*

$$\mathcal{H}_x := \{v \in T_x M : \langle v, w \rangle_x = 0 \ \forall w \in \mathcal{V}_x\}. \tag{34}$$

A key property of quotient manifolds is that a Riemannian metric on $M$ induces a unique metric on $M/\sim$ if it is invariant along fibers. Specifically, if for any $x \sim y$ and any horizontal vectors $u \in \mathcal{H}_x$ and $v \in \mathcal{H}_y$ with $D\pi(x)[u] = D\pi(y)[v]$, we have $\langle u, u \rangle_x = \langle v, v \rangle_y$, then we can define a well-posed metric on the quotient

$$\langle \xi, \zeta \rangle_{[x]} = \langle \hat{\xi}, \hat{\zeta} \rangle_x, \tag{35}$$

where $\hat{\xi}$ and $\hat{\zeta}$ are the horizontal lifts of tangent vectors $\xi, \zeta \in T_{[x]}(M/\sim)$. This makes $M/\sim$ a Riemannian quotient manifold.

Readers seeking a more concrete treatment of these abstract concepts may refer to Appendix B.5, where we examine them in the context of the Normal-Gamma distribution.

---

[3] Throughout, we treat the equivalence class $[x]$ as a *point* of the quotient manifold $M/\sim$; the fibre is the full pre-image of that point. We write $T_{[x]}(M/\sim)$ (with parentheses) only to emphasise that the tangent is taken *after* the quotient, not the quotient of the tangent space at point $x$. The shorter $T_{[x]}$ or $T_{[x]}M/\sim$ can be used whenever no confusion arises.

## B.2 Orthogonal projection onto the horizontal space

At every point $\boldsymbol{\lambda} \in \Lambda$ the tangent space splits as $T_\lambda \Lambda = \mathcal{H}_\lambda \oplus \mathcal{V}_\lambda$, where $\mathcal{V}_\lambda := \ker D\pi(\lambda)$ is the *vertical* subspace (directions that leave the marginal unchanged) and $\mathcal{H}_\lambda$ is its $F(\lambda)$-orthogonal complement (horizontal directions that do change the marginal). For gradient–based optimisation we need a fast way to remove the vertical component of an arbitrary vector $g \in T_\lambda \Lambda$.

To do so, let $K(\boldsymbol{\lambda}) \in \mathbb{R}^{d \times r}$ be any matrix whose columns span $\mathcal{V}_\lambda$ (where $\dim V_\lambda = r$). Write the desired horizontal part as $g_\lambda^\perp = g - K\alpha$ for some coefficient vector $\alpha \in \mathbb{R}^r$. Imposing $F(\boldsymbol{\lambda})$-orthogonality to *every* vertical vector $Kv$ gives the normal equations

$$K^\top F(\boldsymbol{\lambda})\big(g - K\alpha\big) = 0 \implies \alpha = \big[K^\top F(\boldsymbol{\lambda})K\big]^{-1} K^\top F(\boldsymbol{\lambda})g.$$

Substituting this $\alpha$ yields the explicit projector

$$P_{\mathcal{H}}(\boldsymbol{\lambda}) = I - K(\boldsymbol{\lambda})\big[K(\boldsymbol{\lambda})^\top F(\boldsymbol{\lambda})K(\boldsymbol{\lambda})\big]^{-1} K(\boldsymbol{\lambda})^\top F(\boldsymbol{\lambda}), \tag{36}$$

so that $g_\lambda^\perp = P_{\mathcal{H}}(\boldsymbol{\lambda})\, g$.

The matrix to be inverted is only $r \times r$ with $r = \dim \mathcal{V}_\lambda$. In our Normal–Wishart example $r = 1$; (36) then reduces to a single scalar division, completely sidestepping the $\mathcal{O}(d^3)$ cost of inverting the full Fisher matrix.

## B.3 Proof of Theorem 1

This section is devoted to the proof of Theorem 1. Before proceeding with the proof, we recall the notation established in the main text.

**Theorem 3** (Induced Fisher–Rao metric). *Assume the setting of Theorem 1 and equip the natural-parameter space $\Lambda$ with its Fisher information metric $F_\lambda$. Then:*

- *(i) The map $\pi$, that project the Riemannian manifold $(\Lambda, F_\lambda)$ on $\Xi$, induces a Riemannian quotient manifold structure on $\Xi$;*
- *(ii) The Riemannian quotient metric on $\Xi$ is then the Fisher metric of $\Xi$.*

**Setting and notation.** In this paragraph, we restate the symbols used in the main text, all in one place. We work on a measurable product space $\mathcal{Z}_{\text{ext}} = \mathcal{Z}_U \times \mathcal{Z}_V$ and write $z_{\text{ext}} = (z_U, z_V)$ for a generic element. The block $z_U$ collects the coordinates whose distribution we ultimately care about, whereas $z_V$ will be integrated out. Therefore, to shorten our notation, we refer to $\mathcal{Z}_U$ and $z_U$ as $\mathcal{Z}$ and $z$, respectively.

The marginal family that defines a distribution over $\mathcal{Z}$ is parametrized by $\Xi$. Its ambient "parent" is a minimal, regular exponential family with open natural–parameter space $\Lambda$ that defines a distribution over $\mathcal{Z}_{\text{ext}}$.

The key connection between $\Xi$ and $\Lambda$ is a marginal relation

$$q_\xi(\mathbf{z}) = \int q_\lambda(\mathbf{z}, \mathbf{z}_V)\, \mathrm{d}\mathbf{z}_V. \tag{37}$$

This relation naturally defines a function $\pi : \Lambda \to \Xi.$. And the function $\pi$ naturally defines the corresponding equivalence relation $\sim_\pi$ on $\Lambda$ in the following way:

$$\boldsymbol{\lambda}_1 \sim \boldsymbol{\lambda}_2 \Leftrightarrow \pi(\boldsymbol{\lambda}_1) = \pi(\boldsymbol{\lambda}_2). \tag{38}$$

That is, two points in $\Lambda$ are equivalent if they yield the same marginal distribution.

Note that, for a minimal regular exponential family, the log-partition function $A(\lambda)$ is infinitely differentiable, and its derivatives correspond to the moments of the sufficient statistics. The marginalization can be expressed in terms of these moments, which inherit the smoothness properties of $A(\lambda)$.

We remind the reader that in our setting to prove Theorem 1 it suffices to show that $\pi$ is a *smooth submersion* (see the condition $(*)$). For convenience, we quote the theorem we are about to prove in the notation fixed above.

**Theorem 1 (Marginalization yields a smooth quotient manifold).** *Let $q_\lambda$ be a minimal, regular exponential family with parameter space $\Lambda \subset \mathbb{R}^d$. Suppose a partition $\mathcal{Z}_{ext} = (\mathcal{Z}, \mathcal{Z}_V)$ is chosen so that the marginal family $\{q_\xi\}_{\xi \in \Xi}$ obtained via $\pi : \Lambda \to \Xi$ is moment-parametrized (with $\dim \Xi = k$) (Definition 1). Then $\Xi$ is the quotient manifold of $\Lambda$ induced by $\pi$.*

*Proof.* Recall the marginal relation

$$q_\xi(\mathbf{z}) = \int_{\mathcal{Z}_V} q_\lambda(\mathbf{z}_U, \mathbf{z}_V)\, \mathrm{d}\mathbf{z}_V, \quad \mathbf{z}_U \in Z_U.$$

**Step 0. Set-up and notation.** Write $T = (T_1, \dots, T_d)$ and $\boldsymbol{\lambda} = (\lambda^1, \dots, \lambda^d)$. For any bounded measurable $\varphi : \mathcal{Z}_U \to \mathbb{R}$ set

$$\langle \varphi \,;\, q_\xi \rangle := \int_{\mathcal{Z}_U} \varphi(\mathbf{z}_U) q_\xi(\mathbf{z}_U)\, \mathrm{d}\mathbf{z}_U.$$

**Step 1. A finite-dimensional probe of the marginals.** Because the marginal family is moment-parameterized with $\dim \Xi = k$, it is attached to $k$ integrable functions $m_1, \dots, m_k \in L^\infty(\mathcal{Z}_U)$:

$$e : \Lambda \longrightarrow \mathbb{R}^k, \qquad e^i(\lambda) := \langle m_i \,;\, q_{\pi(\lambda)} \rangle \quad (i = 1, \dots, k).$$

Writing the marginal as a single integral gives the equivalent form

$$e^i(\lambda) = \int_{\mathcal{Z}_{ext}} m_i(\mathbf{z}_U) q_\lambda(\mathbf{z}_{ext})\, \mathrm{d}\mathbf{z}_{ext}.$$

The goal is to show that $e$ is a smooth submersion of constant rank $r = \operatorname{rank} D\pi(\lambda)$ (the same $r$ for every $\lambda$). Once established, each fibre $e^{-1}(y)$ is automatically an embedded submanifold of $\Lambda$.

**Step 2. Computing the derivative of $e$.** Fix $\lambda \in \Lambda$ and a tangent vector $v = (v^1, \dots, v^d) \in T_\lambda \Lambda$. Differentiate under the integral (dominated convergence allows this):

$$D_v e^i(\lambda) = \sum_{j=1}^d v^j \frac{\partial}{\partial \lambda^j} \int_{\mathcal{Z}_{ext}} \varphi_i(\mathbf{z}_U) q_\lambda(\mathbf{z}_{ext})\, \mathrm{d}\mathbf{z}_{ext}$$

$$= \sum_{j=1}^d v^j \int_{\mathcal{Z}_{ext}} \varphi_i(\mathbf{z}_U) \frac{\partial}{\partial \lambda^j} q_\lambda(\mathbf{z}_{ext})\, \mathrm{d}\mathbf{z}_{ext}.$$

Because $\frac{\partial}{\partial \lambda^j} q_\lambda(x) = \left( T_j(\mathbf{z}_{ext}) - \frac{\partial A(\boldsymbol{\lambda})}{\partial \lambda^j} \right) q_\lambda(x)$, we obtain the *exact* Jacobian entry

$$J_{ij}(\boldsymbol{\lambda}) := \frac{\partial e^i}{\partial \lambda^j}(\boldsymbol{\lambda}) \tag{39}$$

$$= \int_{\mathcal{Z}_{ext}} m_i(\mathbf{z}_U) \left( T_j(\mathbf{z}_{ext}) - \underbrace{\frac{\partial A(\boldsymbol{\lambda})}{\partial \lambda^j}}_{= \mathbb{E}_{q_\lambda}[T_j]} \right) q_\lambda(\mathbf{z}_{ext})\, \mathrm{d}\mathbf{z}_{ext}. \tag{40}$$

Note, that from (39) smoothness trivially follows from the fact that $A(\boldsymbol{\lambda}) \in C^\infty(\Lambda)$.

A convenient way to rewrite equation (40) is with covariances (up to additive constant which does not change the rank):

$$J_{ij}(\boldsymbol{\lambda}) = \operatorname{Cov}_{q_\lambda}[m_i(\mathbf{z}_U), T_j(\mathbf{z}_{ext})]. \tag{41}$$

So each column $j$ of $J(\boldsymbol{\lambda})$ stores the $k$ covariances between the function $m_i$ and the statistic $T_j$ under the *joint* distribution $q_\lambda$.

**Step 3. Why the rank is constant.** Minimality of our exponential family guarantees that the $d \times d$ covariance matrix $F(\lambda) := \operatorname{Cov}_{q_\lambda}[T_i(\mathbf{z}_{ext}), T_j(\mathbf{z}_{ext})]$ is *positive definite* for every $\lambda$ [Brown, 1986, Theorem 4.1]. Denote by

$$H(\boldsymbol{\lambda}) := J(\boldsymbol{\lambda}) F(\boldsymbol{\lambda})^{-1} J(\boldsymbol{\lambda})^\top \in \mathbb{R}^{k \times k}.$$

Because $F_{\boldsymbol{\lambda}} \succ 0$, $H(\boldsymbol{\lambda})$ is *positive semidefinite* for every $\boldsymbol{\lambda}$ and satisfies

$$\det H(\boldsymbol{\lambda}) = 0 \iff \operatorname{rank} J(\boldsymbol{\lambda}) < k.$$

Suppose that $\det H(\boldsymbol{\lambda}_0) > 0$ at some point $\boldsymbol{\lambda}_0$. Then the matrix $H(\boldsymbol{\lambda}_0) = J(\boldsymbol{\lambda}_0)F(\boldsymbol{\lambda}_0)^{-1}J(\boldsymbol{\lambda}_0)^{\top}$ is positive definite, and in particular, the Jacobian matrix $J(\boldsymbol{\lambda}_0)$ has full rank $k$. This means the map $\boldsymbol{\lambda} \mapsto e(\boldsymbol{\lambda})$ has full rank at $\boldsymbol{\lambda}_0$. Since both $F(\boldsymbol{\lambda})$ and $J(\boldsymbol{\lambda})$ are real-analytic functions of $\boldsymbol{\lambda}$, the composition $H(\boldsymbol{\lambda}) = J(\boldsymbol{\lambda})F(\boldsymbol{\lambda})^{-1}J(\boldsymbol{\lambda})^{\top}$ is also real-analytic. Consequently, $\det H(\boldsymbol{\lambda})$ is a real-analytic scalar function on the parameter space. By a basic property of real-analytic functions, if $\det H(\boldsymbol{\lambda})$ is not identically zero, then its zero set has empty interior. Since $\det H(\boldsymbol{\lambda}_0) > 0$, the function cannot be identically zero, and hence there exists an open neighborhood of $\boldsymbol{\lambda}_0$ where $\det H(\boldsymbol{\lambda}) > 0$. Thus, $\operatorname{rank}(J(\boldsymbol{\lambda})) = k$ in a neighborhood of $\boldsymbol{\lambda}_0$. Therefore, the rank of $J(\boldsymbol{\lambda})$ cannot drop in any open neighborhood where $\det H(\boldsymbol{\lambda})$ is positive. If rank were to drop at some point, this would force $\det H(\boldsymbol{\lambda}) = 0$ at that point, contradicting the real-analyticity and strict positivity nearby. Hence, the rank remains full wherever it is full once.

**Step 4. Submersion $\Rightarrow$ embedded fibres.** We now consider the smooth map $e : \Lambda \to \mathbb{R}^k$, which we have shown to have constant rank $k$. By the finite-dimensional *constant-rank theorem* [Lee, 2012][Theorem 5.12], it follows that each fibre $e^{-1}(y)$ is an embedded submanifold of $\Lambda$ of codimension $k$ (i.e., of dimension $d - k$) and these fibres vary smoothly with $y$, forming a *regular foliation* of $\Lambda$. Moreover, since $e(\boldsymbol{\lambda})$ depends on $\boldsymbol{\lambda}$ only through the marginal distribution $q_\xi$, we have:

$$e^{-1}(e(\boldsymbol{\lambda})) = \{\boldsymbol{\lambda}' \in \Lambda : q_{\pi(\lambda')} = q_{\pi(\lambda)}\} = \pi^{-1}(q_{\pi(\lambda)}),$$

where $\pi : \Lambda \to Q_\Xi$ denotes the map sending $\boldsymbol{\lambda}$ to its marginal distribution $q_{\pi(\lambda)}$. Therefore, each marginal pre-image is an embedded submanifold of $\Lambda$, and the space of parameters decomposes smoothly according to level sets of the marginal.

$\square$

## B.4  Proof of Theorem 2

This section is devoted to the proof of Theorem 2. Before proceeding with the proof, we recall the statement of the theorem from the main text. We use the notation established in the previous section.

**Theorem 2 (Induced Fisher–Rao metric).** *Assume the setting of Theorem 1 and equip the natural-parameter space $\Lambda$ with its Fisher information metric $F_\lambda$. Then:*

  *(i) The map $\pi$, that project the Riemannian manifold $(\Lambda, F_\lambda)$ on $\Xi$, induces a Riemannian quotient manifold structure on $\Xi$;*

  *(ii) The Riemannian quotient metric on $\Xi$ is then the Fisher metric of $\Xi$.*

*Proof.* Consider $\xi \in \Xi$ and $\boldsymbol{\lambda} \in \pi^{-1}(\xi)$, then let $f(\mathbf{z}, \xi)$ denote the log-density of the distribution $q_\xi(\mathbf{z})$, and let $\widetilde{f}(\mathbf{z}_{\text{ext}}, \boldsymbol{\lambda})$ be the log-density of the distribution $q_\lambda(\mathbf{z}_{\text{ext}})$.

Because $\pi : \Lambda \to \Xi$ is a smooth submersion of constant rank (proved in Theorem 1), the Local Section Theorem [Lee, 2012, Theorem 4.26] guarantees that for every $\xi \in \Xi$ and every $\lambda \in \pi^{-1}(\xi)$ there exists an open neighbourhood $U \subset \Xi$ of $\xi$ and a smooth map $\sigma : U \to \Lambda$ such that $\pi \circ \sigma = \operatorname{id}_U$ and $\sigma(\xi) = \boldsymbol{\lambda}$. Patching these local sections with a smooth partition of unity yields a smooth global section $\sigma : \Xi \to \Lambda$ satisfying $\pi \circ \sigma = \operatorname{id}_\Xi$.

The log-density of the distribution $q_\xi(\mathbf{z})$ can be related to $f(\mathbf{z}_{\text{ext}}, \lambda)$ in the following way:

$$f(\mathbf{z}, \xi) = \log \int_{\mathcal{Z}_V} \exp\bigl(\widetilde{f}(\mathbf{z}_{\text{ext}}, \sigma(\xi))\bigr) \, \mathrm{d}\mathbf{z}_V. \tag{42}$$

Consider the following helpful function

$$\Phi(\mathbf{z}, \boldsymbol{\lambda}) := \int_{\mathcal{Z}_V} \exp\bigl(\widetilde{f}(\mathbf{z}, \mathbf{z}_V, \boldsymbol{\lambda})\bigr) \, \mathrm{d}\mathbf{z}_V. \tag{43}$$

Then we can express $f(\mathbf{z}, \xi)$ in the following way:

$$f(\mathbf{z}, \xi) = \log \Phi(\mathbf{z}, \sigma(\xi)). \tag{44}$$

Now, we differentiate both sides of the identity (44) with respect to $\xi$ and we get the following:

$$\partial_{\xi^i} f(\mathbf{z}, \xi) = \frac{1}{q_\xi(\mathbf{z})} \int_{\mathcal{Z}_V} \exp\left(\widetilde{f}\right) \partial_\xi \widetilde{f}(\mathbf{z}, \mathbf{z}_V, \sigma(\xi)) \mathrm{d}\mathbf{z}_V \qquad \text{(Leibniz + dominated convergence)}$$

$$= \frac{1}{q_\xi(\mathbf{z})} \int_{\mathcal{Z}_V} \exp\left(\widetilde{f}\right) \sum_{j=1}^{d} \partial_{\lambda^j} \widetilde{f}(\mathbf{z}, \mathbf{z}_V, \boldsymbol{\lambda}) \partial_{\xi^i} \sigma^j(\xi) \mathrm{d}\mathbf{z}_V \qquad \text{(chain rule)}$$

$$= \mathbb{E}_{z_V | z}\left[ \sum_{j=1}^{d} \partial_{\lambda^j} \widetilde{f}(\mathbf{z}, \mathbf{z}_V, \boldsymbol{\lambda}) \partial_{\xi^i} \sigma^j(\xi) \right] \qquad \text{(recognize conditional density)}.$$

The last identity can be re-written in a vector form in the following way:

$$\nabla_\xi f(\mathbf{z}, \xi) = D\sigma(\xi)^\top \mathbb{E}_{z_v | z}\left[ \nabla_\lambda \widetilde{f}(\mathbf{z}, \mathbf{z}_v, \sigma(\xi)) \right]. \tag{46}$$

Introduce the *conditional joint score*

$$s(\mathbf{z}, \mathbf{z}_V, \boldsymbol{\lambda}) := \nabla_\lambda \widetilde{f}(\mathbf{z}, \mathbf{z}_V, \boldsymbol{\lambda}), \qquad g(\mathbf{z}, \boldsymbol{\lambda}) := \mathbb{E}_{q_\lambda}[s \mid \mathbf{z}].$$

With this notation (46) reads

$$\nabla_\xi f(\mathbf{z}, \xi) = D\sigma(\xi)^\top g(\mathbf{z}, \sigma(\xi)).$$

Taking the outer product and integrating over $z \sim q_\xi$,

$$F_\Xi(\xi) := \mathbb{E}_z\left[ \nabla_\xi f \, \nabla_\xi f^\top \right] = D\sigma(\xi)^\top \underbrace{\mathbb{E}_z\left[ g \, g^\top \right]}_{=:M(\lambda)} D\sigma(\xi), \quad \lambda = \sigma(\xi). \tag{47}$$

Write $F(\boldsymbol{\lambda}) = \mathbb{E}[ss^\top]$ for the Fisher matrix in $\Lambda$ and $C(\boldsymbol{\lambda}) = \mathbb{E}_z[\mathrm{Var}[s \mid z]]$ for the average conditional covariance. Then using the total variance decomposition, we obtain the following:

$$F(\boldsymbol{\lambda}) = C(\boldsymbol{\lambda}) + M(\boldsymbol{\lambda}) \implies M(\boldsymbol{\lambda}) = F(\boldsymbol{\lambda}) - C(\boldsymbol{\lambda}). \tag{48}$$

Let

$$R(\boldsymbol{\lambda}) := J(\boldsymbol{\lambda})^\top \left[ J(\boldsymbol{\lambda}) J(\boldsymbol{\lambda})^\top \right]^{-1}, \qquad J(\boldsymbol{\lambda}) := D\pi(\boldsymbol{\lambda}). \tag{49}$$

Because every residual score $s_\perp := s - \mathbb{E}[s \mid z]$ satisfies $J s_\perp = 0$, the matrix $C(\boldsymbol{\lambda})$ acts entirely in the vertical space $\ker J$; consequently

$$R^\top C(\lambda) R = 0. \tag{50}$$

Inserting the facts(48)-(50) into the pullback formula (47), we obtain the following:

$$F_\Xi(\xi) = R(\boldsymbol{\lambda})^\top F(\boldsymbol{\lambda}) R(\boldsymbol{\lambda}), \quad J(\boldsymbol{\lambda}) F(\boldsymbol{\lambda}) J(\boldsymbol{\lambda})^\top = F_\Xi(\xi). \tag{51}$$

Hence the Fisher information of the marginal family $\{q_\xi\}$ is obtained from the full Fisher on $\Lambda$ simply by pushing it forward—equivalently, pulling it back—through the Jacobian $J = D\pi(\lambda)$. This metric compatibility (51) fulfills exactly the hypotheses of the Riemannian-quotient theorem, so all conditions of [Boumal, 2023, Theorem 9.35] are satisfied: $\Lambda/\sim_\pi$ inherits the *unique* Riemannian metric that turns $\pi$ into a Riemannian submersion, that is compatible with relation (51).

$\square$

## B.5 Univariate Student-$t$ as a quotient manifold of Normal-Gamma

This section is dedicated to a concrete example of a Riemannian quotient manifold theory applied to marginalization. Even if the formal derivation of the mathematical objects introduced in Sections B.1 and B.2 is not required to implement our main result: Algorithm 1 ; it offers intuition on how a quotient manifold and QBLR work.

A Normal-Gamma distribution is a joint distribution $q_{(\mu,\sigma^{-1},\alpha,\beta)}(x,\tau)$ over a random variable $(x,\tau)$ defined by the following relationship:

$$x|\tau \sim \mathcal{N}(\mu, (\sigma^{-1}\tau)^{-1}), \tag{52a}$$

$$\tau \sim \text{Gamma}(\alpha, \beta). \tag{52b}$$

It is straightforward to rewrite $q_{(\mu,\sigma^{-1},\alpha,\beta)}(x,\tau)$ into the minimal exponential family representation

$$q_\lambda(x,\tau) = \frac{1}{\sqrt{2\pi}} \exp\left(T(x,\tau)^\top \boldsymbol{\lambda} - A(\boldsymbol{\lambda})\right), \tag{53}$$

where the natural parameters, sufficient statistics, and the logpartition are:

$$\boldsymbol{\lambda} = \left(\sigma^{-1}\mu, -\frac{\sigma^{-1}}{2}, \alpha - \frac{1}{2}, -\beta - \frac{\sigma^{-1}\mu^2}{2}\right) \in \Lambda = \mathbb{R} \times \mathbb{R}_- \times \left(\mathbb{R}_+ - \frac{1}{2}\right) \times \mathbb{R}_-, \tag{54a}$$

$$T(x,\tau) = (x\tau, x^2\tau, \log\tau, \tau), \tag{54b}$$

$$A(\boldsymbol{\lambda}) = \log\Gamma\left(\lambda_3 + \frac{1}{2}\right) - \frac{1}{2}\log(-2\lambda_2) - \left(\lambda_3 + \frac{1}{2}\right)\log\left(-\lambda_4 + \frac{\lambda_1^2}{4\lambda_2}\right). \tag{54c}$$

The marginalization over $\tau$ defines a mapping from the Normal-Gamma parameter space $\boldsymbol{\lambda} = (\lambda_1, \lambda_2, \lambda_3, \lambda_4)$ to the Student-$t$ parameter space $\xi = (\mu, \sigma^2, \nu)$ via the following marginalization (quotient) map:

$$\pi(\boldsymbol{\lambda}) = \left(\frac{\lambda_1}{-2\lambda_2}, \frac{-\lambda_4 + \lambda_1^2/(4\lambda_2)}{-2\lambda_2(\lambda_3 + 1/2)}, 2\lambda_3 + 1\right). \tag{55}$$

Our construction starts with $\mathcal{V}_\lambda = \ker D\pi(\boldsymbol{\lambda})$ the vertical space; to obtain it, we need to compute the differential of our quotient map. Then we are left to compute $\mathcal{H}_\lambda = (\mathcal{V}_\lambda)^\perp$.

**Differential of the quotient map.** The Jacobian $J(\boldsymbol{\lambda}) = D\pi(\boldsymbol{\lambda}) \in \mathbb{R}^{3\times4}$ is

$$J(\boldsymbol{\lambda}) = \begin{pmatrix} -\frac{1}{2\lambda_2} & \frac{\lambda_1}{2\lambda_2^2} & 0 & 0 \\ -\frac{\lambda_1}{4\lambda_2^2(\lambda_3+1/2)} & \frac{\lambda_1^2-4\lambda_2\lambda_4}{8\lambda_2^3(\lambda_3+1/2)} & -\frac{\lambda_1^2-4\lambda_2\lambda_4}{2\lambda_2(2\lambda_3+1)^2} & \frac{1}{2\lambda_2(\lambda_3+1/2)} \\ 0 & 0 & 2 & 0 \end{pmatrix}. \tag{56}$$

**Vertical space.** With the Jacobian of the quotient map in natural coordinates (Eq. (56)), the vertical space at $\boldsymbol{\lambda}$ is simply its kernel:

$$\mathcal{V}_\lambda = \ker D\pi(\boldsymbol{\lambda}) = \text{span}\{k(\boldsymbol{\lambda})\}, \quad k(\boldsymbol{\lambda}) := \left(4\lambda_1\lambda_2, 4\lambda_2^2, 0, \lambda_1^2 + 4\lambda_2\lambda_4\right)^\top.$$

A direct row–by–row multiplication shows $D\pi(\boldsymbol{\lambda}) k(\boldsymbol{\lambda}) = 0$, so $k(\boldsymbol{\lambda})$ lies in the kernel. Since $D\pi(\boldsymbol{\lambda})$ has full row rank 3 for every $\boldsymbol{\lambda} \in \Lambda$, the kernel is one–dimensional and $\dim\mathcal{V}_\lambda = 1$. At this stage, no Riemannian metric is needed—the vertical space is determined purely by the quotient map $\pi$.

**Horizontal space.** To define the horizontal space, we must specify a Riemannian metric, as different metrics generally yield different horizontal spaces. In our case, we employ the Fisher-Rao metric, which for regular minimal exponential families equals the Hessian of the logpartition function. For the Normal-Gamma distribution specifically, the Fisher information matrix takes the following form

$$F(\boldsymbol{\lambda}) = \begin{pmatrix} \frac{b}{2\lambda_2 a} - \frac{\lambda_1^2 b}{4\lambda_2^2 a^2} & -\frac{\lambda_1 b}{2\lambda_2^2 a} + \frac{\lambda_1^3 b}{8\lambda_2^3 a^2} & -\frac{\lambda_1}{2\lambda_2 a} & \frac{\lambda_1 b}{2\lambda_2 a^2} \\ -\frac{\lambda_1 b}{2\lambda_2^2 a} + \frac{\lambda_1^3 b}{8\lambda_2^3 a^2} & \frac{1}{2\lambda_2^2} - \frac{\lambda_1^4 b}{16\lambda_2^4 a^2} + \frac{\lambda_1^2 b}{2\lambda_2^3 a} & \frac{\lambda_1^2}{4\lambda_2^2 a} & -\frac{\lambda_1^2 b}{4\lambda_2^2 a^2} \\ -\frac{\lambda_1}{2\lambda_2 a} & \frac{\lambda_1^2}{4\lambda_2^2 a} & \psi_1\left(\frac{1}{2} + \lambda_3\right) & \frac{1}{a} \\ \frac{\lambda_1 b}{2\lambda_2 a^2} & -\frac{\lambda_1^2 b}{4\lambda_2^2 a^2} & \frac{1}{a} & \frac{\frac{1}{2}+\lambda_3}{a^2} \end{pmatrix}, \tag{57}$$

where $a = \frac{\lambda_1^2}{4\lambda_2} - \lambda_4$, $b = -\frac{1}{2} - \lambda_3$, and $\psi_1$ is the trigamma function. Equipped with the Fisher information matrix in natural coordinates (57), the horizontal space is defined as the $F(\boldsymbol{\lambda})$-orthogonal complement of the vertical line $\mathcal{V}_\lambda = \mathrm{span}\{k(\boldsymbol{\lambda})\}$: a tangent vector $v = (v_1, v_2, v_3, v_4)^\top$ is horizontal iff

$$\langle v, k(\boldsymbol{\lambda}) \rangle_F = k(\boldsymbol{\lambda})^\top F(\boldsymbol{\lambda})\, v = 0.$$

The $F(\boldsymbol{\lambda})$-orthogonality condition $k(\boldsymbol{\lambda})^\top F(\boldsymbol{\lambda})\, v = 0$ is equivalent to requiring $v$ to be orthogonal (in the *Euclidean* sense) to the single vector

$$n(\boldsymbol{\lambda}) := F(\boldsymbol{\lambda})\, k(\boldsymbol{\lambda}).$$

A short calculation with the entries of $F(\boldsymbol{\lambda})$ in (57) gives

$$n_1 = \frac{2\lambda_1 b\lambda_4}{a^2}, \qquad n_3 = \frac{4\lambda_2 \lambda_4}{a},$$
$$n_2 = 2 - \frac{\lambda_1^2 b\lambda_4}{\lambda_2 a^2}, \quad n_4 = -\frac{4b\lambda_2 \lambda_4}{a^2},$$

Provided $\lambda_4 \neq 0$ (true on the admissible domain $\Lambda$), we have $n_4 \neq 0$, so the linear constraint $n^\top v = 0$ can be solved explicitly:

$$v_4 = -\frac{n_1}{n_4}\, v_1 - \frac{n_2}{n_4}\, v_2 - \frac{n_3}{n_4}\, v_3.$$

Choosing $v_1, v_2, v_3$ successively as the standard basis vectors produces an $F$-orthogonal basis of the horizontal space:

$$h^{(1)}(\boldsymbol{\lambda}) = (1,\, 0,\, 0,\, -n_1/n_4),$$
$$h^{(2)}(\boldsymbol{\lambda}) = (0,\, 1,\, 0,\, -n_2/n_4),$$
$$h^{(3)}(\boldsymbol{\lambda}) = (0,\, 0,\, 1,\, -n_3/n_4).$$

Any natural gradient $g$ can now be decomposed as $g = g_\| + g_\perp$ with

$$g_\| = (g{\cdot}n)\, \frac{k(\boldsymbol{\lambda})}{k(\boldsymbol{\lambda})^\top n(\boldsymbol{\lambda})} \quad \text{and} \quad g_\perp = g - g_\|,$$

so that $g_\perp \in \mathcal{H}_\lambda$ is the direction used in Algorithm 1.

## C  Normal-Wishart

### C.1  Definition and properties

A random variable $(z, S)$ follows a multivariate Normal-Wishart distribution with parameters $(\mu, \Psi, \kappa, \nu)$ if

$$z|S \sim \mathcal{N}(\mu, (\kappa S)^{-1}) \tag{58}$$
$$S \sim \mathcal{W}(\nu, \Psi) \tag{59}$$

where $\mu \in \mathbb{R}^d$ is the location parameter, $\Psi \in \mathbb{R}^{d \times d}$ is a positive definite scale matrix, $\kappa > 0$ is a scaling parameter, and $\nu > d - 1$ is the degree of freedom parameter.

The joint probability density function of the Normal-Wishart distribution is given by

$$p(z, S|\mu, \Psi, \kappa, \nu) = p(z|S, \mu, \kappa) p(S|\Psi, \nu). \tag{60}$$

### C.2  Marginalization and the Multivariate Student-t

A Normal–Wishart distribution $S \sim \mathcal{W}_d(\nu, \Psi)$, $z \mid S \sim \mathcal{N}(\mu, (\kappa S)^{-1})$ marginalizes to a multivariate Student-$t$ (see [Murphy, 2007, Section 9])

$$p(z \mid \mu, \Psi, \kappa, \nu) = \int p(z, S \mid \mu, \Psi, \kappa, \nu)\, \mathrm{d}S = \mathcal{T}_d(z \mid \mu, \Sigma, \nu'), \tag{61}$$

where $\Sigma = \frac{\Psi^{-1}}{\kappa(\nu - d + 1)}$, $\nu' = \nu - d + 1$. Hence the mapping

$$(\mu, \Psi, \kappa, \nu) \longmapsto \mathcal{T}_d(\mu, \Psi^{-1}/[\kappa(\nu - d + 1)], \nu - d + 1)$$

is many-to-one: a fixed $(\mu, \nu)$ and a fixed product $\kappa\Psi$ determines a unique Student-$t$. So by changing $(\kappa, \Psi)$ while keeping their product fixed, we yield the same Student-$t$.

### C.2.1 Canonical exponential family form

The Normal-Wishart distribution can be written in exponential family form in the following way:

$$p(z, S \mid \boldsymbol{\lambda}) = \exp\left(\boldsymbol{\lambda}^\mathsf{T} T(z, S) - A(\boldsymbol{\lambda})\right), \tag{62}$$

where the sufficient statistics are

$$T(z, S) = \begin{bmatrix} Sz \\ S \\ z^\mathsf{T} S z \\ \log \det S \end{bmatrix}, \tag{63}$$

and the natural parameters are defined trough the standard parameters $(\mu, \Psi, \kappa, \nu)$ in the following way:

$$\boldsymbol{\lambda} = \begin{bmatrix} \lambda_1 \\ \lambda_2 \\ \lambda_3 \\ \lambda_4 \end{bmatrix} = \begin{bmatrix} \kappa\, \mu \\ -\frac{1}{2}(\Psi^{-1} + \kappa\, \mu\mu^\mathsf{T}) \\ -\frac{\kappa}{2} \\ \frac{\nu - d}{2} \end{bmatrix}. \tag{64}$$

Then the log-partition function is

$$A(\boldsymbol{\lambda}) = -\frac{d}{2}\log(-2\lambda_3) - \frac{d + 2\lambda_4}{2}\log \det S \tag{65}$$
$$+ \frac{d(d + 2\lambda_4)}{2}\log(2) + \log \Gamma_d\left(\frac{d + 2\lambda_4}{2}\right) + \frac{d}{2}\log(2\pi).$$

As established in equation (7), the mean parameters are given by the gradient of the log-partition function

$$\boldsymbol{\theta} = \nabla_{\boldsymbol{\lambda}} A(\boldsymbol{\lambda}). \tag{66}$$

For the Normal-Wishart distribution, these parameters are

$$\theta_1 = \frac{\mathrm{d}A(\boldsymbol{\lambda})}{\mathrm{d}\lambda_1} = \mathbb{E}\left[Sz\right] = \nu\Psi\mu,$$

$$\theta_2 = \frac{\mathrm{d}A(\boldsymbol{\lambda})}{\mathrm{d}\lambda_2} = \mathbb{E}\left[S\right] = \nu\Psi,$$

$$\theta_3 = \frac{\mathrm{d}A(\boldsymbol{\lambda})}{\mathrm{d}\lambda_3} = \mathbb{E}\left[z^\mathsf{T} S z\right] = \nu\mu^\mathsf{T}\Psi\mu + \frac{d}{\kappa}, \tag{67}$$

$$\theta_4 = \frac{\mathrm{d}A(\boldsymbol{\lambda})}{\mathrm{d}\lambda_4} = \mathbb{E}\left[\log \det S\right] = \log \det \Psi + d\log 2 + \psi_d\left(\frac{\nu}{2}\right),$$

where $\psi_d$ is the multivariate digamma function.

*Proof.* $\theta_1$ is computed using conditional expectation,

$$\mathbb{E}_{(\nu, S)}\left[S\nu\right] = \mathbb{E}_S\left[S\mathbb{E}_\nu\left[\nu\right]\right] = \mathbb{E}_S\left[S\mu\right] = \nu\Psi\mu. \tag{68}$$

The value of $\theta_2$ is directly the moments of the Wishart distribution Eaton [2007][Proposition 8.3] and $\theta_3$ can be derived from them as follows:

$$\mathbb{E}\left[z^{\mathsf{T}} S z\right] = \sum_{i,j} \mathbb{E}\left[z_i S_{i,j} z_j\right] \tag{69a}$$

$$= \sum_{i,j} \mathbb{E}_S\left[S_{i,j} \mathbb{E}_{z|S}\left[z_i z_j\right]\right] \tag{69b}$$

$$= \sum_{i,j} \mathbb{E}_S\left[S_{i,j}(\mathrm{Cov}(z_i z_j) + \mathbb{E}_{z|S}\left[z_i\right] \mathbb{E}_{z|S}\left[z_j\right])\right] \tag{69c}$$

$$= \sum_{i,j} \mathbb{E}_S\left[S_{i,j}((\kappa S)^{-1}_{i,j} + \mu_i \mu_j)\right] \tag{69d}$$

$$= \kappa^{-1} \sum_{i,j} \mathbb{E}_S\left[S_{i,j}(S)^{-1}_{i,j}\right] + \sum_{i,j} \mathbb{E}_S\left[S_{i,j}\mu_i \mu_j\right] \tag{69e}$$

$$= \kappa^{-1} \mathbb{E}_S\left[\mathrm{tr}\left(S(S)^{-1}\right)\right] + \sum_{i,j} \nu \Psi_{i,j} \mu_i \mu_j \quad as\ S \in \mathbb{S} \tag{69f}$$

$$= \frac{d}{\kappa} + \nu \mu^{\mathsf{T}} \Psi \mu. \tag{69g}$$

$\theta_4$ is direcly the log-expectation of a Wishart distribution given by Penny [2001]. $\qquad\square$

### C.3 Derivation of the NGD update

Let's consider $q(S) = \mathcal{W}_d(S|\nu, \Psi)$ and $q(\mathbf{z}|S) = \mathcal{N}(\mathbf{z}|\mu, (\kappa S)^{-1})$.

We denote the log-likelihood for the $n$'th data point by $f_n(\mathbf{z}) := -\log p(\mathcal{D}_n|\mathbf{z})$ with a Normal-Wishart prior with parameters $\mu = 0, \Psi = \mathrm{I}, \kappa = 1$, and the degree of freedom parameter $\nu_0$.

We use the lower bound defined in the joint distribution, $p(\mathcal{D}, \mathbf{z}, S)$

$$\mathcal{L}(\boldsymbol{\lambda}) = \mathbb{E}_{q(z,S)}\left[\log p(\mathcal{D}, \mathbf{z}, S) - \log q(\mathcal{D}, \mathbf{z}, S)\right]$$

$$= \mathbb{E}_{q(z,S)}\left[\sum_{n=1}^{N} \underbrace{\log p(\mathcal{D}_n|\mathbf{z})}_{:=-f_n(\mathbf{z})} + \log \frac{\mathcal{N}\left(\mathbf{z}|0_d, S^{-1}\right)}{\mathcal{N}\left(\mathbf{z}|\mu, (\kappa S)^{-1}\right)} + \log \frac{\mathcal{W}_d(S|\nu_0, \mathrm{I}_d)}{\mathcal{W}_d(S|\nu, \Psi)}\right].$$

Our goal is to compute the gradient of this ELBO with respect to the expectation parameters $\boldsymbol{\theta}$ (defined in (67)). Because $\boldsymbol{\theta}$ is an invertible re-parameterization of the standard parameters $(\mu, \Psi, \kappa, \nu)$, their gradients are related by the chain rule as follows:

$$\nabla_{\theta_3}\mathcal{L} = -\frac{\kappa^2}{d} \nabla_\kappa \mathcal{L}, \tag{70a}$$

$$\nabla_{\theta_1}\mathcal{L} = \frac{1}{\nu} \Psi^{-1} \nabla_\mu \mathcal{L} - 2\mu \nabla_{\theta_3}\mathcal{L}, \tag{70b}$$

$$\nabla_{\theta_4}\mathcal{L} = \frac{\mathrm{tr}(\Psi \nabla_\Psi \mathcal{L}) - \nu \nabla_\nu \mathcal{L}}{d - \frac{1}{2}\nu\,\psi'_d(\nu/2)}, \tag{70c}$$

$$\nabla_{\theta_2}\mathcal{L} = \frac{1}{\nu}\left[\nabla_\Psi \mathcal{L} - \nu(\nabla_{\theta_1}\mathcal{L})\mu^{\top} - \nu(\nabla_{\theta_3}\mathcal{L})\mu\mu^{\top} - (\nabla_{\theta_4}\mathcal{L})\Psi^{-1}\right], \tag{70d}$$

note that by $\psi'_d$ we denote the multivariate trigamma function.

The ELBO gradients for the standard parameterization can be obtained as follows

$$\nabla_\mu \mathcal{L}(\boldsymbol{\lambda}) = -\sum_{i=1}^N \nabla_\mu \mathbb{E}_{q(z,S)}\left[f_n(z)\right] - \nu \Psi \mu, \tag{71a}$$

$$\nabla_\kappa \mathcal{L}(\boldsymbol{\lambda}) = -\sum_{i=1}^N \nabla_\kappa \mathbb{E}_{q(z,S)}\left[f_n(z)\right] - \frac{d}{2}\frac{\kappa-1}{\kappa}, \tag{71b}$$

$$\nabla_\Psi \mathcal{L}(\boldsymbol{\lambda}) = -\sum_{i=1}^N \nabla_\Psi \mathbb{E}_{q(z,S)}\left[f_n(z)\right] - \frac{\nu}{2}\mu^\top\mu + \frac{1}{2}\Psi, \tag{71c}$$

$$\nabla_\nu \mathcal{L}(\boldsymbol{\lambda}) = -\sum_{i=1}^N \nabla_\nu \mathbb{E}_{q(z,S)}\left[f_n(z)\right] + \frac{d}{2} - \frac{1}{2}\mu^\top\Psi\mu - \frac{\nu-d-1}{4}\psi_d'\left(\frac{\nu}{2}\right), \tag{71d}$$

where $\psi_d'$ is the multivariate trigamma function.

The last thing to instantiate Algorithm 1 for the Normal-Wishart is to implement the projection onto the horizontal space (see Appendix B.2). For the Normal–Wishart lift, the vertical space is one–dimensional, so the vertical subspace at any $\boldsymbol{\lambda}$ is $\mathcal{V}_\lambda = \mathrm{span}\{k(\boldsymbol{\lambda})\}$ with

$$k(\boldsymbol{\lambda}) = (\lambda_1, \mathrm{vec}(\lambda_2), \lambda_3, 0)^\top,$$

the last natural coordinate $\lambda_4$ always effect the marginal. Given the *natural* gradient $g = \widetilde{\nabla}_\lambda \mathcal{L}$, its Fisher–orthogonal projection is obtained by removing the component along $k(\boldsymbol{\lambda})$

$$g_\lambda^\perp = g - \frac{k(\boldsymbol{\lambda})^\top F(\boldsymbol{\lambda})\, g}{k(\boldsymbol{\lambda})^\top F(\boldsymbol{\lambda}) k(\boldsymbol{\lambda})}\, k(\boldsymbol{\lambda}). \tag{72}$$

Because $k(\boldsymbol{\lambda})$ is a single vector, the denominator is a *scalar*; evaluating (72) therefore requires only one call to the Fisher-matrix–vector product and one scalar division; no inversion of the full Fisher matrix is ever needed.

**Using the derivations in this section, we can now summarize our algorithm.** Specializing the generic quotient–natural–gradient loop (Algorithm 1) to the Normal–Wishart lift $(\mu, \Psi, \kappa, \nu)$ gives a fully explicit routine:

1. computes the stochastic data–fit gradients in the *standard* parameter space $(g_\mu^{\mathrm{data}}, g_\kappa^{\mathrm{data}}, g_\Psi^{\mathrm{data}}, g_\nu^{\mathrm{data}})$;
2. adds the analytic prior terms (71a)–(71d);
3. converts the result to the *expectation* coordinates $(g_{\theta_1}, \ldots, g_{\theta_4})$ via the chain rule (70a)–(70d);
4. removes the vertical component with the rank-one projector (72);
5. performs a natural-gradient ascent step of size $\beta_t$ in the horizontal direction and back-transforms to $(\mu, \Psi, \kappa, \nu)$.

The whole procedure, including the projection (72), is collected in Algorithm 2 below.

**Algorithm 2** One step of the quotient natural-gradient update for Normal–Wishart parameters

---

**Input:** current standard parameters $(\mu, \Psi, \kappa, \nu)$, minibatch $\mathcal{B}_t$, dataset size $N$, step size $\beta_t$

▷ *Data–fit contribution*

$$\left(g_\mu^{\text{data}}, g_\kappa^{\text{data}}, g_\Psi^{\text{data}}, g_\nu^{\text{data}}\right) \leftarrow -\frac{N}{|\mathcal{B}_t|} \sum_{n \in \mathcal{B}_t} \nabla_{(\mu, \kappa, \Psi, \nu)} \mathbb{E}_q\big[f_n(z)\big]$$

▷ *Add prior terms* (Eqs. (71a)–(71d))

$$g_\mu \leftarrow g_\mu^{\text{data}} - \nu \Psi \mu,$$

$$g_\kappa \leftarrow g_\kappa^{\text{data}} - \frac{d}{2} \frac{\kappa - 1}{\kappa},$$

$$g_\Psi \leftarrow g_\Psi^{\text{data}} + \frac{\nu}{2}\big(\Psi^{-1} - \mu\mu^\top\big),$$

$$g_\nu \leftarrow g_\nu^{\text{data}} + \frac{d}{2} - \frac{1}{2}\mu^\top \Psi \mu - \frac{\nu - d - 1}{4}\, \psi'_d(\nu/2).$$

▷ *Chain rule* (Eqs. (70a)–(70d))

$$\left(g_{\theta_1}, g_{\theta_2}, g_{\theta_3}, g_{\theta_4}\right) \leftarrow \texttt{ChainRule}(g_\mu, g_\kappa, g_\Psi, g_\nu)$$

▷ *Horizontal projection (rank–one)* (Eq. (72))

$$\alpha \leftarrow \frac{k(\boldsymbol{\lambda})^\top F(\boldsymbol{\lambda})\, g_\theta}{k(\boldsymbol{\lambda})^\top F(\boldsymbol{\lambda})\, k(\boldsymbol{\lambda})}, \qquad g_\theta^\perp \leftarrow g_\theta - \alpha\, k(\boldsymbol{\lambda})$$

▷ *Natural-gradient update in $\boldsymbol{\lambda}$–space*

$$\lambda_i \leftarrow \lambda_i + \beta_t\, g_{\theta,i}^\perp, \qquad i = 1{:}4$$

▷ *Back-transform to standard parameters* (Eq. (64))

$$\kappa \leftarrow -2\lambda_3, \ \ \mu \leftarrow \lambda_1/\kappa, \ \ \Psi^{-1} \leftarrow -2\lambda_2 - \kappa\mu\mu^\top, \ \ \nu \leftarrow 2\lambda_4 + d.$$

---

## C.4 Path-gradients for Normal-Wishart

In Section 5, we implement Algorithm 1 for Student-$t$ distribution through the Normal-Wishart marginal representation. For a concise implementation of the algorithm, refer to Appendix C.3. This implementation requires *unbiased gradient estimators* $\widehat{\nabla}_\mu \mathcal{L}, ; \widehat{\partial}_\kappa \mathcal{L}; \widehat{\nabla}_\Phi \mathcal{L}$. For the Normal–Wishart variational family, these estimators can be obtained from the general gradient form provided in Theorem 4 for a function $f : \mathbb{R}^d \to \mathbb{R}$.

In the following statements, we will use the so-called Lyapunov operator

$$\mathcal{T}_A[Y] : \mathbb{S}^d \to \mathbb{S}^d : \ T_A[Y] = AY + YA. \tag{73}$$

We denote with $\mathcal{T}^{-1}$ the inverse Lyapunov operator, defined by:

$$\mathcal{T}_A^{-1}[B] = Y, \tag{74a}$$

$$\text{with } AY + YA = B, \ \ A \in \mathbb{S}_{++}^d, B \in \mathbb{S}^d. \tag{74b}$$

According to Bartels and Stewart [1972], $A \succ 0$ is a sufficient condition for $\mathcal{T}^{-1}$ to be correctly defined. We will also refer to the operator *Sym* that associates a matrix to the sum of its transpose and itself as follow:

$$Sym : A \in \mathbb{R}^{k \times k} \to A + A^\top, \quad \forall k \in \mathbb{N}^\star. \tag{75}$$

**Theorem 4** (Gradient Identities for the Normal-Wishart Distribution). *For a dimension $d \geq 1$ and parameters $\mu \in \mathbb{R}^d$, $\kappa > 0$, $\Psi \in \mathbb{S}_{++}^d$, $\nu > d + 1$, consider the joint density of the Normal-Wishart distribution*

$$q_{\mu,\kappa,\Psi,\nu}(z,S) = \underbrace{\mathcal{N}\big(z \mid \mu, (\kappa S)^{-1}\big)}_{\phi_{\mu,S}(z)} \underbrace{\mathcal{W}_d\big(S \mid \nu, \Psi\big)}_{\omega_{\nu,\Psi}(S)}.$$

*Let $f : \mathbb{R}^d \to \mathbb{R}$ be a twice-differentiable function that is integrable with respect to $q_{\mu,\kappa,\Psi,\nu}\mathrm{d}z$, and whose first and second derivatives are also integrable. The ensuing gradient identities are valid:*

1. ***Gradient with respect to $\mu$ (Bonnet identity):***

$$\nabla_\mu \mathbb{E}_q[f(z)] = \mathbb{E}_q[\nabla_z f(z)]$$

2. ***Gradient with respect to $\kappa$ (Price identity):***

$$\frac{\partial}{\partial \kappa} \mathbb{E}_q[f(z)] = -\frac{1}{2\kappa^2} \mathbb{E}_q\big[\mathrm{tr}(S^{-1} \nabla_z^2 f(z))\big]$$

3. ***Gradient with respect to $\Phi = \Psi^{-1}$ (Price identity):***

$$\nabla_\Phi \mathbb{E}_q[f(z)] = -\frac{1}{2\kappa}\mathbb{E}_{z,B}\left[\mathcal{T}_{\Phi^{\frac{1}{2}}}^{-1}\left[Sym(\Phi^{-1}B\Phi^{\frac{1}{2}}B^{-1}\Phi^{\frac{1}{2}}\nabla_z^2 f(z)\Phi^{\frac{1}{2}}B^{-1})\right]\right],$$

*where $B \sim \mathcal{W}(\nu, \mathbb{I})$ and $z|B \sim \mathcal{N}\big(\mu, (\kappa\Phi^{-1/2}B\Phi^{-1/2})^{-1}\big)$.*

The gradient of any real function in the mean parametrization (including ELBO) can be straightforwardly deduced from the equations of Theorem 4. Detailed proofs for each identity are provided in Lemmas 1, 2, and 4, respectively.

**Lemma 1** (Bonnet identity for the Normal–Wishart lift). *Under the conditions of the theorem 4, the following identity holds*

$$\nabla_\mu \mathbb{E}_q[f(z)] = \mathbb{E}_q[\nabla_z f(z)]. \tag{76}$$

*Proof.*

$$\nabla_\mu \mathbb{E}_q[f(z)] = \mathbb{E}_{\omega_{\nu,\Psi}(S)}\big[\nabla_\mu \mathbb{E}_{\phi_{\mu,S}(z)}[f(z)]\big] \text{ (dominated convergence + Fubini)} \tag{77}$$

$$= \mathbb{E}_{\omega_{\nu,\Psi}(S)}\big[\mathbb{E}_{\phi_{\mu,S}(z)}[\nabla_z f(z)]\big] \text{ (by Lin et al. [2025, Theorem 1])} \tag{78}$$

$$= \mathbb{E}_q[\nabla_z f(z)]. \tag{79}$$

$\square$

The proof above employs the vanishing surface term, mirroring the classical Bonnet proof (in French) [Bonnet, 1964]. A more contemporary explanation of the same finding is provided in Lin et al. [2025, Theorem 1]).

**Lemma 2** ($\kappa$–Price identity for the Normal–Wishart lift). *Under the conditions of the theorem 4 the following identity holds*

$$\frac{\partial}{\partial \kappa} \mathbb{E}_q[f(z)] = -\frac{1}{2\kappa^2} \mathbb{E}_q\Big[\mathrm{tr}\big(S^{-1} \nabla_z^2 f(z)\big)\Big]. \tag{80}$$

*Proof.* Let $\phi_{\mu,S}(z) = \mathcal{N}(z \mid \mu, \Sigma)$ with $\Sigma = (\kappa S)^{-1}$ then

$$\partial_\kappa \mathbb{E}_q[f] = \mathbb{E}_{\omega_{\nu,\Psi}}\Big[\partial_\kappa \mathbb{E}_{\phi_{\mu,S}}[f]\Big] \qquad\qquad \text{(Fubini's theorem)}$$

$$= \mathbb{E}_{\omega_{\nu,\Psi}}\Big[\big\langle \partial_\kappa \Sigma, \nabla_\Sigma \mathbb{E}_{\phi_{\mu,S}}[f]\big\rangle\Big] \qquad\qquad \text{(chain rule)}$$

$$= \mathbb{E}_{\omega_{\nu,\Psi}}\Big[\big\langle -\kappa^{-2}S^{-1}, \tfrac{1}{2}\mathbb{E}_{\phi_{\mu,S}}[\nabla_z^2 f]\big\rangle\Big] \qquad \text{(by [Lin et al., 2025, Theorem 4])}$$

$$= -\frac{1}{2\kappa^2}\mathbb{E}_q\big[\mathrm{tr}(S^{-1}\nabla_z^2 f(z))\big]$$

The first line exchanges the differentiation operator and integration operator, which is possible because the derivative of $q_{\mu,\kappa,\Psi,\nu}$ can be bounded from above by an integrable function. The second applies the chain rule. The third uses two facts: (1) $\Sigma = \kappa^{-1}S^{-1}$ implies $\partial_\kappa \Sigma = -\kappa^{-2}S^{-1}$, and (2) the classical Price formula [Lin et al., 2025, Theorem 4] $\nabla_\Sigma \mathbb{E}_\phi[f] = \frac{1}{2}\mathbb{E}_\phi[\nabla_z^2 f]$. The final line simplifies using the trace inner product, the linearity of the trace, and the expectation. $\square$

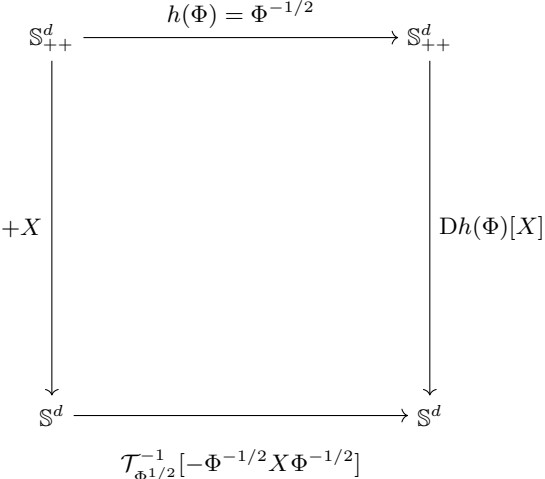

**Figure 3:** Commutativity diagram for Lemma 3. The following commutative diagram illustrates the Fréchet differentiability of the inverse square-root map $h : \Phi \mapsto \Phi^{-1/2}$ on the space of symmetric positive-definite matrices $\mathbb{S}_{++}^d$. The vertical arrows represent perturbations in the input space and the corresponding linearized response in the output space via the derivative $\mathrm{D}h(\Phi)$. This diagram expresses the fact that applying a small symmetric perturbation $X \in \mathbb{S}^d$ to the input $\Phi$ corresponds, under the linearization of $h$, to a symmetric output given by the Lyapunov operator. The bottom arrow represents this linear transformation. Commutativity of the diagram means that the effect of first perturbing $\Phi$ and then applying $h$, versus first applying $h$ and then differentiating, yields the same result to first order in $X$.

**Lemma 3** (Fréchet differential of the inverse square-root). *Let $\Phi \in \mathbb{S}_{++}^d$. The map $h : \Phi \mapsto \Phi^{-\frac{1}{2}}$ is Fréchet differentiable with:*

$$\mathrm{D}h(\Phi) : X \in \mathbb{S}^d \to \mathcal{T}_{\Phi^{\frac{1}{2}}}^{-1}[-\Phi^{-\frac{1}{2}}X\Phi^{-\frac{1}{2}}] \in \mathbb{S}^d \,. \tag{81}$$

Figure 3 illustrates the commutative structure of the differential relationship provided in the Lemma 3.

*Proof.* The Fréchet differentiability of the square root in $\mathbb{S}_{++}^d$ is a direct implication of Moral and Niclas [2018, Theorem 1.1]. In

$$\Phi^{-\frac{1}{2}}\Phi\Phi^{-\frac{1}{2}} = Id \,, \tag{82}$$

we substitute the functions with their respective Taylor expansion at point $\Phi$ in a direction $X \in \mathbb{S}^d$. With that substitution, we get the following:

$$\left(\Phi^{-\frac{1}{2}} + \mathrm{D}h(\Phi)[X] + o(\|X\|)\right)(\Phi + X)\left(\Phi^{-\frac{1}{2}} + \mathrm{D}h(\Phi)[X] + o(\|X\|)\right) = Id \tag{83a}$$

$$\Phi^{-\frac{1}{2}}\Phi\Phi^{-\frac{1}{2}} + \Phi^{-\frac{1}{2}}X\Phi^{-\frac{1}{2}} + \Phi^{-\frac{1}{2}}\Phi\mathrm{D}h(\Phi)[X] + \mathrm{D}h(\Phi)[X]\Phi\Phi^{-\frac{1}{2}} + o(\|X\|) = Id \tag{83b}$$

$$\Phi^{-\frac{1}{2}}X\Phi^{-\frac{1}{2}} + \Phi^{\frac{1}{2}}\mathrm{D}h(\Phi)[X] + \mathrm{D}h(\Phi)[X]\Phi^{\frac{1}{2}} + o(\|X\|) = 0 \,. \tag{83c}$$

Given that $\Phi \succ 0$, equation (83c) implies that we can define $\mathrm{D}h(\Phi)[X]$ as $\mathcal{T}_{\Phi^{\frac{1}{2}}}^{-1}[-\Phi^{-\frac{1}{2}}X\Phi^{-\frac{1}{2}}]$, which is precisely the statement of the lemma. $\qquad\square$

**Lemma 4** ($\Phi$–Price identity, Lyapunov version). *Under the conditions of the theorem 4 the following identity holds*

$$\nabla_\Phi \mathbb{E}_q[f(z)] = -\frac{1}{2\kappa}\mathbb{E}_{z,B}\left[\mathcal{T}_{\Phi^{\frac{1}{2}}}^{-1}\left[Sym(\Phi^{-1}B\Phi^{\frac{1}{2}}B^{-1}\Phi^{\frac{1}{2}}\nabla_z^2 f(z)\Phi^{\frac{1}{2}}B^{-1})\right]\right] \,, \tag{84}$$

*where $B \sim \mathcal{W}(\nu, \mathbb{I})$, $z|B \sim \mathcal{N}(\mu, (\kappa\Phi^{-1/2}B\Phi^{-1/2})^{-1})$ and $Sym(A) := \frac{1}{2}(A + A^\top)$.*

*Proof.* We compute the gradient of $\Phi \mapsto \mathbb{E}_{\mathcal{N}(z|\mu,(\kappa\Phi^{-1/2}B\Phi^{-1/2})^{-1})}[f(z)]$ by treating it as the composition of two functions: first, $\phi : \Phi \mapsto (\kappa\Phi^{-1/2}B\Phi^{-1/2})^{-1}$, and second, $\sigma : \Sigma \mapsto \mathbb{E}_{\mathcal{N}(z|\mu,\Sigma)}[f(z)]$.

$$\nabla_\Phi(\sigma \circ \phi)) = (D\phi(\Phi))^*\left[\nabla_{\phi(\Phi)}\sigma\right], \tag{85}$$

where $(D\phi(\Phi))^*$ represents the adjoint operator of $D\phi(\Phi)$.

Under the conditions on $f$ from Theorem 1, we can apply Lin et al. [2025][Theorem 4] to obtain the following: $\nabla_{\phi(\Phi)}\sigma = \frac{1}{2}\mathbb{E}_{\mathcal{N}(z|\mu,\phi(\Phi))}\left[\nabla_z^2 f(z)\right]$.

We express $\phi$ as the composition of three functions:

$$\phi_1 : \Phi \in \mathbb{S}_{++} \mapsto \Phi^{-\frac{1}{2}} \quad \text{with differential} \quad D\phi_1(\Phi) : X \in \mathbb{S} \mapsto \mathcal{T}_{\Phi^{\frac{1}{2}}}^{-1}\left[\Phi^{-\frac{1}{2}}X\Phi^{-\frac{1}{2}}\right], \quad \text{(86a)}$$

$$\phi_2 : A \in \mathbb{S}_{++} \mapsto ABA \quad \text{with differential} \quad D\phi_2(A) : X \in \mathbb{S} \mapsto ABX + XBA, \quad \text{(86b)}$$

$$\phi_3 : S \in \mathbb{S}_{++} \mapsto (\kappa S)^{-1} \quad \text{with differential} \quad D\phi_3(S) : X \in \mathbb{S} \mapsto -\kappa^{-1}S^{-1}XS^{-1}. \quad \text{(86c)}$$

We recall that any Riemannian metric on the manifold $\mathbb{S}_{++}$ can be expressed as $\langle X, Y \rangle \mapsto \mathrm{tr}(\Phi^{-1}X\Phi^{-1}Y)$ where $\mathbb{S}$ is isomorphic to the tangent space of $\mathbb{S}_{++}$ at $\Phi$ and $X,Y \in \mathbb{S}$ (see Ohara et al. [1996]). Based on the form of the Riemannian metric on the tangent space of $\mathbb{S}_{++}$, we can state that for any $A, \Lambda \in \mathbb{S}_{++}$ the differentials of $D\phi_2(A)$ and $D\phi_3(\Lambda)$ are self-adjoint. According to [Tippett et al., 2000], $\mathcal{T}^{-1}$ is also self-adjoint, making $D\phi_1(\Phi)$ self-adjoint for any $\Phi \in \mathbb{S}_{++}$. The differential $D\phi(\Phi)$, $\Phi \in \mathbb{S}_{++}$ is then self-adjoint as the composition of self-adjoint operators. The gradient of $\sigma \circ \phi$ can be expressed using the formula (85) as follows:

$$\nabla_\Phi(\sigma \circ \phi)) = (D\phi_3 \circ \phi_2 \circ \phi_1(\Phi))^*[\nabla_{\phi(\Phi)}\sigma] \tag{87a}$$

$$= \left(D\phi_3(\Phi^{-\frac{1}{2}}B\Phi^{-\frac{1}{2}}) \circ D\phi_2(\Phi^{-\frac{1}{2}}) \circ D\phi_1(\Phi)\right)^* [\nabla_{\phi(\Phi)}\sigma] \tag{87b}$$

$$= D\phi_1(\Phi)^* \circ D\phi_2(\Phi^{-\frac{1}{2}})^* \circ D\phi_3(\Phi^{-\frac{1}{2}}B\Phi^{-\frac{1}{2}})^*[\nabla_{\phi(\Phi)}\sigma] \tag{87c}$$

$$= -\kappa^{-1}\mathcal{T}_{\Phi^{\frac{1}{2}}}^{-1}\left[Sym(\Phi^{-1}B\Lambda^{-1}\nabla_{\phi(\Phi)}\sigma\Lambda^{-1}\Phi^{-\frac{1}{2}})\right] \tag{87d}$$

$$= -\kappa^{-1}\mathcal{T}_{\Phi^{\frac{1}{2}}}^{-1}\left[Sym(\Phi^{-1}B\Phi^{\frac{1}{2}}B^{-1}\Phi^{\frac{1}{2}}\nabla_{\phi(\Phi)}\sigma\Phi^{\frac{1}{2}}B^{-1})\right] \tag{87e}$$

$$= -\frac{\kappa^{-1}}{2}\mathcal{T}_{\Phi^{\frac{1}{2}}}^{-1}\left[Sym(\Phi^{-1}B\Phi^{\frac{1}{2}}B^{-1}\Phi^{\frac{1}{2}}\mathbb{E}_{\mathcal{N}(z|\mu,\phi(\Phi))}\left[\nabla_z^2 f(z)\right]\Phi^{\frac{1}{2}}B^{-1})\right] \tag{87f}$$

We can apply our formula (87f) directly under the expectation over $B \sim \mathcal{W}(Id, \nu)$ and under the linear operator $\mathcal{T}^{-1}$ to obtain our final gradient as follows:

$$\nabla_\Phi\mathbb{E}_{z,B}\left[f(z)\right] = \mathbb{E}_B\left[\nabla_\Phi\mathbb{E}_{z|B}\left[f(z)\right]\right] \tag{88a}$$

$$= \mathbb{E}_B\left[\nabla_\Phi\nabla_\Phi(\sigma \circ \phi))\right] \tag{88b}$$

$$= \mathbb{E}_B\left[-\frac{\kappa^{-1}}{2}\mathcal{T}_{\Phi^{\frac{1}{2}}}^{-1}\left[Sym(\Phi^{-1}B\Phi^{\frac{1}{2}}B^{-1}\Phi^{\frac{1}{2}}\mathbb{E}_{z|B}\left[\nabla_z^2 f(z)\right]\Phi^{\frac{1}{2}}B^{-1})\right]\right] \tag{88c}$$

$$= -\frac{\kappa^{-1}}{2}\mathbb{E}_{z,B}\left[\mathcal{T}_{\Phi^{\frac{1}{2}}}^{-1}\left[Sym(\Phi^{-1}B\Phi^{\frac{1}{2}}B^{-1}\Phi^{\frac{1}{2}}\nabla_z^2 f(z)\Phi^{\frac{1}{2}}B^{-1})\right]\right]. \tag{88d}$$

$$\square$$

# D   Experimental setup and reproducibility protocol

**Benchmarks.**

**Pre-processing and splits.** (1) 80/20 stratified train–test split with `random_state=42`; (2) feature-wise standardisation using training means/variances only.

**Models and inference schemes.** All tasks use Bayesian logistic regression (BLR). We compare three variational-inference schemes:

| Abbrev. | Variational family | Optimiser |
|---------|--------------------|-----------|
| BBVI* | Student-$t$ | Black-box VI [Roeder et al., 2017] |
| NG-LIN | Student-$t$ | Natural-gradient VI of Lin et al. [2020a] |
| NG-Ours | Normal–Wishart (lift) | Quotient Natural Gradient (Alg. 1) |

**Optimisation schedules (parameter-free).** To eliminate hand-tuned learning rates, we use the *Distance-over-Gradients* (DoG) rule of Ivgi et al. [2023] and its Riemannian generalisation (RDoG) [Dodd et al., 2024]. Both schedules set the step size adaptively from quantities the algorithm can measure on-the-fly.

**Euclidean DoG (for BBVI*).** Let $x_t$ be the parameters and $g_t$ the Euclidean gradient. Maintain

$$\bar{r}_t \;=\; \max\!\Big(\epsilon,\, \max_{s\le t}\|x_s - x_0\|_2\Big), \qquad G_t \;=\; \sum_{i\le t}\|g_i\|_2^2,$$

and set

$$\eta_t \;=\; \frac{\bar{r}_t}{\sqrt{G_t}}, \qquad x_{t+1} \;=\; x_t - \eta_t\, g_t.$$

We use $\epsilon = 10^{-3}$.

**Riemannian DoG (for NG-LIN and NG-Ours).** Replace Euclidean norms by natural-gradient norms and the Euclidean distance by the geodesic distance $d(\cdot,\cdot)$ associated with the Fisher–Rao metric:

$$\bar{r}_t \;=\; \max\!\Big(\epsilon,\, \max_{s\le t} d(x_s, x_0)\Big), \quad G_t \;=\; \sum_{i\le t}\|g_i\|_{g,x_i}^2, \quad \eta_t \;=\; \frac{\bar{r}_t}{\sqrt{\zeta_\kappa(\bar{r}_t)\, G_t}},$$

$$x_{t+1} \;=\; \exp_{x_t}\!\big(-\eta_t\, g_t\big).$$

We set $\epsilon = 10^{-3}$ and, unless noted, use the non-positive curvature correction $\zeta_\kappa \equiv 1$ (i.e. $\kappa = 0$). For NG-LIN, $d(\cdot,\cdot)$ is approximated by the symmetric KL between two Student-$t$ distributions, estimated with a fixed set of 64 Monte-Carlo draws anchored at the start point to reduce variance. For NG-Ours, $d(\cdot,\cdot)$ is the *exact* KL in the *lifted* minimal exponential family (Normal–Wishart), available in closed form.

**Safety of the lift-based distance.** Let $\pi : \Lambda \to \Xi$ be the marginalisation map from the lift to the marginal parameters. The quotient-metric result (Theorem 2) implies that, for $\lambda_i \in \Lambda$ with $\xi_i = \pi(\lambda_i)$,

$$\mathrm{KL}\big(q_{\lambda_1} \,\|\, q_{\lambda_2}\big) \;\ge\; \mathrm{KL}\big(q_{\xi_1} \,\|\, q_{\xi_2}\big).$$

Thus the lifted KL we plug into RDoG is an *upper bound* on the (unknown) marginal KL. Because DoG/RDoG chooses $\eta_t = \bar{r}_t/\sqrt{\zeta_\kappa(\bar{r}_t)\, G_t}$, a larger distance yields a (mildly optimistic) larger step. A tighter, future alternative is the fibre-minimised lift distance, $\inf_{\lambda_i \in \pi^{-1}(\xi_i)} \mathrm{KL}(q_{\lambda_1}\|q_{\lambda_2}) = \mathrm{KL}(q_{\xi_1}\|q_{\xi_2})$, i.e. the true quotient metric.

**Hyper-parameters (shared).**

- **Epochs** $= 8{,}000$; **mini-batch size** $= 32$.
- **Step sizes:** *no hand-tuned learning rate*. DoG/RDoG schedules determine $\eta_t$ with $\epsilon = 10^{-3}$; curvature correction disabled by default ($\kappa = 0$).
- **Monte-Carlo samples:** 10 per update for BBVI*; 1 for NG variants (gradients), plus 64 fixed draws for the NG-LIN symmetric-KL distance used by RDoG.

**Software environment.** Python 3.11 (CPU-only); jax 0.6.0, numpy 2.2.4, scikit-learn 1.6.1, torch 2.6.0, pandas 2.2.3. A version-pinned pyproject.toml is included in the repository.

**Hardware and runtime.** All experiments run on a single CPU-only machine (no GPU/TPU). End-to-end wall-clock time to regenerate Table 2 is **2 hours**.

**Reporting.** For every (method, dataset) pair we report: (1) posterior-mean accuracy ($\mathrm{acc}_\mu$), (2) its standard error of the mean (SEM). Results are produced by `python run_vi_comparison.py`.

**Reproducibility assets.**

- **Code** (MIT): `https://github.com/biaslab/QBLR`. One command reproduces all numbers and figures.
- **Determinism.** NumPy, JAX and scikit-learn PRNGs fixed to $42$; JAX in deterministic mode.
- **Environment capture.** `pyproject.toml` and a generated `requirements-lock.txt` freeze packages; a Markdown "compute card" records CPU model, cores, OS, and energy draw.

## E Complexity Analysis

In Algorithm 1, two operations have a non-trivial computational complexity: the natural gradient computation and its projection onto the horizontal space. Based on the current literature, we analyze these complexities. In Appendix E.1, we explain why projecting the natural gradient is negligible compared to its estimation. In Appendix E.2, in the context of the Normal-Wishart example, we propose a methodology to reduce the complexity of the natural gradient estimation.

We thank anonymous reviewers for raising this question.

### E.1 Complexity Analysis

**Projection computational cost.** The projection operator $P_{\mathcal{H}}(\boldsymbol{\lambda})$ defined in Equation (16) is used to compute the horizontal component of the natural gradient. For clarity, we denote the projection of the natural gradient $g_\theta \in \mathbb{R}^{\dim \Lambda}$ as follows:

$$g_H = P_{\mathcal{H}}(\boldsymbol{\lambda})[g_\theta].$$

A naïve approach to compute $g_H$ includes the inversion of the symmetric positive definite matrix

$$RV(\boldsymbol{\lambda}) := K(\boldsymbol{\lambda})^\top F(\boldsymbol{\lambda}) \, K(\boldsymbol{\lambda}) \in \mathbb{R}^{d_v \times d_v} \,,$$

where $K(\boldsymbol{\lambda})$ is the matrix representation of a basis of the vertical space $\ker D\pi(\boldsymbol{\lambda})$ (33) (with $\pi : \Lambda \to \Xi$ is the marginalization map (10)), $F(\boldsymbol{\lambda})$ the Fisher information matrix (3a), $d_v = \dim V_\lambda$ (with $V_\lambda$ the vertical space). This approach would require computing the matrix inverse, with cost $\mathcal{O}(d_v^3)$. However, we avoid materializing the inverse. Instead, we solve the linear system

$$RV(\boldsymbol{\lambda})v = K(\boldsymbol{\lambda})^\top F(\boldsymbol{\lambda}) \, g_\theta \,, \tag{89}$$

using a small dense linear solver: due to the fact that $RV(\boldsymbol{\lambda})$ is symmetric positive definite, the conjugate gradient solver by Hestenes et al. [1952] is applicable. This reduces the computational cost to $\mathcal{O}(d_v^2)$.

In practice, a well-constructed lifting ensures that $d_v \ll \dim \Lambda$, making the cost of projection negligible relative to the Fisher–vector product $F(\boldsymbol{\lambda}) \, g_\theta$, which has cost $\mathcal{O}(\dim \Lambda^2)$.

In our main Normal–Wishart case, the vertical space is one-dimensional ($d_v = 1$), so the projection reduces to a single scalar division.

### E.2 Future Improvement

Quadratic time is acceptable up to a few hundred dimensions on commodity hardware, but larger problems call for additional structure. In the following, we propose our plan to reduce computational complexity.

Let us denote by $d$ the dimension of a sample. We propose two different factorization methods to reduce the $\mathcal{O}(d^2)$ computational cost of the natural gradient while preserving its geometric property.

**Structured covariances**

Restrict the scale to $B$ blocks $\Psi = \mathrm{diag}(\Psi_1, \ldots, \Psi_B)$ with sizes $d_b$.

- **Sampling**: $\sum_b \mathcal{O}(d_b^2)$; purely diagonal $\Psi$ needs only $\mathcal{O}(d)$ Gamma draws.
- **Fisher products & Hessian trace** both factor block-wise, yielding the same $\sum_b \mathcal{O}(d_b^2)$ and $\mathcal{O}(d)$ in the diagonal case.

**Low-rank with diagonal factorization**

Decompose the scale matrix as

$$\Psi \;=\; LL^\top \;+\; \mathrm{diag}(v), \qquad L \in \mathbb{R}^{d \times k}, \; k \ll d,$$

so that only $kd + d$ free parameters are stored instead of $\frac{1}{2}d(d+1)$.

- **Sampling.** Each column of $L$ is drawn from a matrix-normal and the diagonal entries of $v$ from independent Gammas. The two draws require $kd$ and $d$ random numbers, respectively, hence $\mathcal{O}(kd)$ time and memory.
- **Fisher–vector products.** In the horizontal projector we need $y \;=\; \big(\mathrm{diag}(v) + LL^\top\big)x$. Compute it as

$$y = \underbrace{\mathrm{diag}(v)\,x}_{\mathcal{O}(d)} \;+\; \underbrace{L\big(L^\top x\big)}_{2\,\mathcal{O}(kd)}.$$

  Both multiplies with $L$ cost $\mathcal{O}(kd)$, so the total is $\mathcal{O}(kd)$ per Fisher product—linear in $d$ for any fixed rank $k$.
- **Hessian trace (two Price identities).** Both the $\kappa$-Price and the $\Phi$-Price terms require $\mathrm{tr}(S^{-1}\nabla_z^2 f)$. Rather than forming the dense Hessian, we use the Hutchinson estimator [Hutchinson, 1989]

$$\mathrm{tr}(S^{-1}\nabla^2 f) \;=\; \tfrac{1}{R}\sum_{r=1}^{R} u_r^\top\big(S^{-1}\nabla^2 f\big)u_r, \quad u_r \sim \{\pm 1\}^d.$$

  Each term needs one Hessian–vector product (HVP) and one multiplication with $S^{-1}$. The HVP is model-specific; the $S^{-1}$–vector multiply uses the Woodbury identity:

$$S^{-1}x \;=\; D^{-1}x \;-\; D^{-1}L\big(I_k + L^\top D^{-1}L\big)^{-1}L^\top D^{-1}x, \quad D = \mathrm{diag}(v),$$

  which is again $\mathcal{O}(kd)$. Choosing $R \le k$ probes keeps the overall trace cost bounded by $\mathcal{O}(k^2 d)$.

Both variants leave the vertical space one-dimensional, so the horizontal projection stays a single scalar divide.

**Summary**

Dense QBLR is $\mathcal{O}(d^2)$; with a diagonal or block-diagonal scale it drops to $\mathcal{O}(d)$, and with rank-$k$ plus diagonal it is $\mathcal{O}(k^2 d)$ (linear in $d$ for fixed $k$). These paths scale QBLR to far larger latent spaces without sacrificing its geometry or requiring matrix inversions.

