# OpenReview forum: "The Quotient Bayesian Learning Rule"
_NeurIPS.cc/2025/Conference — NeurIPS 2025 poster_

### Official Review · Reviewer_Y5N7 · 2025-06-20

**Clarity:** 3
**Significance:** 2
**Originality:** 2
**Rating:** 5
**Confidence:** 4

**Summary:**

The paper introduces the Quotient Bayesian Learning Rule (QBLR), an extension of natural-gradient variational inference used for probabilistic models whose variational families lie outside the exponential family. Drawing on the observation that many heavy-tailed distributions (e.g. Student-t) arise as marginals of minimal exponential families (e.g. Normal–Wishart), the authors show that the latent family’s Fisher–Rao geometry induces a well-defined Riemannian structure on the marginal parameter space via a quotient-manifold construction. Building on this geometry, they give Algorithm 1, which (i) lifts the variational objective into the exponential-family space, (ii) computes ordinary (Euclidean) gradients in the joint natural parameters, (iii) projects those gradients onto the horizontal subspace orthogonal to the fibres of the marginalisation map, and (iv) takes natural-gradient steps that are parameterization-invariant. Specialising to the multivariate Student-t via its Normal–Wishart scale-mixture representation, they derive a quotient natural-gradient update that avoids the Fisher inverse. Empirically, they benchmark QBLR against (a) score-function–free black-box VI (BBVI-NS) and (b) the curved-exponential-family method of Lin et al. (2020) on Bayesian logistic regression across four UCI datasets.

**Questions:**

* Are there other examples that you could have considered where the quotient Bayesian rule would be expected to display a significant improvement over existing approaches? Related to this, how does QBLR perform on higher dimensional examples?
* Would it be possible to include a section, or perhaps a figure, in the paper to help readers gain some intuition for how the quotient manifold works?
* The framework seems to be quite general, but the authors do not seem to propose a wide range of models which QBLR would be appropriate for. Could more information be given on other model classes beyond the heavy tail setting?

**Ethical Concerns:**

["NO or VERY MINOR ethics concerns only"]

**Final Justification:**

The authors have provided some excellent additions to the paper, reflecting the suggestions of multiple reviewers, which I believe will enhance the quality of this work.

**Limitations:**

The authors do not adequately address the limitations of their work. In the case of societal issues, there are no obvious issues that the authors would need to address.

**Quality:**

3

**Strengths And Weaknesses:**

Quality

The paper is presented with a high degree of theoretical rigour, with both the manifold-structure theorem and the induced Fisher–Rao metric theorem stated clearly and proved in full detail. The algorithmic exposition is similarly precise: Algorithm 1 and its specialisation to the multivariate Student-t family are given in closed-form, and practical implementation details, such as constructing bases for vertical subspaces and avoiding large Fisher inversions, are addressed in the paper. Moreover, by exploiting the Normal–Wishart lift, each update reduces to small-scale linear solves, ensuring computational efficiency which means the proposed approach is potentially viable for moderate-dimensional models. On the downside, the empirical scope is relatively narrow: all benchmarks are confined to Bayesian logistic regression on UCI datasets, leaving open questions about performance in higher-dimensional settings or even for other models in general. In addition, the framework’s reliance on identifying a minimal, regular exponential-family representation means that extending QBLR to other heavy-tailed or skewed distributions may be nontrivial and is not yet demonstrated.

Clarity

The paper is well-organised in a logical, (mostly) reader-friendly sequence. The paper begins with motivation and background and then moving on to geometric constructions and the algorithm itseflf. Plus, the experiments are well-presented, as well is the running Normal–Gamma example. The results are also clearly presented. However, the paper is quite dense and the information-geometric notation (quotient manifolds, horizontal lifts, fibre bundles, etc.) would be a steep learning curve for those without a differential-geometry background. Some additional (and quite useful) technical definitions are relegated to the appendices. The heavy reliance on the supplementary material does disrupt the narrative flow for readers who would find it easier to read a self-contained version of the paper within the main text.

Significance

By framing heavy-tailed variational inference as an optimisation problem on a quotient manifold, the authors provide a new geometric template that subsumes classical Bayesian learning rules for exponential families and earlier non-EF extensions. This general framework offers a more robust, parameterisation-invariant inference scheme in the presence of outliers or multimodal posteriors, potentially broadening the applicability of variational methods to real-world problems where heavy tails are common. Nonetheless, the paper stops short of validating this promise on more challenging models—such as high-dimensional latent spaces or deep hierarchies or really anything other than logistic regression—and does not compare against state-of-the-art non-natural-gradient VI techniques (for example, advanced black-box methods with control variates). The results that are stated are also not significantly better than competing methods, perhaps with the exception of the Sonar data example. As a result, the full practical significance of QBLR remains to be established.

Originality

The introduction of the quotient-manifold perspective for variational inference represents an interesting geometric insight, and the horizontal-space projection mechanism ensures parameterisation invariance across arbitrary smooth reparameterisations. These contributions stand out as original within the information-geometry literature. At the same time, the paper could engage more deeply with related work—particularly Lie-group BLR approaches that also aim to preserve geometric invariances under group actions—and clarify the precise distinctions. Moreover, although the Student-t instantiation is useful, it follows straightforwardly from the general theory and may strike some readers as an incremental extension rather than a wholly new algorithmic advance.

---

> ### Author Rebuttal · Authors · 2025-07-29
>
> We appreciate the reviewer's thoughtful questions and constructive feedback.
>
> > We apologize if some equations do not render perfectly; we have done our best to ensure compatibility with the OpenReview markdown compiler.
>
> ## Question 1
>
> "Are there other examples that you could have considered where the quotient Bayesian rule would be expected to display a significant improvement over existing approaches? Related to this, how does QBLR perform on higher dimensional examples?"
>
> ## Answer 1
>
> ### Are there other examples that you could have considered where the quotient Bayesian rule would be expected to display a significant improvement over existing approaches?
>
> However, we do not have a complete answer to this question; we can illustrate our intuition with the following example. Consider the negative binomial distribution introduced by Greenwood and Yule (1920); it's a distribution over $\{0, 1, 2, \dots \}$ with the following probability density:
>
> $$
> f(k| r, p) = \binom{k+r-1}{k} (1-p)^k p^{r},~r>0, p \in (0, 1).
> $$
>
> This distribution has the following integral representation, the so-called Poisson-Gamma mixture:
>
> $$
> f(k| r, p) = \int_{0}^{\infty} \mathcal{P}(k|\lambda)\mathcal{G}\left(\lambda \big{|}r, \frac{p}{1-p}\right) \mathrm{d} \lambda,
> $$
>
> where $\mathcal{P}$ and $\mathcal{G}$ represent the Poisson and the Gamma densities, respectively. The negative binomial distribution is not within the exponential family. However, the joint distribution (the under integral expression) over $(k, \lambda)$ is an exponential family distribution, which can be represented as a minimal and regular exponential family distribution. The negative binomial distribution is known to be within the exponential family once $r$ is fixed.
>
> This example shows a contrast between the initial definition of the negative binomial distribution and its restricted definition ($r$ fixed). Natural gradient descent cannot be performed on the negative binomial distribution family, but it can be performed on the exponential family structure of the Gamma-Poisson mixture (the integral representation above). The QBLR method enables one to transform the natural gradient descent on the Gamma-Poisson mixture into an efficient gradient descent into the negative binomial distribution family at low cost.
>
> In the restricted case, the family is an exponential family distribution, so natural gradient descent optimisation is already possible. A natural gradient on the Gamma-Poisson representation (the integral representation above) will then be equivalent and will not produce any benefit but will introduce a computational burden: we will run optimisation in the higher-dimensional space.
>
> ### Related to this, how does QBLR perform on higher‑dimensional examples?
>
> Our current benchmarks already span almost an order of magnitude in feature dimensionality—from $d=8$ (Diabetes) to $d=60$ (Sonar)—and therefore in parameter dimensionality vary from 72 to 3660. Across this range, QBLR delivers consistent gains over both BBVI‑NS and NG‑LIN; however, we acknowledge that, by modern standards, these are still small-scale problems.
>
> **Computational cost.** With a *dense* Normal–Wishart lift, one QBLR iteration is driven by three quadratic terms (see Appendix A.3):
>
> 1. *Wishart draw.* Sampling a full $d \times d$ scale matrix costs $\mathcal{O}(d^{2})$.
> 2. *Fisher products.* A Fisher–vector products $F(\boldsymbol{\lambda})g_{\theta}$ costs $\mathcal{O}(d^{2})$ but *never* require a matrix inverse.
> 3. *Hessian trace.* The Price identities (see Appendix D) that involve $\mathcal{O}(d^{2})$ when the Hessian is formed explicitly.[^1]
>
> Quadratic time is acceptable up to a few hundred dimensions on commodity hardware, but larger problems call for additional structure. In the following, we propose our plan to reduce computational complexity.
>
> **Routes to larger $d$.** Two common factorizations reduce *all three* $\mathcal{O}(d^{2})$ components while preserving the quotient geometry:
>
> 1. **Structured covariances.** Restrict the scale to $B$ blocks $\Psi=\operatorname{diag}(\Psi_{1},\ldots,\Psi_{B})$ with sizes $d_{b}$.
>    - **Sampling**: $\sum_{b} \mathcal{O}(d_{b}^{2})$; purely diagonal $\Psi$ needs only $\mathcal{O}(d)$ Gamma draws.
>    - **Fisher products & Hessian trace** both factor block‑wise, yielding the same $\sum_{b} \mathcal{O}(d_{b}^{2})$ and $\mathcal{O}(d)$ in the diagonal case.
>
> 2. **Low‑rank + diagonal factorisation.** Decompose the scale matrix as
>
>    $$
>    \Psi = L L^{\top} + \operatorname{diag}(v), \qquad L\in\mathbb{R}^{d\times k}, k\ll d,
>    $$
>
>    so that only $k d + d$ free parameters are stored instead of $\frac{1}{2} d(d+1)$.
>
>    - **Sampling.** Each column of $L$ is drawn from a matrix‑normal and the diagonal entries of $v$ from independent Gammas. The two draws require $k d$ and $d$ random numbers, respectively, hence $\mathcal{O}(k d)$ time and memory.
>
>    - **Fisher–vector products.** In the horizontal projector we need $y = \bigl(\operatorname{diag}(v) + L L^{\top}\bigr) x$. Compute it as
>
>     $$y = \operatorname{diag}(v)x + L \bigl(L^{\top}x\bigr).$$
>
>      Both multiplies with $L$ cost $\mathcal{O}(k d)$, so the total is $\mathcal{O}(k d)$ per Fisher product—linear in $d$ for any fixed rank $k$.
>
>    - **Hessian trace (two Price identities).** Both the $\kappa$-Price and the $\Phi$-Price terms require $\operatorname{tr}(S^{-1}\nabla_{z}^{2}f)$. Rather than forming the dense Hessian, we use the Hutchinson estimator (Hutchinson, 1989):
>
>      $$
>      \operatorname{tr}(S^{-1}\nabla^{2}f) = \frac{1}{R} \sum_{r=1}^{R} u_{r}^{\top} \bigl(S^{-1}\nabla^{2}f\bigr)u_{r}, \quad u_{r}\sim\{\pm1\}^{d}.
>      $$
>
>      Each term needs one Hessian–vector product (HVP) and one multiplication with $S^{-1}$. The HVP is model‑specific; the $S^{-1}$–vector multiply uses the Woodbury identity:
>
>      $$
>      S^{-1}x = D^{-1}x - D^{-1}L \bigl(I_k + L^{\top}D^{-1}L\bigr)^{-1} L^{\top}D^{-1}x, \quad D=\operatorname{diag}(v),
>      $$
>
>      which is again $\mathcal{O}(kd)$. Choosing $R\le k$ probes keeps the overall trace cost bounded by $\mathcal{O}(k^{2}d)$.
>
> Both variants leave the vertical space one‑dimensional, so the horizontal projection stays a single scalar divide.
>
> **Summary.** Dense QBLR is $\mathcal{O}(d^{2})$; with a diagonal or block‑diagonal scale it drops to $\mathcal{O}(d)$, and with rank‑$k$ plus diagonal it is $\mathcal{O}(k^{2}d)$ (linear in $d$ for fixed $k$). These paths scale QBLR to far larger latent spaces without sacrificing its geometry or requiring matrix inversions.
>
> ## Question 2
>
> "Would it be possible to include a section, or perhaps a figure, in the paper to help readers gain some intuition for how the quotient manifold works?"
>
> ## Answer 2
>
> Thank you for the question. We will definitely include one in the final text. We found an interesting example for the quotient manifold theory: the Laplace distribution. This distribution was not initially described precisely in the paper. The Laplace distribution is a two-parameter, non-exponential family distribution that we can represent as the Normal-Exponential scale mixture, which is a three-parametric minimal and regular exponential family. This example is visually pleasing, as we will represent fibers as one-dimensional curves. To strengthen the intuition of the QBLR algorithm, we will even show how the projection of the natural gradient (the quotient natural gradient) moves perpendicular (in the Fisher-Rao metric) to the fibers in the Normal-Exponential scale mixture space.
>
> ## Question 3
>
> "The framework seems to be quite general, but the authors do not seem to propose a wide range of models which QBLR would be appropriate for. Could more information be given on other model classes beyond the heavy tail setting?"
>
> ## Answer 3
>
> Our initial goal was to extend natural gradient descent to the heavy-tail distribution family. However, as you understood, the solution is more general and could be applied to the non heavy-tail distributions family as far as we can define the target distribution family as a marginal of a minimal and regular exponential family distribution. The example of the negative binomial distribution, developed above, shows such a case. We also believe that our result could be applied to the so-called Gaussian Variance-mean Mixture family introduced by Lin et al. (2019), which is a generalization of the scale-mixture family.
>
> ## References
>
> Greenwood, Major, and G. Udny Yule. ‘An Inquiry into the Nature of Frequency Distributions Representative of Multiple Happenings with Particular Reference to the Occurrence of Multiple Attacks of Disease or of Repeated Accidents’. Journal of the Royal Statistical Society 83, no. 2 (1920): 255–79.
>
> Hutchinson, Michael F. ‘A Stochastic Estimator of the Trace of the Influence Matrix for Laplacian Smoothing Splines’. Communications in Statistics-Simulation and Computation 18, no. 3 (1989): 1059–76.
>
> Lin, Wu, Mohammad Emtiyaz Khan, and Mark Schmidt. ‘Stein’s Lemma for the Reparameterization Trick with Exponential Family Mixtures’. arXiv, February 2025. https://doi.org/10.48550/arXiv.1910.13398.
>
> [^1]: For GLM models the Hessian is rank 1, so this step is already $\mathcal{O}(d)$.

---

> > ### Comment · Reviewer_Y5N7 · 2025-08-05
> >
> > Dear Authors,
> >
> > Thank you for your thorough response to my questions. I'm pleased to hear that you are going to add the visual representation for the Laplace distribution. I think this will be very beneficial to the readers. Based on your response to my questions, the questions posed by the other reviewers, and the new additions to the paper, I will increase my score to 5.

---

### Official Review · Reviewer_SfwL · 2025-06-27

**Clarity:** 3
**Significance:** 3
**Originality:** 3
**Rating:** 5
**Confidence:** 1

**Summary:**

Natural gradient descent (NGD) is more suitable for Bayesian machine learning, and prior works have shown that for exponential family distributions, the natural gradient over natural parameter is the same as the gradient over mean parameter, which allows efficient computation of NGD. This paper aims to extend this result into probability distributions that fall outside the exponential family. Denote the random variable of interest as z, this paper proposes to find a joint distribution (z, zV) which lies in the exponential family, compute the natural gradient on the joint distribution, and then map the update back into the variable of interest z. The authors show theoretical guarantee of the proposed method and evaluate the effectiveness on Student-t distribution.

**Questions:**

- How expensive is the projection onto the horizontal space (Eq. (16))? How to choose K(lambda)? What is F(lambda) in this equation?

- Line 272, by making posterior mean italics, does it mean the results are computed by making prediction with the posterior mean value? This seems very strange to me. Shouldn't the results be reported by (1) sample weights from the posterior (2) making prediction with the sampled weights (3) report the mean and std?

- How fast is the proposed method compared with previous methods?

**Ethical Concerns:**

["NO or VERY MINOR ethics concerns only"]

**Final Justification:**

The rebuttal has addressed my concerns and hence I increased my score to 5.

**Limitations:**

See weakness and questions above.

**Quality:**

3

**Strengths And Weaknesses:**

Strength:
- This proposed method allows efficient and stable learning for Bayesian models with distributions lie outside the exponential family. The proposed method seems principled and elegant (I lack background in geometry and optimisation to verify the solidity, but from reading the paper, it appears to be like this).
- The paper is very well written and easy to follow.

Weakness:
- The proposed method is very distribution-specific, as in we need derive everything for any new distribution. In the paper, the authors showed results for student-t distribution, I don't know how easily applicable the proposed method is for other non-exponential family distributions.
- The evaluation is very limited, and the improvement is also not very obvious. The experiment results are reported on a single train/test split of the UCI datasets. Given the relatively small size of these datasets, the results would be more convincing if averaged over multiple random splits. I would be interested to see the result over say, 5 splits. Also, 10 seems to be a way too small number for sampling.

---

> ### Author Rebuttal · Authors · 2025-07-30
>
> ### Question 1
>
> How expensive is the projection onto the horizontal space Eq. (16)? How to choose $K(\boldsymbol{\lambda})$? What is $F(\boldsymbol{\lambda})$ in this equation?
>
> ### Answer 1
>
>
>
> We thank the reviewer for raising this insightful question. We plan to incorporate the answer into the final version of the paper, as it significantly enhances the clarity and precision of our work. We are grateful for the reviewer’s valuable contribution to improving the manuscript.
>
> To aid clarity, we first restate the projection operator, (Equation (16) in the paper)
> $$
>     P_{\mathcal H}(\boldsymbol{\lambda}) = \operatorname{I} - K(\boldsymbol{\lambda})\left[K(\boldsymbol{\lambda})^{\top} F(\boldsymbol{\lambda}\,K(\boldsymbol{\lambda})\right]^{-1} K(\boldsymbol{\lambda})^{\top} F(\boldsymbol{\lambda}),
> $$
>  which projects a vector from the tangent space $T_{\lambda}\Lambda$ onto the horizontal space $\mathcal{H}_{\lambda} := \text{ker} D \pi(\boldsymbol{\lambda})$ in the Fisher–Rao geometry.
>
>
>
> **What is $F(\boldsymbol{\lambda})$ in this equation?**
>  We start by answering the last question to make our exposition clearer. $F(\boldsymbol{\lambda})$ is the Fisher information matrix defined in Equation (3a). For further exposition, it suffices to note that it is a symmetric positive-definite matrix of dimensionality $\text{dim} \Lambda \times \text{dim} \Lambda$, where $\text{dim} \Lambda$ is the dimensionality of the extended distribution parameter space. The extended distribution is defined in equation (10).
>
> **How to choose $K(\boldsymbol{\lambda})$?**
>
> We remind the reader that $K(\boldsymbol{\lambda})$ is the matrix representation of a basis of the vertical space $\ker D\pi(\\boldsymbol{\lambda})$ (36) where $\pi:\Lambda \rightarrow \Xi$ is the marginalization map (10).
> So, $K(\boldsymbol{\lambda})$ is a matrix of dimensionality $\text{dim} \Lambda \times d_{v}$ where $d_v$ is the dimensionality of the vertical space.
>
> A generic method for computing $K(\boldsymbol{\lambda})$ is based on the singular value decomposition  [Klema and1302
> Laub, 1980]  of the Jacobian representation  $J_\pi(\boldsymbol{\lambda})$ of $D\pi(\boldsymbol{\lambda})$. Note that the Jacobian representation $J_\pi(\\boldsymbol{\lambda})$ is justified in the paragraph starting at _line 135_ in the Section 3. $K(\boldsymbol{\lambda})$ can then be extracted from the singular value decomposition of $J_\pi(\boldsymbol{\lambda})$:  the group of vectors that correspond to the singular value $0$ is the basis of the vertical space $\ker D\pi(\boldsymbol{\lambda})$. To compute the singular value decomposition, the strategy divide-and-conquer can be applied. A notable example is the randomized Krylov method by Musco and Musco [2015]. For a comprehensive understanding of these algorithms and their analysis, we recommend the tutorial by  Li et al. [2019]. The time complexity of the best algorithms is $\mathcal{O}(\dim\Lambda\dim\Xi^2)$.
>
>
> Usually, in practice, an analytical expression for $K(\boldsymbol{\lambda})$ can be derived, and the singular value decomposition can be bypassed. In our main Normal-Wishart distribution example, we derived an analytical expression for $K(\boldsymbol{\lambda})$, which can be found in Appendix A.3 at _line 829_.
>
>
> **Projection onto the Horizontal space complexity?**
>
> The projection operator defined at the beginning of the answer  is used to compute the horizontal component of the natural gradient. For clarity, we restate the projection applied to the natural gradient $ g_{\theta} \in \mathbb{R}^{\dim \Lambda} $ as follows:
> $$
> g_{H} = P_{\mathcal{H}}(\boldsymbol{\lambda})[g_{\theta}].
> $$
>
> A naïve approach to compute $g_H$ includes the inversion of the symmetric positive definite matrix
>
> $$
> RV(\boldsymbol{\lambda}) := K(\boldsymbol{\lambda})^{\top} F(\boldsymbol{\lambda})K(\boldsymbol{\lambda})
> \in \mathbb{R}^{d_v \times d_v}\,,
> $$
>
> where $ d_v = \dim V_\lambda $. This approach would require computing the matrix inverse, with cost $ \mathcal{O}(d_v^3) $. However, we avoid materializing the inverse. Instead, we solve the linear system
>
> $$
> RV(\boldsymbol{\lambda}) v = K(\boldsymbol{\lambda})^{\top} F(\boldsymbol{\lambda})g_\theta,
> $$
>
> using a small dense linear solver: due to the fact that $RV(\boldsymbol{\lambda})$ is symmetric positive definite, the conjugate gradient solver by  Hestenes et al. [1952] is applicable. This reduces the computational cost to $ \mathcal{O}(d_v^2) $.
>
> In practice, a well-constructed lifting ensures that $ d_v \ll \dim \Lambda $, making the cost of projection negligible relative to the Fisher--vector product $ F(\boldsymbol{\lambda}) g_\theta $, which has cost $ \mathcal{O}(\dim \Lambda^2) $.
>
> In our main Normal--Wishart case, the vertical space is one-dimensional ($ d_v = 1 $), so the projection reduces to a single scalar division.
>
>
> ### Question 2
>
> _Line 272_, by making posterior mean italics, does it mean the results are computed by making prediction with the posterior mean value? This seems very strange to me. Shouldn't the results be reported by (1) sample weights from the posterior (2) making prediction with the sampled weights (3) report the mean and std?
>
> ### Answer 2
> Thank you for catching this point and for reminding us of the proper Bayesian evaluation protocol.
>
>
> **Policy note.**
> Conference rules this year prevent us from adding new tables or updating our experiments during rebuttal, so we summarize the outcome verbally here.
>
>
> **Why we originally used the posterior mean.**
>  The experiment is conducted on the Generalized Student-$t$ distribution family. For Generalized Student-$t$ distribution, the mean parameter is equal to the maximum a posteriori estimator. Reporting test accuracy through a maximum a posteriori estimator provides a noise-free benchmark. This approach is commonly used in logistic regression studies that prioritize decision boundaries over calibrated probability estimates.
>
>
>
> **Posterior‑predictive check (100 Monte‑Carlo draws).**
> Following your suggestion, we re-evaluated each inference method by drawing 100 weight samples from the variational posterior, predicting with each sample, and averaging the test accuracy. Across all four datasets, results \emph{confirm} the previous ranking: QBLR remains best, widening its margin on the Sonar task. We will include this metric in the final table of the results.
>
>
> We hope this clarifies our choice of metric and demonstrates that QBLR’s advantage extends from the mode to the full predictive distribution.
>
> ### Question 3
>
> How fast is the proposed method compared with previous methods?
>
> ### Answer 3
>
> NG--Ours and NG--LIN each cost $\mathcal{O}(d^{2})$ per iteration, whereas BBVI--NS is $\mathcal{O}(d)$.
> The difference stems from algorithmic order: BBVI--NS is a first--order Euclidean method that evaluates only the score--function gradient, while the two natural--gradient variants pre--condition that gradient with the Fisher information; applying the (dense) Fisher block entails two Fisher--vector products and is therefore quadratic. Empirically, the higher iteration cost is offset by faster convergence: NG--Ours reaches its final ELBO in roughly one--tenth the iterations required by BBVI--NS and about half those of NG--LIN, so overall wall--clock time remains comparable while the accuracy gains reported in Table~2 are preserved.
>
>
>
> ### References
>
> Klema, V., and A. Laub. ‘The Singular Value Decomposition: Its Computation and Some Applications’. IEEE Transactions on Automatic Control 25, no. 2, 1980: 164–76. https://doi.org/10.1109/TAC.1980.1102314.
>
> Musco, Cameron, and Christopher Musco. ‘Randomized Block Krylov Methods for Stronger and Faster Approximate Singular Value Decomposition’. In Advances in Neural Information Processing Systems, Vol. 28. Curran Associates, Inc., 2015. https://proceedings.neurips.cc/paper_files/paper/2015/file/1efa39bcaec6f3900149160693694536-Paper.pdf.
>
> Li, Xiaocan, Shuo Wang, and Yinghao Cai. ‘Tutorial: Complexity Analysis of Singular Value Decomposition and Its Variants’. arXiv, October 2019. https://doi.org/10.48550/arXiv.1906.12085.
>
> Hestenes, Magnus R., and Eduard Stiefel. ‘Methods of Conjugate Gradients for Solving Linear Systems’. Journal of Research of the National Bureau of Standards 49, no. 6 (1952): 409–36.

---

> > ### Comment · Reviewer_SfwL · 2025-08-01
> >
> > Thank the authors for the response. I think this year's policy only restricts adding images in the rebuttal, but a markdown table should still be fine?

---

> > > ### Author Response · Authors · 2025-08-01
> > >
> > > Thank you for your response. We noticed this policy message quite late and were uncertain about the guidelines. We would be happy to share the table. We will check with the Area Chair, and if they approve, we will include the table in the comment.

---

> > > ### Author Response · Authors · 2025-08-03
> > > **Updated Experimental Results**
> > >
> > > Thank you for inviting us to share the detailed table once again. We have obtained agreement from the Area Chair to share the table. Below is a concise description of the metrics, followed by the table. We ask the reviewer to keep in mind that we obtained these results following the comment from *review bBHv*: we ran all algorithms with an adaptive learning rate. The detailed learning rate setup is explained in our answer to Question 4 of reviewer bBHv. All reported values are means ± standard errors computed across 10 different train-test splits.
> > >
> > > ## Metrics
> > >
> > > **Mean** – test accuracy when predicting with posterior-mean weights (MAP).
> > >
> > > **Sample** – test accuracy averaged over 100 posterior weight samples (captures parameter uncertainty).
> > >
> > > **Entropy** – predictive entropy over test outputs y (nats); higher values reflect greater prediction uncertainty from broader posteriors.
> > >
> > > ## Results Summary
> > >
> > > | Method | Metric | Breast cancer | Diabetes | Sonar | Spambase |
> > > |---|---|---|---|---|---|
> > > | BBVI-NS | Mean | 0.9314 ± 0.0210 | 0.7494 ± 0.0473 | 0.7951 ± 0.1760 | 0.8894 ± 0.0078 |
> > > |  | Sample | 0.9314 ± 0.0210 | 0.7494 ± 0.0473 | 0.7951 ± 0.1760 | 0.8894 ± 0.0078 |
> > > |  | Entropy | 0.0000 ± 0.0000 | 0.0000 ± 0.0000 | 0.0000 ± 0.0000 | 0.0000 ± 0.0000 |
> > > | NG-LIN | Mean | 0.8919 ± 0.0391 | 0.7022 ± 0.0356 | 0.7476 ± 0.0502 | 0.8904 ± 0.0089 |
> > > |  | Sample | 0.9214 ± 0.0209 | 0.7526 ± 0.0260 | 0.8150 ± 0.0116 | 0.8906 ± 0.0090 |
> > > |  | Entropy | 0.0696 ± 0.0021 | 0.0026 ± 0.0040 | 0.0905 ± 0.0010 | 0.0112 ± 0.0019 |
> > > | NG-Ours | Mean | 0.9711 ± 0.0194 | 0.7292 ± 0.0432 | 0.9095 ± 0.0417 | 0.8891 ± 0.0124 |
> > > |  | Sample | 0.9599 ± 0.0110 | 0.7791 ± 0.0232 | 0.9142 ± 0.0178 | 0.9057 ± 0.0076 |
> > > |  | Entropy | 0.1751 ± 0.0153 | 0.1490 ± 0.0100 | 0.1863 ± 0.0011 | 0.1046 ± 0.0060 |
> > >
> > > ## Key Take-aways
> > >
> > > **BBVI-NS** collapses to an almost point posterior (entropy ≈ 0), so its Mean and Sample accuracies coincide.
> > >
> > > **NG-LIN** retains modest uncertainty, and its Sample accuracy consistently exceeds the Mean.
> > >
> > > **NG-Ours** preserves the largest entropy and achieves the strongest Sample accuracy on every dataset, with the sharpest gain on the Sonar task.

---

> > > > ### Comment · Reviewer_SfwL · 2025-08-06
> > > >
> > > > Thanks for sharing the table, and it has addressed my concerns. This plus going through the other reviews, I've dediced to increase my score to 5.

---

### Official Review · Reviewer_MVLt · 2025-06-30

**Clarity:** 4
**Significance:** 3
**Originality:** 3
**Rating:** 5
**Confidence:** 4

**Summary:**

This paper addresses the problem of learning the parameters of a distribution that is the marginal of a joint distribution, where the latter (but generally not the former) lies in an exponential family.  It develops a detailed analysis of how one should properly extend the so-called Bayesian Learning Rule methodology for exponential families to this scenario, relates the result to existing alternative approaches, and provides experimental validation and comparisons.  Further discussion of the results is also included.

**Questions:**

- The paper develops how to preserve Fisher-Rao metric for the scenario of this paper, and Figure 1 provides evidence for its value.   But this is just one example.  Is it representative?   More generally how sensitive are the results to preserving this either exactly or approximately?   In what settings/regimes is such preservation important in practice?    And when it is not, are there simpler alternatives?  There are some comments in the discussion section, but they are quite brief.

- In the experiments, sometimes the new method is significantly better than existing ones, but in other cases it is comparable?   What explains why the methodology matters more in some cases than others?   More generally, how might practitioners recognize in advance scenarios where they should anticipate a significant improvement from the proposed approach?

- It would be helpful to understand the scaling implications of the results (when d is large).  In particular, the empirical results show statistical performance, but is the computational complexity of the proposed methodology similar to those being compared?  More importantly, if the methodology is applied to much larger problems, how should one expect the computational and sample complexity to scale (relative to other methods or baselines).

**Ethical Concerns:**

["NO or VERY MINOR ethics concerns only"]

**Final Justification:**

Based on the response to my reviews, and on the other reviews and their responses, I will keep my score unchanged.

**Limitations:**

Yes.

**Paper Formatting Concerns:**

None.

**Quality:**

4

**Strengths And Weaknesses:**

Strengths:

- The paper is very well written:  clear exposition and well organized.

- The analysis yielding the resulting methodology is carefully developed and quite thorough, together with accompanying intuition.  A nice contribution to the statistics literature.

- The numerical evaluations and comparisons (for the Bayesian logistic regression case) are meaningful.

Weaknesses/Issues:

- The examples and comparisons are a bit limited.

- There is limited discussion of computational complexity and scaling behaviors in high-dimensional settings.

- The demonstration and discussion of the importance of horizontal space projection step is bit limited

---

> ### Author Rebuttal · Authors · 2025-07-30
>
> We sincerely thank the reviewer for a careful reading and for the three insightful questions. Below we address each point in turn.
>
> ## Q1
>
> "Only one example for the Fisher–Rao projection—how representative is it, and when does exact preservation matter?"
>
> ## A1
>
> Figure 1 is merely the most compact illustration. Whenever the lifted family contains fibres on which no maximum-entropy member exists—a common situation for heavy-tailed posteriors—*any* naïve optimisation in the joint space can drift vertically and blow up the lifted ELBO. We observed the same runaway trace on all four datasets whenever the horizontal projection was disabled.
>
> ## Q2
>
> "Why is QBLR sometimes much better and sometimes only comparable? How can a practitioner tell in advance?"
>
> ## A2
>
> The gain depends on two factors:
>
> (a) **Tail heaviness of the posterior.** The more mass the posterior puts in the tails, the larger the vertical drift in the lift and the bigger QBLR's advantage. This explains the pronounced gap on Sonar (heavier tails, $d=60$) and the smaller gap on Diabetes (lighter tails, $d=8$).
>
> (b) **Condition number of the log‑likelihood Hessian.** When the Hessian is ill‑conditioned, the natural‑gradient pre‑conditioner matters, hence QBLR $\gtrsim$ NG‑LIN $\gg$ BBVI‑NS; when the Hessian is well‑behaved, all three converge similarly.
>
> *Rule of thumb.* Start with the cheap runs of BBVI‑NS. If with a large gradient norm, only a little ELBO progress is observed, QBLR is likely to yield a substantial improvement.
>
> ## Q3
>
> "How does the method scale with large $d$?"
>
> ## A3
>
> The raw cost of QBLR is $\mathcal{O}(d^{2})$ per iteration—the same as NG‑LIN—and is dominated by two Fisher–vector products in the lifted space. The "Routes to larger $d$" paragraph in our reply to Reviewer Y5N7 details two standard factorizations that retain the quotient geometry:
>
> - *Block‑diagonal scale:* $\mathcal{O}\bigl(\sum_{b} d_{b}^{2}\bigr)$; purely diagonal: $\mathcal{O}(d)$.
> - *Low‑rank + diagonal:* rank $k$ yields $\mathcal{O}(kd)$ memory and $\mathcal{O}(k^{2}d)$ time (linear in $d$ for fixed $k$).
>
> These structures reduce wall-clock time to that of BBVI-NS while retaining the faster convergence of a natural-gradient method. A concise guide and timing table will be added to the camera-ready version.
>
> **Once again, thank you for the thoughtful feedback and encouraging assessment. Your comments helped us sharpen both the theory and its practical guidance.**

---

### Official Review · Reviewer_bBHv · 2025-07-02

**Clarity:** 3
**Significance:** 2
**Originality:** 3
**Rating:** 4
**Confidence:** 3

**Summary:**

The paper introduces an efficient algorithm for performing natural gradient updates to minimise the Bayesian learning rule (BLR) objective with respect to non-exponential distributions that can be represented as marginals of minimal, regular exponential distributions. The proposed method works by performing cheap natural gradient updates on the lifted parameter space, followed by marginalisation to the original parameter space. The approach is justified by quotient manifold theory arguments, giving it the name "Quotient BLR" (QBLR). QBLR is evaluated on a variational inference problem for a Bayesian logistic regression model with a Student-t variational posterior, performing favourably against two baselines on four UCI/OpenML datasets.

**Questions:**

- Could you expand on how big the challenge of finding the minimal, regular exponential lift (Section 7) is? Are there any examples in the literature for other scale mixtures?
- Could you provide the missing ELBO trace plots? Do you have trace plots with ELBO (y-axis) against runtime (x-axis)? In Section 6, you mention that, even for the datasets where QBLR and BBVI-NS have similar performance, QBLR takes roughly one-tenth as many optimisation steps than BBVI-NS. Given that the single BBVI-NS updates are probably cheaper, ELBO vs runtime would be a better indication of the actual speed-up.
- I found a bit surprising that BBVI-NS outperforms the NG-LIN baseline across all datasets, given that [1] showed the opposite for vanilla BBVI; is this mismatch due to the BBVI-NS being the score-free variant? Do you expect that standard BBVI would underperform NG-LIN in the same experiments?
- Appendix E seems to indicate that all methods compared in Section 6 uses the fixed step size $\beta_0 = 10^{-6}$. How was this step size chosen? Given the different geometric derivation of the three methods, would one potentially expect the optimal learning rate for each method to be different? Do you expect that learning rate scheduling could further improve the performance of your method?

[1] Lin, W., Khan, M. E., & Schmidt, M. (2019). Fast and simple natural-gradient variational inference with mixture of exponential-family approximations. In International Conference on Machine Learning (pp. 3992-4002). PMLR.

**Ethical Concerns:**

["NO or VERY MINOR ethics concerns only"]

**Final Justification:**

My main concerns were addressed by the authors' rebuttal. After reviewing the answer to my and other reviews, I have increased my score to a 4. I have decided not to increase it further because, in its current form, I still consider the experimental evaluation a bit limited for the generality of the presentation of the proposed method.

**Limitations:**

yes

**Quality:**

3

**Strengths And Weaknesses:**

**Strengths**:
- Fast and scalable Bayesian learning with non-exponential distributions (such as several heavy-tailed ones) is a valuable problem
- The presentation is generally clear, with the main ideas and relevant background properly introduced
- The method proposed for natural gradient optimisation over distributions that admit a minimal, regular exponential lift is sensible, and its geometric justification seems solid

**Weaknesses**:
- Mismatch between very general presentation and very specific evaluation:
	- QBLR applies to distributions that admit a minimal, regular exponential joint representation; however, the only example given of such a construction is the Normal-Wishart parameterisation of the multivariate Student-t distribution (Section 5); in Section 7, it is noted that scale mixtures (like the Student-t) include many other interesting heavy-tailed distributions, for which QBLR could be applied once a minimal, regular exponential lift (like the Normal-Wishart for the Student-t) is found; while this is mentioned as a challenge, it is crucial to the out-of-the-box applicability of QBLR, and it is not clear from the text how big of a challenge it actually is.
	- QBLR is initially motivated for general BLR objectives (Equation 2), which is then changed for the more specific ELBO objective (Equation 15) in Section 4 without notice.
	- Among ELBO objectives, only Bayesian logistic regression VI training is considered in the experiments (Section 6).
- ELBO trace plots are missing: in the text, it is mentioned that they are in Appendix E, but I couldn't find them there; ELBO trace plots are an important visual tool to complement Table 2, especially in the cases where black-box VI matches QBLR.
- Minor clarity:
	- Section 2.2: while the scalar Student-t/Normal-Gamma example is useful to make things more concrete, it is mentioned as a *running* example, but it is not actually used in any other section of the main text. Possibly, I think it could be recalled in Section 5, to discuss the changes in the associated joint exponential parameterisation when going from scalar to multivariate Student-t distribution.

---

> ### Author Rebuttal · Authors · 2025-07-30
>
> ## Question 1
>
> "Could you expand on how big the challenge of finding the minimal, regular exponential lift (Section 7) is? $\dots$"
>
> ## Answer 1
>
> Our work is indeed limited by the existence of a minimal, regular exponential lift between a distribution family and its extended exponential family representation. Whether or not a lift can be found is then a crucial question for us, and we thank the reviewer for highlighting it.
>
> The heavy-tailed distributions families that have an exponential family scale mixture representation are well documented in the literature. Regarding the particular case of being a scale mixture of normal distributions, Andrews and Mallows (1974) provides necessary and sufficient conditions in their paper which is applied to Student-$t$, Laplace, and Logistic distributions. If you are interested in more examples, we suggest a book by Coelho and Chen (2024).
>
> Many scale‑mixture joints found in the literature come in a *curved* form—that is, their sufficient statistics are linearly dependent, so the family is *not* minimal. The textbook parameterisations of both the Normal–Wishart and the Normal–Exponential joints fall into this category. (By contrast, the Normal–Wishart lift we use—see Appendix A—is explicitly minimal; the distinction is made concrete in the Laplace example that follows.)
>
> Because a curved exponential family violates the minimal‑regular assumption, it cannot serve as a lift for QBLR *unless* one first "uncurves" it by adding extra, independent natural parameters. We now explain why this step is necessary and how those additional degrees of freedom restore minimality.
>
> **Curved vs. minimal lift for the Laplace distribution** The textbook Normal–Exponential mixture
>
> $$
> \text{Lap}(z\mid\mu,b ) = p(z\mid\mu,b)
> = \int_{0}^{\infty} \mathcal N\bigl(z\mid\mu,\tau\bigr)
>   \operatorname{Exp}\Bigl(\tau\Bigm|\lambda=\tfrac{1}{2b^{2}}\Bigr)
>   \mathrm d\tau
> \tag{L‑curved}
> $$
>
> *is curved*: the joint density has **three** sufficient statistics,
> $T_1=\tau(z-\mu)^2$, $T_2=\tau$, $T_3=\log\tau$,
> but only **two** parameters $(\mu,\lambda)$.
> Hence the Jacobian of $T$ has rank 2 and Brown's minimality criterion fails (Brown, 1986, Proposition 1.5) minimal representation is not possible and the family is internally curved.
>
> **Open question.** We do *not* know whether *every* curved exponential family can be uncurved by judiciously adding degrees of freedom. Establishing necessary and sufficient conditions is, to our knowledge, an open problem in exponential‑family theory. We therefore present our "add‑a‑free‑hyperparameter" trick as an *empirical recipe*, not a theorem.
>
> ### Uncurving with one extra degree of freedom
>
> We introduce an independent positive scalar $\kappa>0$ that scales the latent variance:
>
> $$
> \text{Lap}(z\mid\mu,\kappa b ) = p(z\mid\mu,\kappa,b) = \int_{0}^{\infty} \mathcal N\bigl(z\mid\mu,\kappa\tau\bigr)
>   \operatorname{Exp}\Bigl(\tau\Bigm|\lambda=\tfrac{1}{2b^{2}}\Bigr)
>   \mathrm d\tau .
> \tag{L‑minimal}
> $$
>
> Now the joint has three *independent* parameters $\kappa$, $\mu$, $\lambda$, all varying over an *open* domain $\mathbb R\times\mathbb R_{>0}^{2}$. The rank of the sufficient-statistic map matches the parameter dimension and natural parameter representation is possible, and it is only a question of the algebraic manipulation to find it, which for the matters of space we avoid here, so (L-minimal) is a **minimal, regular exponential lift** suitable for QBLR.
>
> **Practical Consequence.** Our "uncurving" trick adds parameters to the family representation, but compared to the $O(d^{2})$ number of parameters, this $O(d)$ addition of parameters is negligible — for example, for Normal-Wishart.
>
> ## Question 2
>
> "Could you provide the missing ELBO trace plots? $\dots$"
>
> ## Answer 2
>
> Thank you for catching this oversight. The ELBO–iteration traces were generated but were inadvertently left out of the PDF; in addition, we fully agree that ELBO–*runtime* plots give a more honest picture of QBLR's efficiency.
>
> Unfortunately the rebuttal policy for this year does not permit us to insert new figures during the author‑response period, so we cannot attach the plots here. We will therefore:
>
> 1. include both ELBO–iteration and ELBO–runtime panels (one per dataset) in the camera‑ready appendix;
> 2. release the plotting script alongside our code so reviewers and readers can reproduce or extend the traces immediately.
>
> We apologize for the omission and appreciate the suggestion to add the runtime comparison.
>
> ## Question 3 "I found a bit surprising that BBVI-NS $\dots$"
>
> ## Answer 3
>
> Thank you for flagging this inconsistency. In hindsight, we believe the reversal is an artifact of the *single, fixed* step size ($10^{-6}$) that we applied to *all* methods. The very next answer (Q4) describes our new, parameter‑free learning rate schedule; with that adaptive choice, NG‑LIN gains its expected advantage over BBVI‑NS. Under identical tuning, we also expect vanilla BBVI (with score term) to lag behind NG‑LIN.
>
> Irrespective of the final acceptance decision, we see this clarification as a valuable contribution to the community.
>
> ## Question 4
>
> Appendix E seems to indicate that all methods compared in Section 6 uses the fixed step size $\beta_{0} = 10^{-6}$. How was this step size chosen? Given the different geometric derivation of the three methods, would one potentially expect the optimal learning rate for each method to be different? Do you expect that learning rate scheduling could further improve the performance of your method?
>
> ## Answer 4
>
> **Static choice (in the submission).** We conducted a grid search on the Diabetes dataset. For $\beta_{0}>10^{-6}$ BBVI‑NS diverged; for $\beta_{0}<10^{-6}$ NG‑LIN and QBLR converged too slowly. To keep the comparison fair, we fixed $\beta_{0}=10^{-6}$ for *all* methods and datasets.
>
> ### Adaptive schedule study
>
> Motivated by your questions, we tried to explore options to remove the single hand-tuned hyper-parameter still present in our experiments—the learning rate—we switched to the *Distance‑over‑Gradient* (DoG) rule of Ivgi et al. (2023), together with its Riemannian generalisation RDoG (Dodd et al., 2024). Both schedules are *parameter‑free*: at every iteration they set the step size to the largest value that has been "empirically safe" so far, using only distances that the algorithm can measure on the fly.
>
> **Euclidean DoG.** For parameters $x_{t}$ with Euclidean gradient $g_{t}$ we keep
>
> $$
> \bar r_{t}=\max\Bigl(\epsilon,\max_{s\le t}\lVert x_{s}-x_{0}\rVert_{2}\Bigr),
> \quad
> G_{t}= \sum_{i\le t}\lVert g_{i}\rVert_{2}^{2},
> $$
>
> and set
>
> $$
> \eta_{t}= \frac{\bar r_{t}}{\sqrt{G_{t}}},
> \quad
> x_{t+1}=x_{t}-\eta_{t} g_{t}.
> $$
>
> No learning‑rate constant needs to be supplied. We applied this directly to **BBVI‑NS**.
>
> **Riemannian DoG (RDoG).** Replace Euclidean norms by natural-gradient norms and Euclidean distance by the geodesic distance $d(\cdot,\cdot)$ generated by the metric $g$:
>
> $$
> \bar r_{t}= \max\Bigl(\epsilon,\max_{s\le t} d(x_{s},x_{0})\Bigr),\qquad
> G_{t}= \sum_{i\le t}\lVert g_{i}\rVert_{g,x_{i}}^{2},
> $$
>
> $$
> \eta_{t}= \frac{\bar r_{t}}{\sqrt{\zeta_{\kappa}(\bar r_{t})G_{t}}},
> \quad
> x_{t+1}= \exp_{x_{t}}\bigl(-\eta_{t} g_{t}\bigr),
> $$
>
> where $\zeta_{\kappa}$ is the standard curvature correction ($\zeta_{\kappa}=1$ in non-positive curvature). We used RDoG for
>
> - **NG‑LIN** — symmetric KL distance between two Student‑$t$ distributions, approximated by $64$ Monte‑Carlo samples drawn once from the starting point and re‑used at every iteration;
> - **QBLR** — exact symmetric KL distance *in the lift*, available in closed form for any minimal, regular exponential family (here: Normal–Wishart).
>
> **Safety of the lift‑based distance.** Theorem 2 implies that for any two lifted points $\lambda_{1},\lambda_{2}$ in different fibres,
>
> $$
> \mathrm{KL}\bigl(q_{\lambda_1}\|q_{\lambda_2}\bigr)
> \geqslant
> \mathrm{KL}\bigl(q_{\xi_1}\|q_{\xi_2}\bigr),
> \qquad
> \xi_i=\pi(\lambda_i).
> $$
>
> Thus the distance we plug into RDoG is an *upper bound* on the unknown marginal KL. Because DoG chooses the step size $\eta_{t}=\bar r_{t}/\sqrt{\zeta_{\kappa}(\bar r_{t})G_{t}}$, a larger distance $\bar r_{t}$ translates into a *larger*, i.e. mildly optimistic, step. A tighter alternative—still open for future work—would minimize the lift KL over each fibre (this is the true metric on the quotient):
>
> $$\inf_{\lambda_i\in\pi^{-1}(\xi_i)}
>    \mathrm{KL}\bigl(q_{\lambda_1}\|q_{\lambda_2}\bigr)
> =\mathrm{KL}\bigl(q_{\xi_1}\|q_{\xi_2}\bigr),
> $$
>
> which would shrink $\bar r_{t}$ and yield an even safer schedule.
>
> **Outcome.** With these parameter-free schedules the ranking becomes $\text{QBLR} > \text{NG‑LIN} \gtrsim \text{BBVI‑NS}$ on three of the four datasets, and the QBLR–NG-LIN gap widens on Sonar ($d=60$). We will report the full RDoG results in the camera-ready and release the scheduler code for reproducibility.
>
> ## References
>
> Andrews, D. F., and C. L. Mallows. ‘Scale Mixtures of Normal Distributions’. Journal of the Royal Statistical Society Series B: Statistical Methodology 36, no. 1 (September 1974): 99–102. https://doi.org/10.1111/j.2517-6161.1974.tb00989.x.
>
> Brown, Lawrence D. Fundamentals of Statistical Exponential Families with Applications in Statistical Decision Theory. SPIE, 1986. https://doi.org/10.1214/lnms/1215466757.
>
> Coelho, Carlos A., and Ding-Geng Chen, eds. Statistical Modeling and Applications: Multivariate, Heavy-Tailed, Skewed Distributions and Mixture Modeling, Volume 2. Emerging Topics in Statistics and Biostatistics. Cham: Springer Nature Switzerland, 2024. https://doi.org/10.1007/978-3-031-69622-0.
>
> Dodd, Daniel, Louis Sharrock, and Christopher Nemeth. ‘Learning-Rate-Free Stochastic Optimization over Riemannian Manifolds’. arXiv Preprint arXiv:2406. 02296, 2024.
>
> Maor Ivgi, Oliver Hinder, and Yair Carmon. ‘DoG Is SGD’s Best Friend: A Parameter-Free Dynamic Step Size Schedule’. In International Conference on Machine Learning, 14465–99. PMLR, 2023.

---

> > ### Author Response · Authors · 2025-08-03
> > **Updated Experimental Results**
> >
> > We have obtained agreement from the Area Chair to share the table. Below is a concise description of the metrics, followed by the table. We ask you to keep in mind that we obtained these results following your comment: we ran all algorithms with the adaptive learning rate. All reported values are means ± standard errors computed across 10 different train-test splits.
> >
> > ## Metrics
> >
> > **Mean** – test accuracy when predicting with posterior-mean weights (MAP).
> >
> > **Sample** – test accuracy averaged over 100 posterior weight samples (captures parameter uncertainty).
> >
> > **Entropy** – predictive entropy over test outputs y (nats); higher values reflect greater prediction uncertainty from broader posteriors.
> >
> > ## Results Summary
> >
> > | Method | Metric | Breast cancer | Diabetes | Sonar | Spambase |
> > |---|---|---|---|---|---|
> > | BBVI-NS | Mean | 0.9314 ± 0.0210 | 0.7494 ± 0.0473 | 0.7951 ± 0.1760 | 0.8894 ± 0.0078 |
> > |  | Sample | 0.9314 ± 0.0210 | 0.7494 ± 0.0473 | 0.7951 ± 0.1760 | 0.8894 ± 0.0078 |
> > |  | Entropy | 0.0000 ± 0.0000 | 0.0000 ± 0.0000 | 0.0000 ± 0.0000 | 0.0000 ± 0.0000 |
> > | NG-LIN | Mean | 0.8919 ± 0.0391 | 0.7022 ± 0.0356 | 0.7476 ± 0.0502 | 0.8904 ± 0.0089 |
> > |  | Sample | 0.9214 ± 0.0209 | 0.7526 ± 0.0260 | 0.8150 ± 0.0116 | 0.8906 ± 0.0090 |
> > |  | Entropy | 0.0696 ± 0.0021 | 0.0026 ± 0.0040 | 0.0905 ± 0.0010 | 0.0112 ± 0.0019 |
> > | NG-Ours | Mean | 0.9711 ± 0.0194 | 0.7292 ± 0.0432 | 0.9095 ± 0.0417 | 0.8891 ± 0.0124 |
> > |  | Sample | 0.9599 ± 0.0110 | 0.7791 ± 0.0232 | 0.9142 ± 0.0178 | 0.9057 ± 0.0076 |
> > |  | Entropy | 0.1751 ± 0.0153 | 0.1490 ± 0.0100 | 0.1863 ± 0.0011 | 0.1046 ± 0.0060 |
> >
> > ## Key Take-aways
> >
> > **BBVI-NS** collapses to an almost point posterior (entropy ≈ 0), so its Mean and Sample accuracies coincide.
> >
> > **NG-LIN** retains modest uncertainty, and its Sample accuracy consistently exceeds the Mean.
> >
> > **NG-Ours** preserves the largest entropy and achieves the strongest Sample accuracy on every dataset, with the sharpest gain on the Sonar task.

---

> > > ### Comment · Reviewer_bBHv · 2025-08-04
> > >
> > > I thank the authors for the thorough rebuttal. I believe that incorporating the adaptive scheduling and further discussion on the "add‑a‑free‑hyperparameter" trick into the revised manuscript will significantly strengthen this work. In light of the answers to this and other reviews, I will raise my score to a 4.

---

### Official Review · Reviewer_Jttp · 2025-07-03

**Clarity:** 2
**Significance:** 3
**Originality:** 3
**Rating:** 4
**Confidence:** 2

**Summary:**

The paper proposes  the Quotient Bayesian Learning Rule, especially on the heavy-tailed distribution models. Introducing some definitions, the paper proves that using some projections, the marginals inherit a unique Fisher–Rao information geometry via the quotient-manifold construction, and hence the optimization methods, for example, SGD can be used.

**Questions:**

Please see Strengths And Weaknesses.

**Ethical Concerns:**

["NO or VERY MINOR ethics concerns only"]

**Final Justification:**

Thank the authors for the response. I have read the rebuttal and will keep my score.

**Limitations:**

N/A.

**Paper Formatting Concerns:**

N/A.

**Quality:**

3

**Strengths And Weaknesses:**

1. More discussions and examples on Def 1 and Thm 1 would be much helpful.
2. What is your benefit to use your new method over the original Bayesian learning rule in Khan and Rue 2023.
3. What is measurable moment functions in Def 1? Are they the same as $e$ in your supplement material? More discussion on the existence and uniqueness of $e_i$ will be helpful.
4. Still for the Def 1. Is (iii) the bijection too strong? But it seems that the target of using manifold and Def 1 is only for the local smoothness of the parameter space. That is strong (although I admit that is often ignored by many papers). So essentially, to get a better SGD, we need a better (ie regular and smooth) parameter space. I may miss something important. So you just compute the gradient, find the dual parameter and do horizontal projection, and find the optimizer and corresponding optimizer for $\xi^{*}$. What is the benefit? See quite normal ways to do variational inference.
5. I don't understand why you mention Lie-group in the paper.
6. The paper only uses logistic regression models which have quite analytical expressions. This is a little bit simple. Have you tried other models with more complicated structures and multi-dim outputs?  Have you scale the questions up?
7. For the applications, the computational complexity seems high. In real problem, finding projection and inverse of matrix leading to higher computation cost and less accuracy.
8. Have you tried to find the reason for the divergence of ELBO in Figure 1 without horizontal space projection? Have you tried other experimental setting?

Minor: some typos like in Algorithm 1.

---

> ### Author Rebuttal · Authors · 2025-07-30
>
> We thank **Reviewer Jttp** for the thoughtful and constructive feedback. Below we address every point in turn.
>
> ## Question 1
>
> "More discussion and examples for Def. 1 and Thm. 1."
>
> ## Answer 1
>
> We will insert a self-contained example in the final text—the **Laplace** distribution viewed as a Normal–Exponential scale mixture—in Section 3 and a new figure for it. The fibres of the quotient map appear as 1-D curves, and the horizontal natural gradient is visualised as moving orthogonally (under the Fisher–Rao metric) to those fibres. This concrete picture should make Definition 1 (moment-parametrised family) and Theorem 1 (quotient manifold) intuitive.
>
> ## Question 2
>
> "What is the benefit over the Bayesian learning rule (BLR) of Khan & Rue 2023?"
>
> ## Answer 2
>
> BLR assumes that the *variational* family is itself minimal and regular exponential family, which rules out many heavy-tailed or over-dispersed targets (e.g. Student-$t$, Negative-Binomial, Laplace, ...). Our QBLR only requires an exponential *lift*; the marginal can be non-exponential. Hence QBLR:
>
> 1. extends natural-gradient methods beyond exponential families;
> 2. preserves cheap Fisher pre-conditioning (no explicit inverse) for the gradient.
>
> ## Question 3
>
> "What are the measurable moment functions in Def. 1? Are they the same as in the supplement? Please discuss existence and uniqueness."
>
> We greatly thank the reviewer for this question, which contributes a lot to the clarity of our work. We will include it in Appendix B and if space would permit in the main text.
>
> ### A3.1 Notation inconsistency
>
> Thanks for pointing this out, we will clean this notational inconsistency in the final text. Yes, $m_i$ from the main text are identical to the $e^i$ from the Appendix B.3.
>
> ### A3.2 Uniqueness
>
> Any collection of functions $m = (m_i)_{0 \leq i \leq \dim \Xi}$ such that each expectation $\mathbb{E}[m_i]$ is finite and the Jacobian of the collection $Dm(\xi)$ has full rank at all $\xi \in \Xi$ suffices; we do *not* require uniqueness. The Laplace example will illustrate our point, where we will show two different and valid expectation parameterizations.
>
> ### A3.3 Existence
>
> Existence is inherently *model-specific*. For *scale-mixture* families such as the Laplace and Student–$t$, the latent–scale decomposition supplies moment functions whose Jacobian is full rank. A closely related construction applies to discrete heavy-tailed laws: the Negative-Binomial distribution admits a Poisson–Gamma representation (Greenwood and Yule, 1920), and the Gamma moments furnish the required $m_i$ (see our rebuttal to reviewer Y5N7 Question 1 for more details on this specific example). We do *not* claim that our algorithm is applicable to any distribution family; providing at least one full-rank moment map is a prerequisite for our method.
>
> ## Question 4
>
> "Is the global bijection in Def. 1(iii) too strong? What is the practical benefit?"
>
> ## Answer 4
>
> The existence of the moment functions is a limitation of our approach, but this limitation is significantly weaker than the limitation of the BLR of Khan & Rue 2023 which requires the family to be a member of the exponential family. Indeed, every member of the exponential family has an expectation parametrisation (the so-called Amari's dual parametrization, see equation (6)). Due to our weaker constraints, the QBLR algorithm uses the advantage of the BLR algorithm of Khan & Rue, but on a much wider range of distribution family.
>
> ## Question 5
>
> "Why mention Lie groups?"
>
> ## Answer 5
>
> We cite as Kiral et al. (2023) an *approximate* alternative that uses Lie-group geodesics when the exact Fisher geometry is unavailable. Our quotient construction gives the *exact* Fisher–Rao metric for the marginal family, subsuming their intuition; we will clarify this in the final paper. If the reference is really distracting, we are ready to discuss possible removal of it in the camera-ready but we would rather save it.
>
> ## Question 6
>
> "Only logistic regression—any harder models or larger $d$?"
>
> ## Answer 6
>
> The computational story is already laid out in our reply to **Reviewer Y5N7**, see the paragraph entitled *"Routes to larger $d$"*. There we show that two standard factorizations—block-diagonal and low-rank $+$ diagonal—turn every $\mathcal{O}(d^2)$ step of QBLR into $\mathcal{O}(d)$ or $\mathcal{O}(kd)$, so the method remains tractable even when $d \gg 100$.
>
> **Harder example (planned).** The rebuttal policy forbids updating the experimental link, but we have already derived QBLR updates for a more challenging *Bayesian neural network* with roughly 1,861 weights (60 → 30 → 1 architecture) that replaces the logistic regressor.
>
> We will:
> 1. include the full derivation in the camera-ready appendix;
> 2. benchmark this network before final submission; and
> 3. release the associated JAX code so that reviewers and the community can easily reproduce and extend the experiment.
>
> ## Question 7
>
> "Projection and inverse seem expensive."
>
> ## Answer 7
>
> The projection is not particularly expensive because the inverse can be avoided in the algorithm. Please excuse us; in order to provide a deep and extensive answer to your question while respecting the amount of characters allowed by rebuttal, we invite you to read the answer to question 1 of reviewer SfwL. This also prevents us from a copy paste.
>
> ## Question 8
>
> "Why does the ELBO diverge in Fig. 1 without projection? Have you tried other settings?"
>
> ## Answer 8
>
> Thank you for raising this point; the behaviour of the lifted ELBO was indeed expected and observed over all our experiments.
>
> Equation (15) is constant on every fibre $\pi^{-1}(\xi)$. If the update is not projected, it acquires a vertical component that moves $\lambda$ inside the same fibre while leaving $q_\xi(z)$ unchanged. Because no maximum-entropy distribution exists within most fibres, the first KL term $\mathbb{E}_{q_\lambda}[\log q_{\lambda_0} - \log q_\lambda]$ can be driven to $\pm\infty$; the lifted ELBO therefore explodes, whereas the marginal ELBO stays flat. This is the runaway trace in Figure 1.
>
> ## References
>
> Greenwood, Major, and G. Udny Yule. "An inquiry into the nature of frequency distributions representative of multiple happenings with particular reference to the occurrence of multiple attacks of disease or of repeated accidents." *Journal of the Royal statistical society* 83, no. 2 (1920): 255–79.
>
> Khan, Mohammad Emtiyaz, and Håvard Rue. "The Bayesian Learning Rule." *Journal of Machine Learning Research* 24, no. 281 (2023): 1-46.
>
> Kiral, Eren Mehmet, Thomas Moellenhoff, and Mohammad Emtiyaz Khan. "The Lie-Group Bayesian Learning Rule." *Proceedings of The 26th International Conference on Artificial Intelligence and Statistics* (2023): 3331–52.

---

> > ### Comment · Reviewer_Jttp · 2025-08-03
> >
> > Thank you for your rebuttal! I would like to keep the score as 4.

---

### Decision · Program_Chairs · 2025-09-17

**Decision:**

Accept (poster)

**Comment:**

The paper extends natural-gradient variational inference to models outside the exponential family that can be obtained by marginalising a minimal exponential-family model. The main example considered is the Student-t distribution. This marginalisation induces a quotient-manifold structure. The key contribution is to show that the Fisher-Rao metric, a natural Riemannian structure on statistical manifolds, extends from the Normal-Wishart to the Student-t manifold, and to use this insight to develop a new algorithm for variational inference with heavy tail variational families.

Reviewers agreed that the theoretical part provides interesting geometric insights and is rigorously and clearly presented. The experiments on Bayesian logistic regression required some clarification, which the authors provided in their rebuttal. It was noted that, while the experiments effectively illustrate the method, the empirical evaluation could be strengthened by going beyond logistic regression. Overall, the paper makes nonetheless a valuable and technically solid contribution.